**Alterations in microbial community composition with increasing $f$CO₂: a mesocosm study in the**
**eastern Baltic Sea**
**Katharine J. Crawfurd[1], Santiago Alvarez-Fernandez[2], Kristina D. A. Mojica[3], Ulf Riebesell[4], Corina P.**
**D. Brussaard[1,5]**
[1]{NIOZ Royal Netherlands Institute for Sea Research, Department of Marine Microbiology and
Biogeochemistry and Utrecht University, P.O. Box 59, 1790 AB Den Burg, Texel, The Netherlands}
[2]{Alfred-Wegener-Institut Helmholtz-Zentrum für Polar- und Meeresforschung, Biologische Anstalt
Helgoland, 27498, Helgoland, Germany}
[3] {Department of Botany and Plant Pathology, Cordley Hall 2082, Oregon State University, Corvallis,
Oregon 97331-29052, USA}
[4]{GEOMAR Helmholtz Centre for Ocean Research Kiel, Biological Oceanography, Düsternbrooker
Weg 20, 24105, Kiel, Germany}
[5]{Aquatic Microbiology, Institute for Biodiversity and Ecosystem Dynamics, University of
Amsterdam, P.O. Box 94248, 1090 GE Amsterdam, The Netherlands}
*Correspondence to*: K. Crawfurd (kate.crawfurd@gmail.com) and C. P. D. Brussaard
(corina.brussaard@nioz.nl)

**Abstract**

Ocean acidification, resulting from the uptake of anthropogenic carbon dioxide ($CO_2$) by the ocean, is considered a major threat to marine ecosystems. Here we examined effects of ocean acidification on microbial community dynamics in the eastern Baltic Sea, during the summer of 2012 when inorganic nitrogen and phosphorus were strongly depleted. Large volume in situ mesocosms were employed to mimic present, future and far future $CO_2$ scenarios. All six groups of phytoplankton enumerated by flow cytometry (<20 µm cell diameter) showed distinct trends in net growth and abundance with $CO_2$ enrichment. The picoeukaryotic phytoplankton groups Pico-I and II displayed enhanced abundances, whilst Pico-III, *Synechococcus* and the nanoeukaryotic phytoplankton groups were negatively affected by elevated fugacity of $CO_2$ ($f$CO$_2$). Specifically, the numerically dominant eukaryote, Pico-I, demonstrated increases in gross growth rate with increasing $f$CO$_2$ sufficient to double its abundance. Dynamics of the prokaryote community closely followed trends in total algal biomass despite differential effects of $f$CO$_2$ on algal groups. Similarly, viral abundances corresponded to prokaryotic host population dynamics. Viral lysis and grazing were both important in controlling microbial abundances. Overall our results point to a shift, with increasing $f$CO$_2$, towards a more regenerative system with production dominated by small picoeukaryotic phytoplankton.

**1 Introduction**

Marine phytoplankton are responsible for approximately half of global primary production (Field et al., 1998), with shelf sea communities contributing an average 15-30 % (Kulinski and Pempkowiak, 2011). Since the industrial revolution atmospheric carbon dioxide ($CO_2$) concentrations have increased by nearly 40 % due to anthropogenic emissions, primarily caused by the burning of fossil fuels and deforestation (Doney et al., 2009). Atmospheric $CO_2$ dissolves in the oceans where it forms carbonic acid which reduces seawater pH, a process commonly termed, ocean acidification (OA).

Currently, along with warming sea surface temperatures and changing light and nutrient conditions,
marine ecosystems face unprecedented decreases in ocean pH (Doney et al., 2009; Gruber, 2011).
Ocean acidification is considered one of the greatest current threats to marine ecosystems (Turley
and Boot, 2010) and has been shown to alter phytoplankton primary production with the direction
and magnitude of the responses dependent on community composition (eg. Hein and Sand-Jensen,
1997; Tortell et al., 2002; Leonardos and Geider, 2005; Engel et al., 2007; Feng et al., 2009; Eberlein
et al., 2017). Certain cyanobacteria, including diazotrophs, demonstrate stimulated growth under
conditions of elevated $CO_2$ (Qiu and Gao, 2002; Barcelos e Ramos et al., 2007; Hutchins, 2007;
Dutkiewicz et al., 2015). However, no consistent trends have been found for *Synechococcus* (Schulz
et al., 2017 and references therein). The responses of diatoms and coccolithophores also appear
more variable (Dutkiewicz et al., 2015 and references therein), although coccolithophore calcification
seems generally  negatively impacted (Meyer and Riebesell, 2015; Riebesell et al., 2017). OA has also
been reported to increase the abundances of small-sized photoautotrophic eukaryotes in mesocosm
experiments (Engel et al., 2007; Meakin and Wyman, 2011; Brussaard et al., 2013; Schulz et al.,

67 2017).

Recently, data regarding the effects of OA on taxa-specific phytoplankton growth rates were
incorporated into a global ecosystem model. The results emphasized that elevated $CO_2$
concentrations can cause changes in community structure by altering the competitive fitness, and
thus competition between phytoplankton groups (Dutkiewicz et al., 2015). Moreover, OA was found
to have a greater impact on phytoplankton community size structure, function and biomass than
either warming or reduced nutrient supply (Dutkiewicz et al., 2015).  Many OA studies have been
conducted using single-species under controlled laboratory conditions and therefore cannot account
for intrinsic community interactions that occur under natural conditions. Alternatively, larger-volume
mesocosm experiments allow for OA manipulation of natural communities and as such, are more
likely to capture and quantify the overall response of the natural ecosystems. To date, the majority of
these experiments started under replete nutrient conditions or received nutrient additions (Paul et
al., 2015 and references therein). Thus, little data is available for oligotrophic conditions, which are
present in ~75% of the world's oceans (Corno et al., 2007).
Whilst environmental factors such as temperature, light, nutrient and $CO_2$ concentrations regulate
gross primary production, loss factors determine the fate of this photosynthetically fixed carbon.
Grazing, sinking and viral lysis affect the cycling of elements in different manners, i.e. transferred to
higher trophic levels through grazing, carbon sequestration in deep waters and sediments, and
cellular content release by viral lysis (Wilhelm and Suttle, 1999; Brussaard et al., 2005). Released
detrital and dissolved organic matter (DOM) is quickly utilized by heterotrophic bacteria, thereby
stimulating activity within the microbial loop (Brussaard et al., 2008; Lønborg et al., 2013; Sheik et al.,
2014; Middelboe and Lyck, 2002). Consequently, bacteria may be affected indirectly by OA through
changes in the quality and/or quantity of DOM (Weinbauer et al., 2011). Viral lysis has been found to
be as important as microzooplankton grazing to the mortality of natural bacterio- and phytoplankton
(Weinbauer, 2004; Baudoux et al., 2006; Evans and Brussaard, 2012; Mojica et al., 2016). Thus far,
most studies examining the effects of OA on microzooplankton abundance and/or grazing have
found little or no direct effect (Suffrian et al., 2008; Rose et al., 2009; Aberle et al., 2013; Brussaard et
al., 2013; Niehoff et al., 2013). To our knowledge, no viral lysis rates have been reported for natural
phytoplankton communities under conditions of OA. A few studies have inferred rates based on
changes in viral abundances under enhanced $CO_2$, but the results are inconsistent (Larsen et al.,
2008; Brussaard et al., 2013). Therefore, the effect of OA on the relative share of these key loss
processes is still understudied for most ecosystems.
Here we report on the temporal dynamics of microbes (phytoplankton, prokaryotes and viruses)
under the influence of enhanced $CO_2$ concentrations in the low-salinity (around 5.7) Baltic Sea. Using
large mesocosms with in situ light and temperature conditions, the pelagic ecosystem was exposed
to a range of increasing $CO_2$ concentrations from ambient to future and far future concentrations.
The study was performed during the summer in the Baltic Sea near Tvärminne when conditions were
oligotrophic. Our data show, that over the 43 day long experiment, enhanced $CO_2$ concentrations
elicited distinct shifts in the microbial community, most notably an increase in the net growth of
small picoeukaryotic phytoplankton.

**2 Materials and Methods**
**2.1 Study site and experimental set-up**
The present study was conducted in the Tvärminne Storfjärden (59° 51.5' N, 23° 15.5' E) between 14
June and 7 August, 2012. Nine mesocosms, each enclosing ~55 $m^3$ of water, were moored in a square
arrangement at a site with a water depth of approximately 30 m. The mesocosms consisted of open
ended polyurethane bags 2 m in diameter and 18.5 m in length mounted onto floating frames
covered at each end with a 3 mm mesh. Initially, the mesocosms were kept open for 5 days to allow
for rinsing and water exchange while excluding large organisms from entering with the 3 mm mesh.
During this time, the bags were positioned such that the tops were submerged 0.5 m below the
water surface and the bottoms reached down to 17 m depth in the water column. Photosynthetically
active radiation (PAR) transparent plastic hoods (open on the side) prevented rain and bird droppings
from entering the mesocosms, which would affect salinity and nutrients, respectively. Five days
before the $CO_2$ treatment was to begin, the water column of the mesocosms was isolated from the
influence of the surrounding water. To do so, the 3 mm mesh was removed and sediment traps (2 m
long) were attached to close off the bottom of the mesocosms. The top ends of the bags were raised
and secured to the frame 1.5 m above the water surface to prevent water entering via wave action.
The mesocosms were then bubbled with compressed air for 3.5 min, to remove salinity gradients and
ensure that the water body was fully homogeneous.
The present manuscript includes results from six of the original mesocosms, due to the unfortunate
loss of three mesocosms which were compromised by leakage. The mean fugacity of $CO_2$ ($fCO_2$)
during the experiment, i.e. days 1-43, for the individual mesocosms were as follows: M1, 365 µatm;
M3, 1007 µatm; M5, 368 µatm; M6, 821 µatm; M7, 497 µatm; M8, 1231 µatm (Table 1). The gradient
of non-replicated $fCO_2$ of the present study (as opposed to a smaller number of replicated treatment
levels) was selected as a balance between the necessary, but manageable, number of mesocosms
and minimizing the impact of the high potential for loss of mesocosms to successfully address the
underlying questions of the study (Schulz et al., 2013). Moreover, it maximizes the potential of
identifying a threshold $fCO_2$ level concentration, if present (by allowing for a larger number of
treatment levels). Carbon dioxide manipulation was carried out in four steps and took place between
days 0 to 4 until the target $fCO_2$ was reached. Initial $fCO_2$ was 240 µatm. For $fCO_2$ manipulations, 50
µm filtered natural seawater was saturated with $CO_2$ and then injected evenly throughout the depth
of the mesocosms as described by Riebesell et al. (2013). Two mesocosms functioned as controls and
were treated in a similar manner using only filtered seawater. On day 15, a supplementary $fCO_2$
addition was made to the top 7 m of mesocosms numbered 3, 6, and 8 to replace $CO_2$ lost due to
outgassing (Paul et al., 2015; Spilling et al., 2016). Throughout this study we refer to $fCO_2$ which
accounts for the non-ideal behavior of $CO_2$ gas and is considered the standard measurement
required for gas exchange (Pfeil et al., 2012).
Initial nutrient concentrations were 0.05 µmol $L^{-1}$, 0.15 µmol $L^{-1}$, 6.2 µmol $L^{-1}$ and 0.2 µmol $L^{-1}$ for
nitrate, phosphate, silicate and ammonium, respectively. Nutrient concentrations remained low for
the duration of the experiment (Paul et al., 2015, this issue) and no nutrients were added. Salinity
was relatively constant around 5.7. Temperature was more variable; on average temperature within
the mesocosms (0-17 m) increased from ~8 °C to a maximum on day 15 of ~15 °C and then decreased
again to ~8 °C by day 30. For further details of the experimental set-up, carbonate chemistry
dynamics and nutrient concentrations throughout the experiment we refer to the general overview
paper by Paul et al. (2015).
Collective sampling was performed every morning using depth integrated water samplers (IWS,
HYDRO-BIOS, Kiel). These sampling devices were gently lowered through the water column collecting
~5 L of water gradually between 0-10 m (top) or 0-17 m (whole water column). Water was collected
from all mesocosms and the surrounding water. Subsamples were obtained for enumeration of
phytoplankton, prokaryotes and viruses. Samples for viral lysis and grazing experiments were taken
from 5 m depth using a gentle vacuum-driven pump system. Samples were protected against sunlight
and warming by thick black plastic bags containing wet ice. Samples were processed at in situ
temperature (representative of 5 m depth) under dim light and handled using nitrile gloves. As viral
lysis and grazing rates were determined from samples taken from 5 m depth, samples for microbial
abundances reported here were taken from the top 10 m integrated samples.
The experimental period has been divided into four phases based on major physical and biological
changes (Paul et al., 2015): Phase 0 before $CO_2$ addition (days -5 to 0), Phase I (days 1-16), Phase II
(days 17-30) and Phase III (days 31-43). Throughout this manuscript the data are presented using
three colors (blue, grey and red), representing low (mesocosms M1 and M5) intermediate (M6 and
M7) and high (M3 and M8) $f\mathrm{CO}_2$ levels (Table 1).

**2.2 Microbial abundances**
Microbes were enumerated using a Becton Dickinson FACSCalibur flow cytometer (FCM) equipped
with a 488 nm argon laser. The samples were stored on wet ice and in the dark until counting. The
photoautotrophic cells (<20 μm) were counted directly using fresh seawater and were discriminated
by their autofluorescent pigments. Six phytoplankton clusters were differentiated based on the
bivariant plots of either chlorophyll (red autofluorescence) or phycoerythrin (orange
autofluorescence, for *Synechococcus* and Pico-III) against side scatter. The size of the different
phytoplankton clusters was determined by gentle filtration through 25 mm diameter polycarbonate
filters (Whatman) with a range of pore sizes (12, 10, 8, 5, 3, 2, 1, and 0.8 μm) according to Veldhuis
and Kraay (2004).  Average cell sizes for the different phytoplankton groups were 1, 1, 3, 2.9, 5.2, and
8.8 μm diameter for the prokaryotic cyanobacteria *Synechococcus* spp. (SYN), picoeukaryotic
phytoplankton I, II and III (Pico-I-III), and nanoeukaryotic phytoplankton I, and II (Nano-I, II),
respectively. Pico-III was discriminated from Pico-II (comparable average cell size) by a higher orange
autofluorescence signature, potentially representing small-sized cryptophytes (Klaveness, 1989);
alternatively large single cells or microcolonies of *Synechococcus* (Haverkamp et al., 2009). The
cyanobacterial species *Prochlorococcus* spp. were not observed during this experiment. Counts were
converted to cellular carbon by assuming a spherical shape equivalent to the average cell diameters
determined from size fractionations and applying conversion factors of 237 fg C $\mu m^{-3}$ (Worden et al.,
2004) and 196.5 fg C $\mu m^{-3}$ (Garrison et al., 2000) for pico- and nano-sized plankton, respectively.
Microbial net growth and loss rates were derived from exponential regressions of changes in the cell
abundances over time.
Abundances of prokaryotes and viruses were determined from 0.5 % glutaraldehyde fixed, flash
frozen (-80 °C) samples according to Marie et al. (1999) and Brussaard (2004), respectively. The
prokaryotes include heterotrophic bacteria, archaea and unicellular cyanobacteria, the latter
accounting for maximal 10 % of the total abundance in our samples, as indicated by their
autofluorescence. Briefly, thawed samples were diluted with sterile autoclaved Tris-EDTA buffer (10
mM Tris-HCl and 1 mM EDTA, pH 8.2; Mojica et al., 2014) and stained with the green fluorescent
nucleic acid-specific dye SYBR-Green I (Invitrogen Inc.) to a final concentration of the commercial
stock of $1.0 \times 10^{-4}$ (for prokaryotes) or $0.5 \times 10^{-4}$ (for viruses). Virus samples were stained at 80 °C for
10 min and then allowed to cool for 5 min at room temperature in the dark. Prokaryotes were
stained for 15 min at room temperature in the dark (Brussaard, 2004). Prokaryotes and viruses were
discriminated in bivariate scatter plots of green fluorescence versus side scatter. Final counts were
corrected for blanks prepared and analyzed in a similar manner as the samples. Two groups of
prokaryotes were identified by their stained nucleic acid fluorescence, referred here on as low (LNA)
and high (HNA) fluorescence prokaryotes.

**2.3 Viral lysis and grazing**
Microzooplankton grazing and viral lysis of phytoplankton was determined using the modified
dilution assay, based on reducing grazing and viral lysis mortality pressure in a serial manner allowing
for increased phytoplankton growth (over the incubation period) with dilution (Mojica et al., 2016).
Briefly, two dilution series were created in clear 1.2 L polycarbonate bottles by gently mixing 200 µm
sieved whole seawater with either 0.45 µm filtered seawater (i.e. microzooplankton grazers
removed) or 30 KDa filtered seawater (i.e. grazers and viruses removed) to final dilutions of 20, 40,
70 and 100 %. The 0.45 µm filtrate was produced by gravity filtration of 200 µm mesh sieved
seawater through a 0.45 µm Sartopore capsule filter. The 30 KDa ultrafiltrate was produced by
tangential flow filtration of 200 µm pre-sieved seawater using a 30 kDa Vivaflow 200 PES membrane
tangential flow cartridge (Vivascience). All treatments were performed in triplicate. Bottles were
suspended next to the mesocosms in small cages at 5 m depth for 24 hours. Subsamples were taken
at 0 and 24 h, and phytoplankton abundances of the grazing series (0.45 µm diluent) were
enumerated by flow cytometry. Due to time constraint, the majority of the samples of the 30 kDa
series were fixed with 1 % (final concentration) formaldehyde : hexamine solution (18 % v/v : 10 %
w/v), for 30 min at 4 °C, flash frozen in liquid nitrogen and stored at −80 °C until flow cytometry
analysis in the home laboratory. Fixation had no significant effect (student's t-tests, p-value >0.05) as
tested periodically against fresh samples. The modified dilution assay was only run for Mesocosms 1
(low $f$CO$_2$) and 3 (high $f$CO$_2$) due to the logistics of handling times. Experiments were performed until
day 31. Grazing rates and the combined rate of grazing and viral lysis were estimated from the slope
of a regression of phytoplankton apparent growth versus dilution of the 0.45 µm and 30 kDa series,
respectively. A significant difference between the two regression coefficients (as tested by analysis of
covariance) indicated a significant viral lysis rate. Phytoplankton gross growth rate, in the absence of
grazing and viral lysis, was derived from the y-intercept of the 30 kDa series regression. Similarly,
significant differences between mesocosms M1 and M3 (low and high $f$CO$_2$) were determined
through analysis of covariance of the dilution series for the two mesocosms. A significance threshold
of 0.05 was used and significance is denoted throughout the manuscript by an asterisk (*).
Occasionally, the regression of apparent growth rate versus fraction of natural water resulted in a
positive slope (thus no reduction in mortality with dilution). In addition, very low phytoplankton
abundances can also prohibit statistical significance of results. Under such conditions dilution
experiments were deemed unsuccessful (see for limitations of the modified dilution method,
Baudoux et al., 2006; Kimmance and Brussaard, 2010; Stoecker et al., 2015).
Viral lysis of prokaryotes was determined according to the viral production assay (Wilhelm et al.,
2002; Winget et al., 2005). After reduction of the natural virus concentration, new virus production
by the natural bacterial community is sampled and tracked over time (24 h). Briefly, free viruses were
reduced from a 300 ml sample of whole water by re-circulation over a 0.2 μm pore size polyether
sulfone membrane (PES) tangential flow filter (Vivaflow 50, Vivascience) at a filtrate expulsion rate of
40 ml min$^{-1}$. The concentrated sample was then reconstituted to the original volume using virus-free
seawater. This process was repeated a total of three times to gradually wash away viruses. After the
final reconstitution, 50 ml aliquots were distributed into six polycarbonate tubes. Mitomycin C
(Sigma-Aldrich) (final concentration, 1 μg ml$^{-1}$, maintained at 4 °C), which induces lysogenic bacteria
(Weinbauer and Suttle, 1996) was added to a second series of triplicate samples for each mesocosm.
A third series of incubations with 0.2 μm filtered samples was used as a control for viral loss (e.g.
viruses adhering to the tube walls) and showed no significant loss of free viruses during the
incubations. At the start of the experiment, 1 ml subsamples were immediately removed from each
tube and fixed as previously described for viral and bacterial abundance. The samples were dark
incubated at in situ temperature and 1 ml subsamples were taken at 3 h, 6 h, 9 h, 12 h and 24 h.
Virus production was determined from linear regression of viral abundance over time. Viral
production due to induction of lysogeny was calculated as the difference between production in the
unamended samples and production of samples to which mitomycin C was added. Although
mortality experiments were initially planned to be employed for mesocosms 1, 2, and 3 representing
low, mid and high $f$CO$_2$ conditions, mesocosm 2 was compromised due to leakage. Additionally, due
to logistical reasons assays were only performed until day 21.
To determine grazing rates on prokaryotes, fluorescently labeled bacteria (FLBs) were prepared from
enriched natural bacterial assemblages (originating from the North Sea) labeled with 5-([4,6-
Dichlorotriazin-2-yl]amino) fluorescein (DTAF, 36565 Sigma-Aldrich 40 μg ml$^{-1}$) according to Sherr et
al. (1993). Frozen ampoules of FLB (1-5 % of total bacterial abundance) were added to triplicate 1 L
incubation bottles containing whole water gently passed through 200 µm mesh. Twenty ml samples
were taken immediately after addition (0 h) and the headspace was removed by gently squeezing air
from the bottle. The 1 L bottles were incubated on a slow turning wheel (1 rpm) at in situ light and
temperature conditions (representative of 5 m depth) for 24 h. Sampling was repeated after 24 h. All
samples were fixed to a 1 % final concentration of gluteraldehyde (0.2 µm filtered; 25 % EM-grade),
stained (in the dark for 30 min at 4 °C) with 4',6-Diamidino-2-Phenylindole, Dihydrochloride (DAPI)
solution (0.2 µm filtered; Acrodisc ®25 mm Syringe filters, PALL Life Sciences; 2 µg ml$^{-1}$ final
concentration; Sherr et al., 1993) and filtered onto 25 mm, 0.2 µm black polycarbonate filters (GE
Healthcare life sciences). Filters were then mounted on microscopic slides and stored at -20 °C until
analysis. FLBs present on a ~0.75 mm$^2$ area were counted using a Zeiss Axioplan 2 microscope.
Grazing ($\mu$ d$^{-1}$) was measured according to $N_{T24} = N_{T0} * e^{-\mu t}$, where $N_{T24}$ and $N_{T0}$ are the number of FLBs
present at 24 h and 0 h, respectively.

**2.4 Statistics**
Non-metric multidimensional scaling (NMDS) was used to follow microbial community development
in each mesocosm over the experimental period. NMDS is an ordination technique which represents
the dissimilarities obtained from an abundance data matrix in a *2*-dimensional space (Legendre and
Legendre, 1998). In this case, the data matrix was comprised of abundance data for each
phytoplankton group in each mesocosm for every day of sampling. The treatment effect was
assessed by analysis of similarity (ANOSIM; Clarke, 1993) and inspection of the NMDS biplot. ANOSIM
compares the mean of ranked dissimilarities of mesocosms between $f$CO$_2$ treatments (low: 1, 5, 7;
high: 6, 3, 8) to the mean of ranked dissimilarities within treatments per phase. The NMDS plots
allowed divergence periods in the development and community composition between treatments to
be visually assessed (period 1 from day 3-13 and period 2 from days 16-24). Net growth rates of each
of the different microbial groups were calculated for these identified divergence periods.
Relationships between net growth rates and peak cell abundances with $f\text{CO}_2$ were evaluated by
linear regression against the average $f\text{CO}_2$ per mesocosm during each period or peak day. A
generalized linear model was used to test the relationship between prokaryote abundance and
carbon biomass with an ARMA correlation structure of order 3 to account for temporal
autocorrelation. The model fulfilled all assumptions such as homoscedasticity and avoiding
autocorrelation of the residuals (Zuur et al., 2007). A significance threshold of $p \leq 0.05$ was used and
significance is denoted by an asterisk (*). All analyses were performed using the statistical software
program R, using packages nlme (Pinheiro et al., 2017) and vegan (Oksanen et al., 2017) (R core
Team, 2017). Where average of low and high mesocosm abundance data are reported, values
represent the average of mesocosms 1, 5, 7 (mean $f\text{CO}_2$ 365-497 µatm) and 6, 3, 8 (821-1231 µatm),
respectively.

**3 Results**
**3.1 Total phytoplankton dynamics in response to $\text{CO}_2$ enrichment**
During Phase 0, low variability in phytoplankton abundances in the different mesocosms (1.5 ± 0.05 x
$10^5$ ml$^{-1}$) indicated good replicability of initial conditions prior to $\text{CO}_2$ manipulation (Fig. 1). This was
further supported by the high similarity between microbial communities in the different mesocosms
as indicated by the tight clustering of points in the NMDS plot during this period (Fig. 2). During
Phase 0, the phytoplankton community (<20 µm) was dominated by pico-sized autotrophs, with the
prokaryotic cyanobacteria *Synechococcus* (SYN) and Pico-I accounting for 69 % and 27 % of  total
phytoplankton abundance, respectively. After $\text{CO}_2$ addition, there were two primary peaks in
phytoplankton, which occurred on day 4 in Phase I and day 24 in Phase II (Fig. 1a). The phytoplankton
community became significantly different over time in the different treatments (ANOSIM, p=0.01,
Fig. 2). Two periods were identified based on their divergence (Fig.2), the first (NMDS-based period
1) followed the initial peak in abundance (days 3-13) with highest abundances occurring in the
elevated $\text{CO}_2$ mesocosms (Fig. 1a). During the second period (NMDS-based period 2, days 16-24),
abundances were higher in the low $f$CO$_2$ mesocosms (Fig. 1a). In general the NMDS plot shows that
throughout the experiment, mesocosm M1 followed the same basic trajectory as mesocosms M5 and
M7, whilst mesocosm M3 followed M6 and M8 (Fig. 2). Thus, the two mesocosms (representing high
and low $f$CO$_2$ treatments) deviated from each other during Phase I and were clearly separated during
Phases II and III (Fig. 2).
Phytoplankton abundances in the surrounding water started to differ from the mesocosms during
Phase 0 (on average 44 % lower) which was primarily due to lower abundances of SYN. This effect
was seen from day -1, prior to CO$_2$ addition but following bubbling with compressed air (day -5). On
day 15, a deep mixing event occurred as a result of storm conditions (with consequent alterations in
temperature and salinity) and as a result phytoplankton abundances in the surrounding open water
diverged more strongly from the mesocosms but remained similar in their dynamics (Fig. 3).
Microbial abundances in the 0-17 m samples were slightly lower but showed very similar dynamics to
those in the 0-10 m samples (Fig. S1).

**3.1.1 *Synechococcus***
The prokaryotic cyanobacteria *Synechococcus* (SYN) accounted for the majority of total abundance,
i.e. 74 % averaged across all mesocosms over the experimental period. Abundances of SYN showed
distinct variability between the different CO$_2$ treatments, starting on day 7, with the low CO$_2$
mesocosms exhibiting nearly 20 % lower abundances between days 11-15 as compared to high $f$CO$_2$
mesocosms (Fig. 3a). SYN net growth rates during days 3-13 (NMDS-based period 1) were positively
correlated with CO$_2$ (p=0.10, R$^2$=0.53; Table 2, Fig. S2a). One explanation for higher net growth rates
at elevated CO$_2$ could be the significantly (p<0.05) higher grazing rate in the low $f$CO$_2$ mesocosm M1
(0.56 d$^{-1}$) compared to the high $f$CO$_2$ M3 (0.27 d$^{-1}$) as measured on day 10 (Fig. 4a). After day 16, SYN
abundances increased in all mesocosms and during this period (days 16-24) net growth rates had a
significant negative correlation to $f$CO$_2$ (p=0.05, R$^2$= 0.63; Figs. 3a, Table 2 and Fig. S3a).
Consequently, the net increase in SYN abundances during this period was on average 20 % higher at
low $f$CO$_2$ compared to high $f$CO$_2$. This corresponded to higher total loss rates in high $f$CO$_2$ treatments
measured on day 17 (0.33 vs 0.17 d$^{-1}$ for M3 and M1, respectively; Fig. 4a). The higher net growth
most likely led to the peak in SYN abundance observed on day 24 (max. 4.7 x 10$^5$ ml$^{-1}$), which was
negatively correlated with $f$CO$_2$ (p=0.01, R$^2$=0.80; Table 3, Fig. S4a). After this period (days 24-28),
SYN abundances declined at comparable rates in the different mesocosms, irrespective of $f$CO$_2$ (Fig.
3a). Abundances in the low $f$CO$_2$ mesocosms remained higher into Phase III (Fig. 3a). SYN abundances
in the surrounding water were generally lower than in the mesocosms, with the exception of days

345  17-21.


**3.1.2 Picoeukaryotes**
In contrast to the prokaryotic photoautotrophs, the eukaryotic phytoplankton community showed a
strong positive response to elevated $f$CO$_2$ (Fig. 1b). Pico-I was the numerically dominant group of
eukaryotic phytoplankton, accounting for an average 21-26 % of total phytoplankton abundances.
Net growth rates leading up to the first peak in abundance (from day 1 to 5) had a strong positive
correlation with $f$CO$_2$ (p<0.01, R$^2$=0.90; Fig. 3b, Table 3, Fig. S5a). Accordingly, the peak on day 5
(max. 1.1 x 10$^5$ ml$^{-1}$; Fig. 3b) was also correlated positively with $f$CO$_2$ (p=0.01, R$^2$=0.81; Table 3, Fig.
S4b). During Phase I, from days 3-13 (i.e. NMDS-based period 1), net growth rates of Pico-I remained
positively correlated to CO$_2$ concentration (p=0.01, R$^2$=0.80; Table 2, Fig. S2b). However, during this
period there was also a decline in abundance (days 5-9; p<0.01, R$^2$=0.89; Table 3, Fig. S5b) with 23 %
more cells lost in the low $f$CO$_2$ mesocosms. Accordingly, following this period, gross growth rate was
significantly higher in the high $f$CO$_2$ mesocosm M3 as compared to the low $f$CO$_2$ mesocosm M1 (day
10, p<0.05; Fig. 4b). Pico-I abundances in the surrounding open water started to deviate from the
mesocosms after day 10, and were on average around half that of the low $f$CO$_2$ mesocosms (Fig. 3b).
Following a brief increase (occurring between days 11-13) correlated to $f$CO$_2$ (p<0.01, R$^2$=0.94; Table
3, Fig. S4c), abundances declined sharply between days 13-16 (Fig. 3b), coinciding with a significantly
higher total mortality rate in the high $f$CO$_2$ mesocosm M3 (day 13; Fig. 4b). Viral lysis was a
substantial loss factor relative to grazing, for this group, comprising an average 45 % and 70 % of
total losses in M1 and M3, respectively (Table S1). During NMDS-based period 2, net growth rates of
Pico-I were significantly higher at high $f\mathrm{CO_2}$ (p=0.05, $R^2$=0.64; Table 2, Fig S3b). By day 21,
abundances in the high $f\mathrm{CO_2}$ mesocosms were (on average) ~2-fold higher than at low $f\mathrm{CO_2}$
(maximum abundances 8.7 x $10^4$ ml$^{-1}$ and 5.9 x $10^4$ ml$^{-1}$ for high and low $f\mathrm{CO_2}$ mesocosms; p=0.01,
$R^2$=0.84; Table 3, Fig. S4d). Standing stock of Pico-I remained high in the elevated $f\mathrm{CO_2}$ mesocosms
for the remainder of the experiment (7.9 x $10^4$ vs 4.3 x $10^4$ ml$^{-1}$ on average for high and low $f\mathrm{CO_2}$
mesocosms, respectively; Fig. 3b). Additionally, gross growth rates during this final period were
relatively low (0.14 and 0.16 d$^{-1}$ in M1 and M3, respectively) and comparable to total loss rates
(averaging 0.13 and 0.10 d$^{-1}$ over days 25-31, for M1 and M3, respectively; Fig. 4b).
Another pico-eukaryote group, Pico-II, slowly increased in abundance until day 13, when it increased
more rapidly (Fig. 3c). Gross growth rates measured during Phase I were high (0.69 and 0.72 d$^{-1}$ on
average in the low and high $f\mathrm{CO_2}$ mesocosms M1 and M3, respectively; Fig. 4c), and comparable to
loss processes (0.46 and 0.58 d$^{-1}$), indicative of a relatively high turnover rate of production. Overall
net growth rates during days 3-13 (NMDS-based period 1) did not correlate to $\mathrm{CO_2}$ (p=0.52, $R^2$=0.11;
Table 2, Fig. S2c). However, during periods of rapid increases in net growth, abundances were
positively correlated to $\mathrm{CO_2}$ concentration (days 12-17; p=0.01, $R^2$=0.82; Table 3, Fig. S5c).
Accordingly, the peak in abundances of Pico-II on day 17 displayed a distinct positive correlation with
$f\mathrm{CO_2}$ (p<0.01, $R^2$=0.93; Table 3, Fig. S4e), with maximum abundances of 4.6 x $10^3$ ml$^{-1}$ and 3.4 x $10^3$
ml$^{-1}$ for the high and low $f\mathrm{CO_2}$ mecosoms, respectively (Fig. 3c). In M8 (the highest $f\mathrm{CO_2}$ mesocosm),
abundances increased for an extra day with the peak occurring on day 18, resulting in an average 23
% higher abundances. During the decline of the Pico-II peak (days 16-24), net growth rates were
negatively correlated with $f\mathrm{CO_2}$ (p=0.10, $R^2$=0.52; Table 2, Fig S3c). Moreover, the rate of decline was
faster for the high $f\mathrm{CO_2}$ mesocosms during days 18-21 (p<0.01, $R^2$=0.85). The Pico-II abundances in
the surrounding water were comparable to the mesocosms during Phases 0 and I, lower during
Phase II and higher during Phase III (Fig. 3c).
Pico-III exhibited a short initial increase in abundances in the low $f$CO$_2$ treatments, resulting in nearly
2-fold higher abundances at low $f$CO$_2$ by day 3 compared to the high $f$CO$_2$ treatment (Fig. 3d). After
this initial period, net growth rates of this group had a significant positive correlation with $f$CO$_2$ (days
3-13; p=0.04, $R^2$=0.67; Table 2, Fig. S2d). In general, during Phase I gross growth (p<0.01, days 1, 3,
10; Fig. 4d) and total mortality (p<0.05, days 1, 6, 10; Fig. 4d) were significantly higher in the low $f$CO$_2$
mesocosm M1, as compared to the high $f$CO$_2$ mesocosm M3 resulting in low net growth rates. During
Phase II (days 16-24, NMDS-based period 2) the opposite occurred; i.e. net growth rates were
negatively correlated with $f$CO$_2$ (p<0.01, $R^2$=0.86; Table 2, Fig S.3d). Maximum Pico-III abundances
(day 24: 4.2 x 10$^3$ and 8.3 x 10$^3$ ml$^{-1}$ for high and low $f$CO$_2$) had a strong negative correlation with $f$CO$_2$
(p<0.01, $R^2$=0.91; Table 3, Fig. S4f). Pico-III abundances remained noticeably higher in the low $f$CO$_2$
mesocosms during Phases II and III (on average 80 %; Fig. 3d).  Unfortunately, almost half of the
mortality assays in this second half of the experiment failed (see Materials and Methods), but the
successful assays suggest that losses were minor (<0.15 d$^{-1}$; Fig. 4d) and primarily due to grazing, as
no significant viral lysis was detected (Table S1).

**3.1.3 Nanoeukaryotes**
Nano-I showed maximum abundances (4.3 ± 0.4 x 10$^2$ ml$^{-1}$) on day 6 (except M1 which peaked on day
5), independent of $f$CO$_2$ (p=0.23, $R^2$=0.33; Fig. 3e). There was, however, a negative correlation of net
growth rate with $f$CO$_2$ during days 3-13 (NMDS-based period 1; p=0.01, $R^2$=0.79; Table 2, Fig. S2e). A
second major peak in abundance of Nano-I occurred on day 17, with markedly higher numbers in the
low $f$CO$_2$ mesocosms (4.1 x 10$^2$ ml$^{-1}$ as compared to 2.4 x 10$^2$ ml$^{-1}$ in high $f$CO$_2$ mesocosms; p=0.04,
$R^2$=0.67; Fig. 3e, Table 3 and Fig. S4g). Total loss rates in the high $f$CO$_2$ mesocosm M3 on days 6 and
10 were 2.3-fold higher compared to the low $f$CO$_2$ mescososm M1 (Fig. 4e), which may help to
explain this discrepancy in total abundance between low and high $f$CO$_2$ mesocosms. Viral lysis made
up to 98 % of total losses in the high $f$CO$_2$ mesocosm M3 during this period, whilst in M1 viral lysis
was only detected on day 13 (Table S1). Peak abundances (around 5.0 x 10$^2$ ml$^{-1}$) were much lower
compared to those in the surrounding waters (max ~2.4 x $10^3$ ml$^{-1}$; Figs. 3e and S6a). During Phase II,
Nano-I abundances in the surrounding waters displayed rather erratic dynamics compared to those
of the mesocosms, but converged during certain periods (e.g. days 19-22). No significant relationship
was found between net loss rates and $f$CO$_2$ for the second NMDS-based period (p=0.26, R$^2$=0.30;
Table 2, Fig S.3e). At the end of Phase II, abundances were similar in all mesocosms but diverged
again during Phase III (days 31-39) due primarily to a negative effect of CO$_2$ on Nano-I abundances, as
depicted in the average 36 % reduction in Nano-I.
The temporal dynamics of Nano-II, the least abundant phytoplankton group analysed in our study,
displayed the largest variability (Fig. 3f), perhaps due to the spread of this cluster in flow cytographs
(which may indicate that this group represents several different phytoplankton species). No
significant relationship was found between net growth rate and $f$CO$_2$ for this group for the two
NMDS-based periods (Table 2, Figs S2f and S3f) nor with the peak in abundances on day 17 (p=0.13,
R$^2$=0.46; Fig. S4h). Moreover, no consistent trend was detected in mortality rates (Fig. 4f). Similar to
Nano-I, abundances in the surrounding water were often higher than in the mesocosms (max 3.5 x
$10^2$ ml$^{-1}$ vs 1.1 x $10^4$ ml$^{-1}$, respectively; Figs. 3f and S6b).

**3.1.4 Algal carbon biomass**
The mean combined biomass of Pico-I and Pico-II showed a strong positive correlation with $f$CO$_2$
throughout the experiment (p<0.05, R$^2$=0.95; Fig. 5a), an effect already noticeable by day 2. Their
biomass in the high $f$CO$_2$ mesocosms was, on average 11 % higher than in the low $f$CO$_2$ mesocosms
between days 10-20 and 20 % higher between days 20-39. Conversely, the remaining algal groups
showed an average 10 % reduction in carbon biomass at enhanced $f$CO$_2$ (days 3-39, the sum of SYN,
Pico-III, Nano-I and II ; p<0.01; Fig. 5b). The most notable response was found for the biomass of
Pico-III, which showed an immediate negative response to CO$_2$ addition (Fig. S7a) and remained, on
average, 29 % lower throughout the study period (days 2-39). For Nano-I and II the lower carbon
biomass only became apparent during the end of Phase I and beginning of Phase II (days 14-20; Fig.
S7b). Due to its small cell size, the numerically dominant SYN accounted for an average of 40 % of
total carbon biomass.

**3.2 Prokaryote and virus population dynamics**
Prokaryote abundance in the mesocosms was positively related to total algal biomass independent of
treatment (p<0.05, $R^2$=0.33; Fig. 8) and generally followed total algal biomass (Fig. S7c). The initial
increase in total prokaryote abundances occurred during the first few days following the closure of
the mesocosms (Fig. 6a). This was primarily due to increases in the HNA-prokaryote group (Fig. 6b)
which displayed higher net growth rates (0.22 $d^{-1}$) compared to the LNA-prokaryotes (0.14 $d^{-1}$ on days
-3 to 3; Fig. 6c). A similar, albeit somewhat lower, increase was also recorded in the surrounding
waters (Fig. 6a). The decline of the first peak in prokaryote abundances coincided with the decay in
phytoplankton abundance/biomass (Figs. 1a and S7c). Concurrently the share of viral lysis increased,
representing 37-39 % of total mortality on day 11 (Fig. 7b). No measurable rates of lysogeny were
found for the prokaryotic community during the experimental period (all phases). From days 10 to 15
prokaryote dynamics (total, HNA and LNA) became noticeably affected by $CO_2$ concentration with a
significant positive correlation between net growth and $f\mathrm{CO_2}$ during Phase I (days 3-13 NMDS-based
period 1; Table 2, Fig. S2 g and h). In the higher $f\mathrm{CO_2}$ mesocosms, the decline in prokaryote
abundance occurring between days 13 and 16 (Fig. 6a) was largely (70 %) due to decreasing HNA-
prokaryote numbers (Fig. 6b). The grazing was 1.6-fold higher in the high $f\mathrm{CO_2}$ mesocosm M3
compared to M1 (0.36 ± 0.13 and 0.14 ± 0.08 $d^{-1}$ on day 14; Fig. 7a). At the same time, virus
abundance increased in the high $f\mathrm{CO_2}$ mesocosms (Fig. 6d).
During Phase II, prokaryote abundances increased steadily until day 24 (for both HNA and LNA),
corresponding to increased algal biomass (Figs. 6 and S7c) and lowered grazing rates (Fig. 7a).
Specifically, during days 16-24 (NMDS-based period 2), the HNA-prokaryotes showed an average 10
% higher abundances in the low, as compared to the high $f\mathrm{CO_2}$ mesocosms (Fig. 6b). However, a
significant negative correlation of net growth rates and $f\mathrm{CO_2}$ was only found for LNA (Table 2, Fig S3g
and h). No significant differences in loss rates between M1 and M3 were found during Phase II
(p=0.22, 0.46 days 18 and 21 respectively; Fig. 7). Halfway through Phase II (day 24), the prokaryote
abundance in the surrounding water leveled off (Fig. 6a). Prokaryote abundance ultimately declined
during days 28-35 (Fig. 6a), whereby the net growth of LNA was again negatively correlated with
enhanced $CO_2$ (p=0.02, $R^2$=0.76; Table 2, Fig S3g). Unfortunately, no experimental data on grazing
and lysis of prokaryotes is present after day 25. However, viral abundances increased steadily at 2.2 x
$10^6$ $d^{-1}$, concomitant with a decline in prokaryote abundance (Fig. 6a and d). There was no significant
correlation between viral abundances and $f$CO$_2$ during Phases II and III (p=0.36, $R^2$=0.21).

**4 Discussion**
In most experimental mesocosm studies, nutrients have been added to stimulate phytoplankton
growth (Schulz et al., 2017) therefore little data exists for oligotrophic phytoplankton communities.
In this study, we describe the impact of increased $f$CO$_2$ on the brackish Baltic Sea microbial
community during summer (nutrient depleted; Paul et al., 2015). Small-sized phytoplankton
numerically dominated the autotrophic community, in particular SYN and Pico-I (both about 1 μm
cell diameter). Our results demonstrate variable effects of $f$CO$_2$ manipulation on temporal
phytoplankton dynamics, dependent on phytoplankton group. In particular, Pico-I and Pico-II showed
significant positive responses, whilst the abundances of Pico-III, SYN and Nano-I were negatively
influenced by elevated $f$CO$_2$. The impact of OA on the different groups was, at times, a direct
consequence of alterations in gross growth rate, whilst overall phytoplankton population dynamics
could be explained by the combination of growth and losses. OA effects on community composition
in these systems may have consequences on both the food web and biogeochemical cycling.

**Comparison with surrounding waters**
During Phase 0, the microbial assemblage showed good replicability between all mesocosms,
however they had already began to deviate from the community in the surrounding waters. This was
most likely a consequence of water movement altering the physical conditions and biological
composition of the surrounding water body. The dynamic nature of water movement in this region
has been shown to alter the entire phytoplankton community several times over within a few
months, due to fluctuations in nutrient supply, advection, replacement/mixing of water masses and
water temperature (Lips and Lips, 2010). Alternatively, effects of enclosure and the techniques
(bubbling) used to ensure a homogenous water column may have stimulated SYN within the
mesocosms, which has been found to occur in several mesocosm experiments (Paulino et al., 2008;
Gazeau et al., 2017). By Phases II and III, the microbial abundances within the mesocosms were
distinctly different from the surrounding waters, with generally fewer SYN and Pico-I, and more
Nano-I and Nano-II. Our statistical analysis shows that during this time, there was little similarity
between the surrounding waters and mesocosms regardless of the $CO_2$ treatment level. Thus, the
deviations during this time were most likely due to an upwelling event in the archipelago (days 17-
30; Paul et al., 2015). Cold, nutrient-rich deep water has been shown to upwell during summer, with
profound positive influence on ecosystem productivity (Nômmann et al., 1991; Lehman and Myrberg,
2008). A relaxation from nutrient limitation in vertically stratified waters disproportionately favours
larger-sized phytoplankton, due to their higher nutrient requirements and lower capacity to compete
at low concentrations dictated by their lower surface to volume ratio (Raven, 1998; Veldhuis et al.,
2005). Inside the mesocosms, which were isolated from upwelled nutrients, picoeukaryotes
dominated similar to a stratified water column.  Following this upwelling event, the pH of the
surrounding waters dropped from 8.3 to 7.8, a level comparable to the highest $CO_2$ treatment (M8)
on day 32 (Paul et al., 2015). Suggesting that other factors contributed to the observed differences
between mesocosms and surrounding water, than can be accounted for by $CO_2$ concentration alone
e.g. nutrients. Alternatively, the magnitude and source of mortality occurring in the surrounding
water may have been altered, compared to within the mesocosms, after such an upwelling event.
Although the grazer community in the surrounding waters was not studied during this campaign, it is
likely that the grazing community was completely restructured during the upwelling event (Uitto et
al., 1997). It is nonetheless noteworthy that the phytoplankton groups with distinct responses to $CO_2$
enrichment (either positive or negative) in the low (ambient) $fCO_2$ mesocosms diverged from those in
the surrounding water before the upwelling event occurred.

**Phytoplankton dynamics**
*Synechococcus* showed significantly lower net growth rates and peak abundances at higher $fCO_2$.
Both in laboratory and mesoscosm experiments, *Synechococcus* has been reported to have diverse
responses to $CO_2$, with approximately equal accounts of positive (Lu et al., 2006; Schulz et al., 2017),
negative (Paulino et al., 2008; Hopkins et al., 2010; Traving et al., 2014,) and insignificant changes (Fu
et al., 2007; Lu et al., 2006) in net growth rate with $fCO_2$. This variable response is probably due, at
least in part, to the broad physiological and genetic diversity of this species. In the Gulf of Finland
alone, 46 different strains of *Synechococcus* were isolated in July 2004 (Haverkamp et al. 2009).
Direct effects on physiology have been implied from laboratory studies. One isolate, a phycoerythrin
rich strain of *Synechococcus* WH7803 (Traving et al., 2014) elicited a negative physiological effect on
the growth rate from increased $CO_2$. This was most likely a consequence of higher sensitivity to the
lower pH (Traving et al., 2014), and the cellular cost of maintaining pH homeostasis or conversely a
direct effect on protein export. Additionally, Lu et al. (2006) reported increased growth rates in a
cultured phycocyanin rich but not a phycoerythrin rich strain of *Synechococcus*, suggesting that
pigments may play some part in defining the direct physiological response within *Synechococcus*. In
addition, within natural communities (Paulino et al., 2008; Hopkins et al., 2010; Schulz et al., 2017)
variability can also arise from indirect effects such a altering competition with other picoplankton
(Paulino et al., 2008). The delay and dampened effect of $fCO_2$ on SYN abundances within our study
was more likely due to indirect effects arising from alterations in food web dynamics than to direct
impacts on the physiology of this species. Specifically, significant differences in grazing rates of SYN
between M1 and M3 (days 10 and 17, no significant lysis detected) could be responsible for the
differing dynamics between the mesocosms at the end of Phase I and beginning of Phase II.
The gross growth rates of Pico-I were significantly higher ($p < 0.05$) at high $f\mathrm{CO_2}$ compared to the low
$\mathrm{CO_2}$ concentrations during the first 10 days of Phase I. Moreover, no differences were detected in the
measured loss rates, demonstrating that increases in Pico-I were the due to increases in growth
alone. The stimulation of Pico-I by elevated $f\mathrm{CO_2}$ may be due to a stronger reliance on diffusive $\mathrm{CO_2}$
entry compared to larger cells. Model simulations reveal that whilst near-cell $\mathrm{CO_2}$/pH conditions are
close to those of the bulk water for cells <5 μm in diameter, they diverge as cell diameters increase
(Flynn et al., 2012). This is due to the size-dependent thickness of the diffusive boundary layer, which
determines the diffusional transport across the boundary layer and to the cell surface (Wolf-Gladrow
and Riebesell, 1997; Flynn et al., 2012). It is suggested that larger cells may be more able to cope
with $f\mathrm{CO_2}$ variability as their carbon acquisition is more geared towards handling low $\mathrm{CO_2}$
concentrations in their diffusive boundary layer, e.g. by means of active carbon acquisition and
bicarbonate utilization (Wolf-Gladrow and Riebesell, 1997; Flynn et al., 2012). Moreover, as the Baltic
Sea experiences particularly large seasonal fluctuations in pH and $f\mathrm{CO_2}$ (Jansson et al., 2013) due to
the low buffering capacity of the waters, phytoplankton here are expected to have a higher degree of
physiological plasticity. Our results agree with previous mesocosm studies, which reported enhanced
abundances of picoeukaryotic phytoplankton (Brussaard et al., 2013; Davidson et al, 2016; Schulz et
al., 2017), and particularly the prasinophyte *Micromonas pusilla* at higher $f\mathrm{CO_2}$ (Engel et al., 2007;
Meakin and Wyman, 2011). Furthermore, Schaum et al. (2012) found that 16 ecotypes of
*Ostreococus tauri* (another prasinophyte similar in size to Pico-I) increased in growth rate by 1.4-1.7
fold at 1,000 compared to 400 μatm $f\mathrm{CO_2}$. All ecotypes increased their photosynthetic rates and
those with most plasticity (those most able to vary their photosynthetic rate in response to changes
in $f\mathrm{CO_2}$) were more likely to increase in frequency within the community. It is possible that Pico-I cells
are adapted to a highly variable carbonate system regime and are able to increase their
photosynthetic rate when additional $\mathrm{CO_2}$ is available. This ability would allow them to out-compete
other phytoplankton (e.g. nanoeukaryotes in this study) in an environment when nutrients are
scarce.

The net growth rates and peak abundances of Pico-II were also positively affected by $f$CO$_2$. Gross growth rates were significantly higher at high $f$CO$_2$ on only two occasions (days 10 and 20) and were accompanied by high total mortality rates. Pigment analysis suggests that both Pico-I and Pico-II are chlorophytes (Paul et al., 2015) and as such may share a common evolutionary history (Schulz et al., 2017); thus Pico-II may be stimulated by $f$CO$_2$ in a similar manner to Pico I. Chlorophytes are found in high numbers at this site throughout the year (Kuosa, 1991), suggesting the ecological relevance of Pico-I and Pico-II in this ecosystem. In addition, Pico-II bloomed exactly when Pico-I declined which may suggest potential competitive exclusion.

Pico-III showed the most distinct and immediate response to CO$_2$ addition. The significant reduction in gross growth rates observed during Phase I suggests a direct negative effect of CO$_2$ on the physiology of these cells. For this group, the lower gross growth rates were matched by lower total mortality rates with increased $f$CO$_2$. Although the mean cell size of Pico-III and Pico-II were comparable (2.9 and 2.5 μm, respectively), they showed opposing responses to $f$CO$_2$ enrichment (lower Pico-III abundances at high $f$CO$_2$). These differences may arise from taxonomic differences between the two groups. Pico-III displayed relatively high phycoerythrin orange autofluorescence, likely representing small-sized cryptophytes (Klaveness, 1989), although rod-shaped *Synechococcus* up to 2.9 μm in length (isolated from this region; Haverkamp et al., 2009) or *Synechococcus* microcolonies (often only two cells in the Baltic; Motwani and Gorokhove, 2013) cannot be excluded. In agreement with Pico-III response to CO$_2$ enrichment, Hopkins et al. (2010) reported reduced abundances of small cryptophytes under increased CO$_2$ in a mesocosm study in a Norwegian fjord near Bergen.

Lastly, the two nanoeukaryotic phytoplankton groups also displayed a negative response to $f$CO$_2$ enrichment, whereby Nano-II was the least defined, most likely due to a high taxonomic diversity in this group. Nano-I started to display lower abundances at high $f$CO$_2$ during Phase I (after day 10), which was likely the result of greater differences between gross growth and total mortality (compared to low $f$CO$_2$). Alternatively, enhanced nutrient competition due to increased abundances

of SYN and Pico-I (and later on also Pico-II) at elevated $f$CO$_2$ may also have contributed to the
dampened response of Nano-I in the high $f$CO$_2$ mesocosms. The overall decline in Nano-I, during
Phase II, and sustained low abundances during Phase III may well have been the result of grazing by
the increased mesozooplankton abundances during Phase II (Lischka et al., 2017).

**Microbial loop**
The strong association of prokaryote abundance with algal biomass, present throughout the
experiment, suggests that the effect of CO$_2$ was an indirect consequence of alterations in the
availability of phytoplankton carbon. Others have reported a tight coupling of autotrophic and
heterotrophic communities at this location, with an estimated 35 % of the total net primary
production being utilized directly by bacteria or heterotrophic flagellates (Kuosa and Kivi, 1989),
suggesting a highly efficient microbial loop in this ecosystem. In addition to phytoplankton exudation,
viral lysis may also contribute to the dissolved organic carbon pool (Wilhelm and Suttle, 1999;
Brussaard et al., 2005; Lønborg et al., 2013). We calculated that viral lysis of phytoplankton between
days 9 and 13 resulted in the release of 1.3 and 13.1 ng C ml$^{-1}$ for M1 and M3, respectively. Assuming
a bacterial growth efficiency of 30 % and cellular carbon conversion of 7 fg C cell$^{-1}$ (Hornick et al.,
2017), we estimate that the organic carbon required to support bacterial dynamics during this period
(taking into account the net growth and loss rates) was 2.9 and 11.5 ng C ml$^{-1}$ in low and high $f$CO$_2$
mesocosms M1 and M3, respectively. These results suggest that viral lysis of phytoplankton was an
important source of organic carbon for the bacterial community. Our results are consistent with
bacterial-phytoplankton coupling during this eastern Baltic Sea mesocosm study (Hornick et al.,
2017), and agree with earlier work on summer carbon flow in the northern Baltic Sea showing that
prokaryotic growth was largely supported by recycled carbon (Uitto et al., 1997). The average net
growth rates of the prokaryotes during the first period of increase in Phases 0 and I (0.2 d$^{-1}$) were
comparable to rates reported for this region (Kuosa, 1991). In order to sustain the concomitant daily
mortality (between 0.3-0.5 d$^{-1}$) measured during our study, prokaryotic gross growth rates must have
been close to one doubling a day (0.5-0.7 d$^{-1}$). During Phase I, grazing was the dominant loss factor of
the prokaryotic community although there was also evidence that viral lysis was occurring.
Bermúdez et al. (2016) reported the highest biomass of protozoans around day 15. This was
predominantly the heterotrophic choanoflagellate *Calliacantha natans,* which selectively feeds on
particles <1 µm in diameter (Marchant and Scott, 1993; Hornick et al., 2017). Indeed, an earlier study
in this area showed that heterotrophic nanoflagellates were the dominant grazers of bacteria,
responsible for ingestion of approximately 53 % of bacterial production compared to only 11 % being
grazing by ciliates (Uitto et al., 1997). During the first half of Phase II, grazing was reduced and likely
contributed to the steady increase in prokaryote abundances. Specifically, a negative relationship
between the abundances of HNA-prokaryotes and $f$CO$_2$ was detected and corresponded to reduced
bacterial production and respiration at higher $f$CO$_2$ (Hornick et al., 2017; Spilling et al., 2016).
Although CO$_2$ enrichment may not directly affect bacterial growth, co-occurring global rise in
temperature can increase enzyme activities, affecting bacterial production and respiration rates
(Piontek et al., 2009; Wohlers et al., 2009; Wohlers-Zöllner et al., 2011). Enhanced bacterial re-
mineralization of organic matter may stimulate autotrophic production by the small-sized
phytoplankton (Riebesell et al., 2009; Riebesell and Tortell, 2011; Engel et al., 2013), intensifying the
selection of small cell size.
Mean viral abundances were higher under CO$_2$ enrichment towards the end of Phase I and into Phase
II which is expected under conditions of increased phytoplankton and prokaryote biomass. The
estimated average viral burst size, obtained from this increase in total viral abundance and
concurrent decline in bacterial abundances, was about 30 which is comparable to published values
(Parada et al, 2006; Wommack and Colwell, 2000). Viral lysis rates of prokaryotes were measured
until day 25 and indicated that during days 18-25 an average 10-15 % of the total prokaryote
population was lysed per day. Moreover, the concurrent steady increase in viral abundances during
Phase III indicates that viral lysis of the prokaryotes remained important. Thus, the combined impact
of increased viral mortality together with reduced production (Hornick et al., 2017) ultimately led to
the decline in prokaryote abundance (this study). Lysogeny did not appear to be an important life
strategy of viruses during our campaign. Direct effects of higher $f$CO$_2$ on viruses are not expected, as
marine virus isolates are quite stable (both in terms of particle decay and loss of infectivity) over the
range of pH of the present study (Danovaro et al., 2011; Mojica and Brussaard, 2014). The few
studies which have inferred viral lysis rates based on changes in viral abundances show reduced
abundances of algal viruses (e.g. *Emiliania huxleyi* virus) under enhanced CO$_2$ (Larsen et al., 2008)
while mesocosm results by Brussaard et al. (2013) indicated a stronger impact of viruses on bacterial
abundance dynamics with CO$_2$ enrichment.

**5 Conclusions**
Due to the low buffering capacity of the Baltic Sea and the paucity of data regarding OA impact in
nutrient-limited waters, the results presented here are pertinent to increasing our understanding of
how projected rises in $f$CO$_2$ will affect the microbial communities in this region. Our study provides
evidence that cell size, taxonomy and sensitivity to loss can all play a role in the outcome of CO$_2$
enrichment. Physiological constraints of cell size favour nutrient uptake by small cells under
conditions of reduced nutrients and our results show that these effects can be further exacerbated
by OA. Gross growth rates along with the complementary mortality rates allowed for a more
comprehensive understanding of the phytoplankton population dynamics and thus perception of
how microbial food web dynamics can influence the response of the autotrophic and heterotrophic
components of the community. Our results further suggest that alterations in CO$_2$ concentrations are
expected to affect prokaryote communities (mainly) indirectly through alterations in phytoplankton
biomass, productivity and viral lysis. Overall, the combination of growth and losses (grazing and viral
lysis) could explain microbial population dynamics observed in this study. It is noteworthy to
mention, a recent study in the oligotrophic northeast Atlantic Ocean reported a shift from grazing-
dominated to viral lysis-dominated phytoplankton community with strengthening of vertical
stratification (shoaling the mixed layer depth and enhancing nutrient limitation) (Mojica et al., 2016).
Thus, we highly recommend that future research on OA combine mesocosm studies focusing on
changes in microbial community composition and activity with experiments aimed at understanding
the effects of OA on food web dynamics, i.e. partitioning mortality between grazing and viral lysis
(Brussaard et al., 2008).

**Author Contribution**
Design and overall coordination of research by CB. Organization and performance of analyses in the
field by KC. Data analysis by KC, CB, and SA-F. Design and coordination of the overall KOSMOS
mesocosm project by UR. All authors contributed to the writing of the paper (KC, KM and CB are lead
authors).

**Acknowledgements**
This project was funded through grants to C.B. by the Darwin project, the Netherlands Institute for
Sea Research (NIOZ), and the EU project MESOAQUA (grant agreement number 228224). We thank
the KOSMOS project organisers and team, in particular Andrea Ludwig, the staff of the Tvärminne
Zoological Station and the diving team. We give special thanks to Anna Noordeloos, Kirsten Kooijman
and Richard Doggen for their technical assistance during this campaign. We also gratefully
acknowledge the captain and crew of R/V ALKOR for their work transporting, deploying and
recovering the mesocosms. The collaborative mesocosm campaign was funded by BMBF projects
BIOACID II (FKZ 03F06550) and SOPRAN Phase II (FKZ 03F0611).

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

**Table 1.** $f$CO$_2$ concentrations (µatm) averaged over the duration of the experiment
(following CO$_2$ addition) and subsequent classification as low, intermediate or high.
Mesocosms sampled for mortality assays are denoted by an asterisk. The symbols and
colours are used throughout this manuscript and corresponding articles in this issue.

| Mesocosm | M1* | M5 | M7 | M6 | M3* | M8 |
|---|---|---|---|---|---|---|
| CO$_2$ Level | LOW | LOW | INTERMEDIATE | INTERMEDIATE | HIGH | HIGH |
| Mean $f$CO$_2$ (µatm) days 1-43 | 365 | 368 | 497 | 821 | 1007 | 1231 |
| Symbol |  |  |  |  |  |  |

**Table 2.** The fit ($R^2$) and significance ($p$-value) of linear regressions applied to assess the relationship
between net growth rate and temporally averaged $f$CO$_2$ for the different microbial groups
distinguished by flow cytometry. The results presented are for two periods distinguished from NMDS
analysis; NMDS-based period 1 (days 3-13) and 2 (days 16-24). A significance level of p≤0.05 was
taken and significant results are shown in bold.

| Phytoplankton Group | NMDS period 1 (days 3-13) | | NMDS period 2 (days 16-24) | |
|---|---|---|---|---|
| | *p* | *R²* | *p* | *R²* |
| SYN | 0.10 | 0.53 | **0.05** | **0.63** |
| Pico-I | **0.01** | **0.80** | **0.05** | **0.64** |
| Pico-II | 0.52 | 0.11 | 0.10 | 0.52 |
| Pico-III | **0.04** | **0.67** | **<0.01** | **0.91** |
| Nano-I | **0.01** | **0.79** | 0.26 | 0.30 |
| Nano-II | 0.20 | 0.36 | 0.06 | 0.61 |
| HNA | **0.05** | **0.64** | 0.89 | 0.00 |
| LNA | **<0.01** | **0.95** | **0.02** | **0.76** |

**Table 3.** The fit ($R^2$) and significance (*p*-value) of linear regressions used to relate peak abundances
and net growth rate with temporally averaged $f$CO$_2$ for the different microbial groups distinguished
by flow cytometry during specific periods of interest. A significance level of p≤0.05 was taken and
significant results are shown in bold.

| | Peak abundance | | Net growth rate | |
|---|---|---|---|---|
| | *p* | R2 | *p* | R2 |
| SYN day 24 | 0.01 | 0.80 | | |
| Pico-I day 5 | 0.01 | 0.81 | | |
| Pico-I day 13 | <0.01 | 0.94 | | |
| Pico-I day 21 | 0.01 | 0.84 | | |
| Pico-II day 17 | <0.01 | 0.93 | | |
| Pico-III day 24 | <0.01 | 0.91 | | |
| Nano-I day 17 | 0.04 | 0.67 | | |
| Pico-I days 1-5 | | | <0.01 | 0.90 |
| Pico-I days 5-9 | | | <0.01 | 0.89 |
| Pico-II days 12-17 | | | 0.01 | 0.82 |

**Figure captions**
**Fig. 1. (a)** Time-series plot of depth-integrated (0.3–10 m) total phytoplankton abundance (< 20 µm)
and **(b)** total eukaryotic phytoplankton abundance for each mesoscosm and the surrounding waters
(Baltic). Dotted lines indicate the end of Phase I and end of Phase II. Colours and symbols represent
the different mesocosms and are consistent throughout the manuscript. Mean $f$CO$_2$ during the
experiment  (days 1-43) were: M1, 365  µatm; M3, 1007  µatm; M5, 368  µatm; M6, 821  µatm; M7,
497 µatm; M8, 1231 µatm.
**Fig. 2.** Non-metric multidimensional scaling (NMDS) ordination plot of microbial community
development in each mesocosm and surrounding waters (Baltic) over the experimental period.
Phases are indicated by different open symbols.  Days of experiment (DoE) when communities
separate (3,13,16 and 24) are indicated by different closed symbols. Phytoplankton groups are
denoted as: SYN (Syn), Pico-I (P-I), Pico-II (P-II), Pico-III (P-III), Nano-I (N-I), Nano-II (N-II), Low NA
prokaryotes (LNA) and High NA prokaryotes (HNA).
**Fig. 3.** Time-series plot of depth-integrated (0.3–10 m) abundances of **(a)** *Synechococcus* (SYN) **(b)**
picoeukaryotes I (Pico-I) **(C)** picoeukaryotes II (Pico-II) **(d)** picoeukaryotes III (Pico-III) **(e)**
nanoeukaryotes  I (Nano-I) and  **(f)** nanoeukaryotes II (Nano-II) distinguished by flow cytometric
analysis of the microbial community in each mesocosm and the surrounding waters (Baltic).  Dotted
lines indicate the end of Phase I and end of Phase II, grey areas indicate NMDS-based periods 1 and 2
where net growth rates were analysed.
**Fig. 4.** Total mortality rates (i.e., grazing and lysis, solid bars) and gross growth rates (striped bars) d$^{-1}$
of the different phytoplankton groups in mesocosms M1 (blue) and M3 (red) on the day indicated:
**(a)** *Synechococcus* (SYN) **(b)** picoeukaryotes I (Pico-I) **(C)** picoeukaryotes II (Pico-II) **(d)** picoeukaryotes
III (Pico-III) **(e)** nanoeukaryotes  I (Nano-I) and  **(f)** nanoeukaryotes II (Nano-II).  Significant (p ≤0.05)
differences between mesocosms are indicated by an asterisk above the relevant bar (either total loss
or gross growth).  A colored zero indicates that a rate of zero was measured in the mesocosm of the
corresponding colour and the absence of a bar, or zero indicates a failed experiment. Dotted lines
indicate the end of Phase I and end of Phase II.
**Fig. 5.** Time-series plot of the mean phytoplankton carbon biomass in high $f$CO$_2$ (M3,6,8; red) and low
$f$CO$_2$ (M1,5,7; blue) mesocosms of **(a)** Pico I and II combined and **(b)** SYN, Pico III, Nano I and II
combined. Error bars represent one standard deviation from the mean. Carbon biomass calculated
assuming a spherical diameter equivalent to the mean average cell diameters for each group and
conversion factors of 237 fg C $\mu m^{-3}$ (Worden et al.2004) and 196.5 fg C $\mu m^{-3}$ (Garrison et al. 2000) for
pico- and nano-sized plankton, respectively. Dotted lines indicate the end of Phase I and end of Phase
II.
**Fig. 6.** Time series plot of depth-integrated (0.3–10 m) abundances of **(a)** Total prokaryotes **(b)** High
fluorescent nucleic acid prokaryote population (HNA) **(c)** Low fluorescent nucleic acid prokaryote
population (LNA) and **(d)** Total virus. Dotted lines indicate the end of Phase I and end of Phase II, grey
areas indicate NMDS-based periods where net growth rates were analysed.
**Fig. 7.** Prokaryote mortality rates: **(a)** Total grazing ($d^{-1}$) and **(b)** viral lysis rates as % of prokaryote
standing stock, in mesocosms M1 (low $f$CO$_2$, blue) and M3 (high $f$CO$_2$, red). Grazing rates were
determined from fluorescently labelled prey, viral lysis rates from viral production assays. Error bars
represent one standard deviation of triplicate assays. Significant ($p \leq 0.05$) differences between
mesocosms are indicated by an asterisk. Dotted lines indicate the end of Phase I.
**Fig. 8.** Correlation between total carbon biomass ($\mu mol\ L^{-1}$) and total prokaryote abundance in low
$f$CO$_2$ mesocosms (M1, 5 and 7; blue) and high $f$CO$_2$ mesocosms (M3,6, 8; red) throughout the
experiment (days -2 to 39).

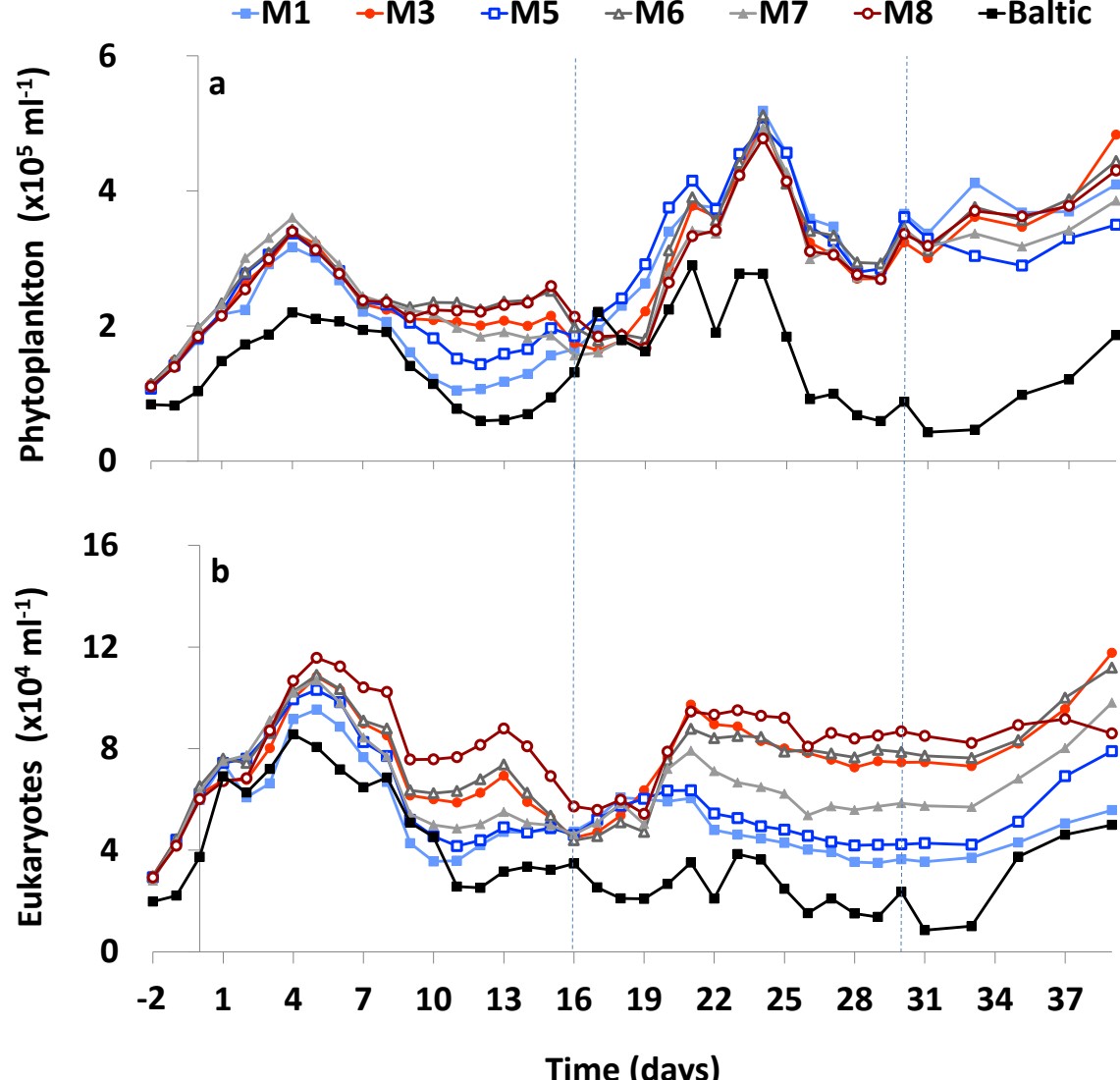

**Figure 1.**

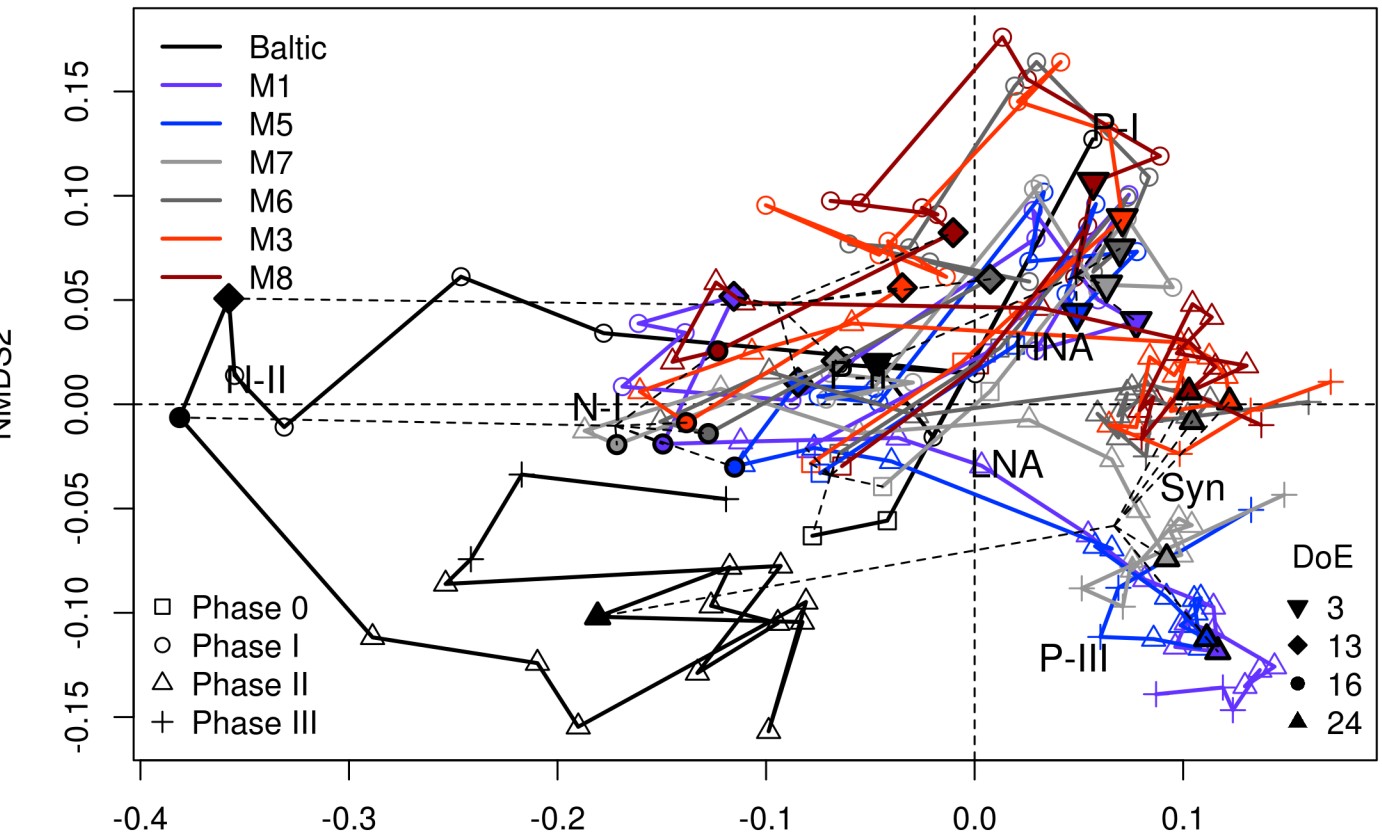

Figure 2.

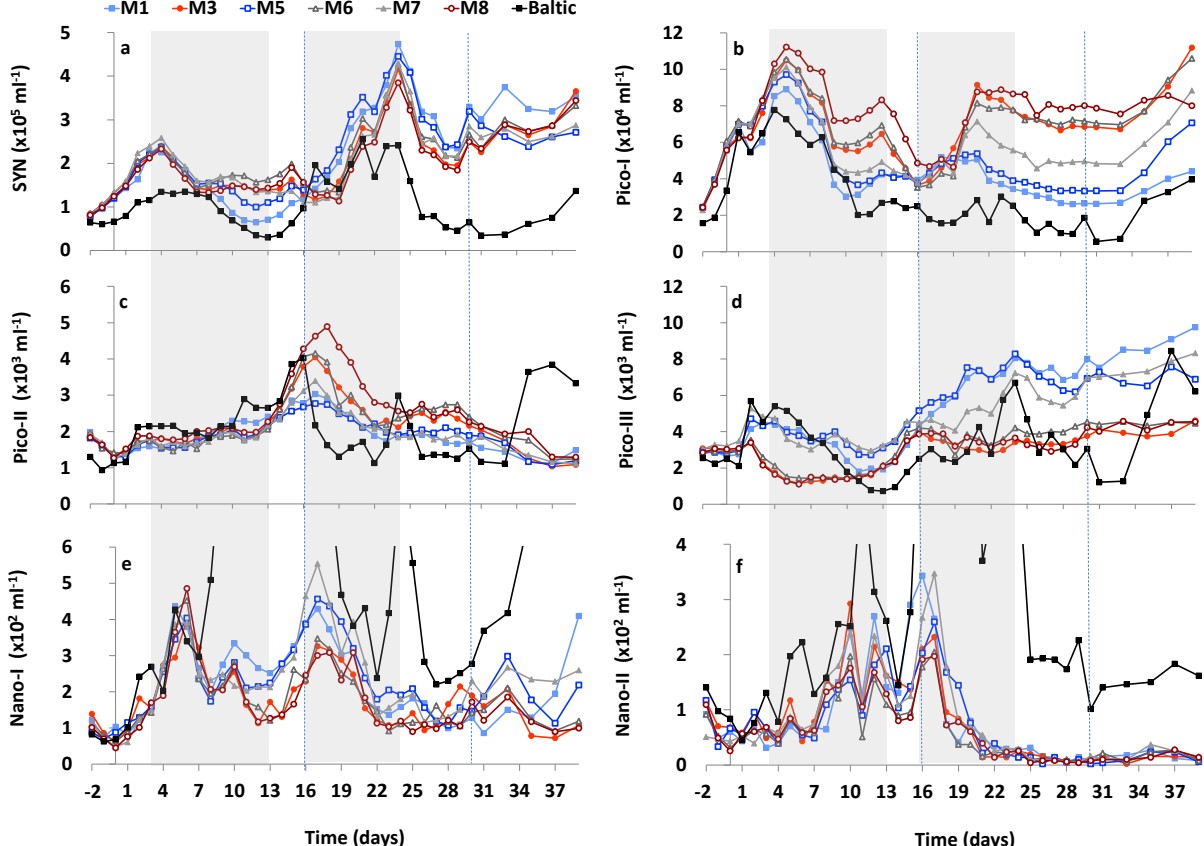

**Figure 3.**

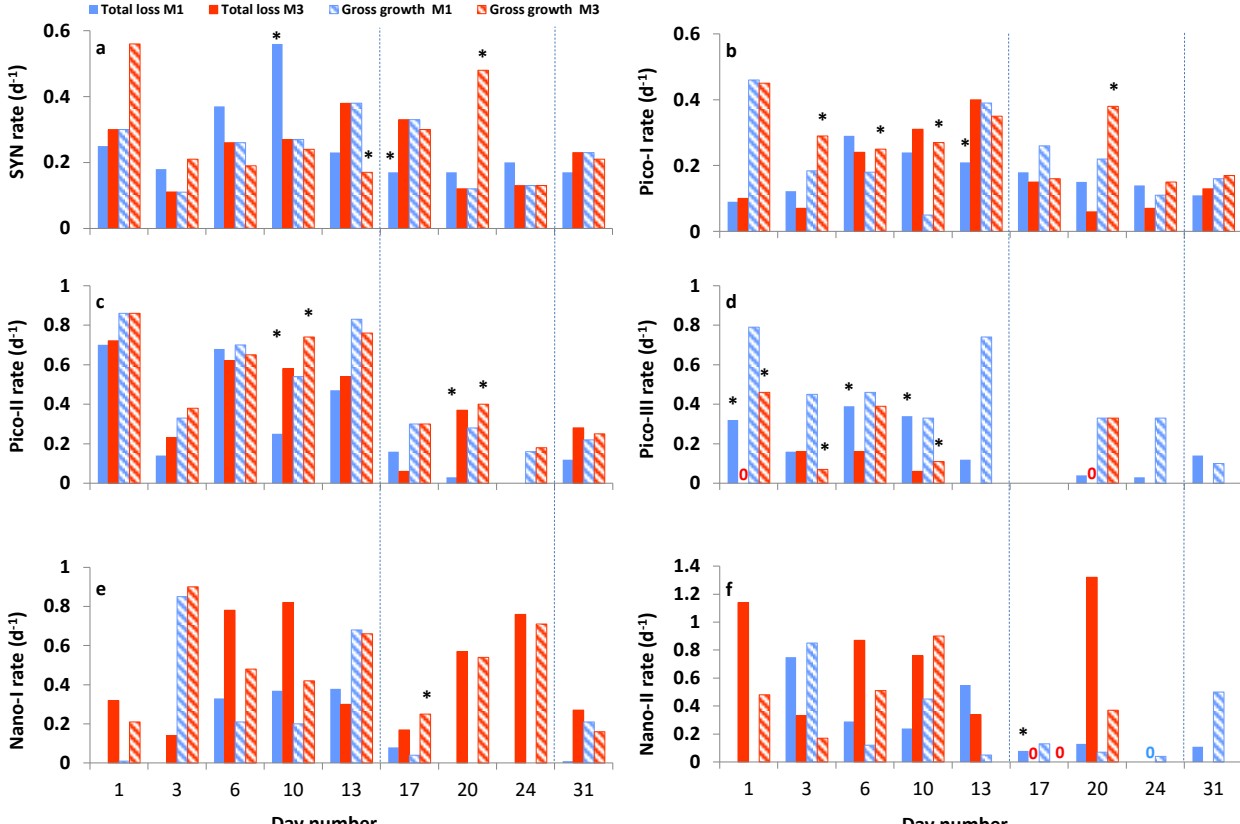

**Figure 4.**

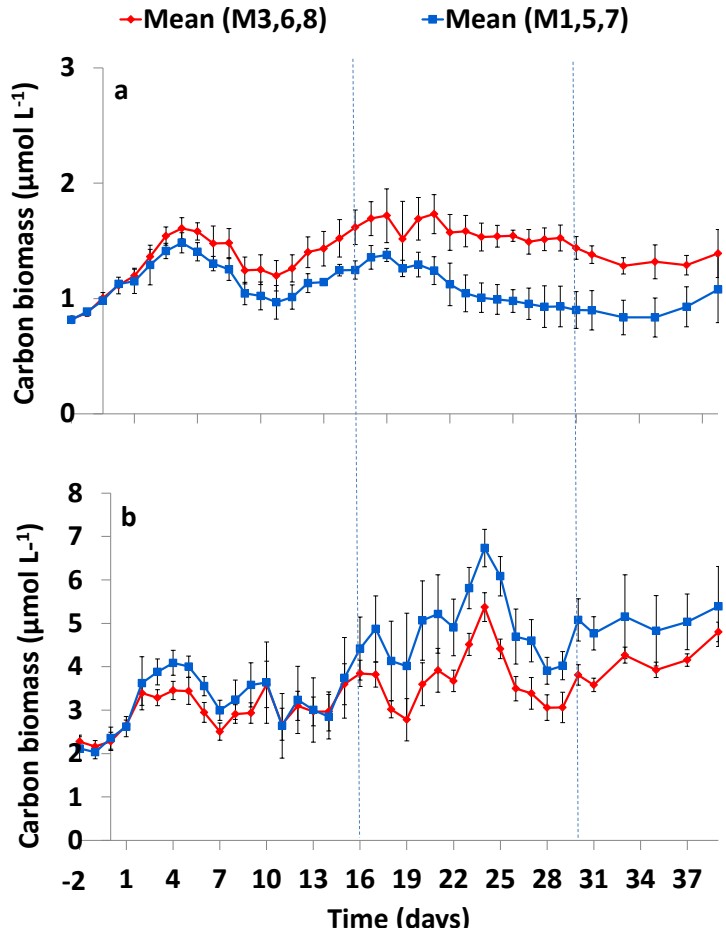

**Figure 5.**

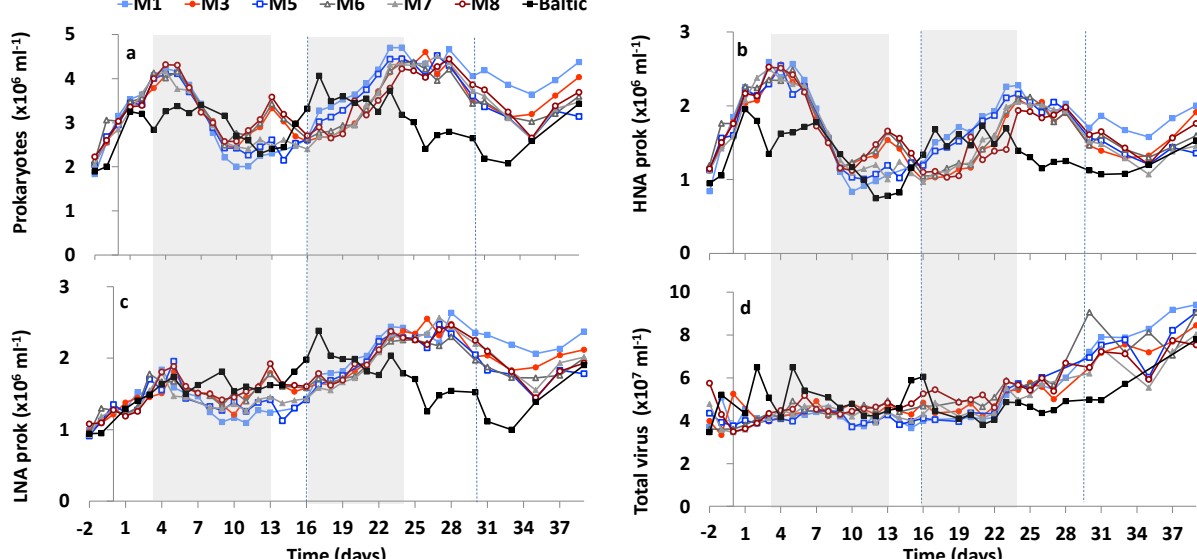

**Figure 6.**

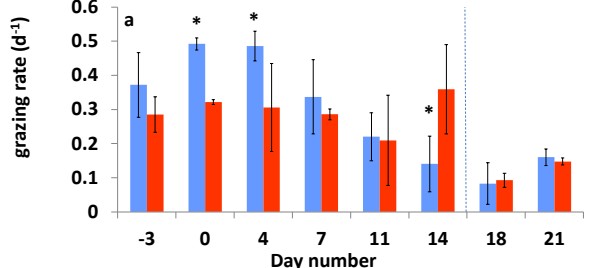 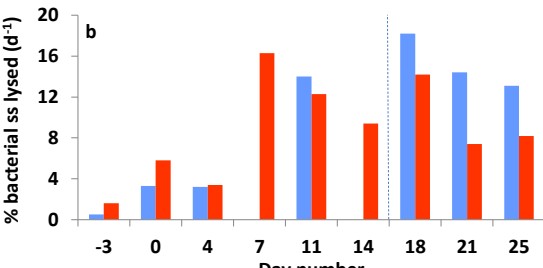

**Figure 7.**

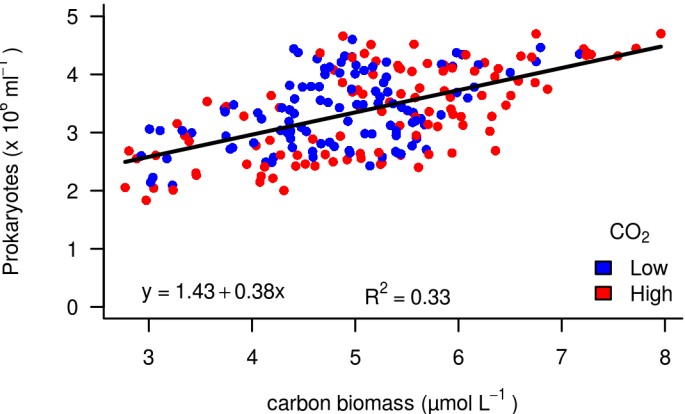

Figure 8.