# Peer review of "Alterations in microbial community composition with increasing $f$CO₂: a mesocosm study in the eastern Baltic Sea"

_Biogeosciences, 2015_

## Referee Comment (RC1) · Anonymous Referee #1 · 3 Apr 2016

Crawfurd et al. present a study on the effect of increased CO2 on different planktonic members including different photoautotrophs and the total bacterial community. One of the main differences of this study compared to other ocean acidification studies seems that there were no nutrients added in this study to avoid an overstimulation of primary production. The dataset is comprehensive and particularly looks at changes in grazing and lysis under the different CO2 treatments. The data presented are an important contribution to understanding the dynamics in microbial communities under increasing atmospheric CO2 concentrations. My main criticisms here are that when I look at the figures, to me it seems as if overall shifts in the microbial community structure (or more precisely changes in the abundances of selected plankton members) do actually change in all the mesocosms but that changes are more pronounced in the 'high CO2' treatments. This means that 1) the title is misleading, and 2) parts of the interpreta-

tion and discussion of the data are misleading. However, this fact is not discussed at all in this manuscript. Unfortunately, the authors omit any discussion of factors other than the two that they investigated. For example, what about the changing temperature during the experiments; it ranges from 8-15 deg C, but this isn't discussed anywhere. They also omit any discussion about any differences between the mesocosms and the surrounding waters and what the differences could mean (as far as I can see from the Suppl. figures, there are differences). Further, I find it stunning that the total phytoplankton doesn't vary that much over the different mesocosm treatments (Fig. 1). The authors do not acknowledge or discuss this anywhere. The authors mention that no nutrients were added to the mesocosms to "resemble the natural bottom-up environmental conditions." Although I can understand why the authors did not add nutrients, however, I doubt that this resembles the natural environmental conditions over a period of six weeks. By enclosing the water masses, the 'natural' nutrient supply, which is either horizontally or vertically, is cut off. But this discussion might also have to be carried out across the different companion manuscripts on these mesocosm experiments submitted to the special issue in BG. But no matter how this discussion turns out, the authors should discuss it in this manuscript. Maybe it is reason, for example, why the start and end abundances of the experiments are sometime quite similar while changes happened in between. The discussion section could benefit from a bit more 'discussion' rather than just the listing of other articles. How do the results you present actually fit into the literature and what does it mean for your data when other studies have shown certain effects (also see comments below). Further comments: In general, several sentences and paragraphs are quite lengthy and not easy to understand (all the way to not understandable at all). The introduction is lengthy and repetitive in some places and could be condensed. Throughout the manuscript, the numbers of the mesocosms are used, e.g. M1 or M3. This is very confusing especially in the discussion. Could be exchanged for LOW1 or HIGH2 or something that designates a treatment to that number, especially since M1 and M5 seem to be replicates, as well as M6 and M7, and M3 and M8.

- what are the 'failed' experiments, are they 'samples lost'? or outliers?

- p2, l3: salinity in the Baltic Sea ranges from near-freshwater to near-full seawater, I wouldn't necessarily call it extremely low salinity implying a negative effect, especially since it varies a lot throughout the Baltic Sea

- p2, l6: "We examined the effects of ocean acidification in the microbial community during..." Do you mean on the community structure or on the carbon export or on primary production rates? Please specify in the abstract.

- p2, l25: the threats don't face the marine ecosystems but the marine ecosystems face the threats

- p3, l2-9: This paragraph doesn't fit here and disrupts the flow. I would place it to where you describe your experiments.

- p3, l10-12: reads awkwardly, split into two sentences

- p3 l26- p4, l3: this is repetitive

- p4, l20: which key knowledge, it's not clear from this sentence

- p4, l24: delete the 'top-down control'

- p5 and following: the experimental set-up could be made much clearer, maybe use a sketch for this

- p5, l23: nitrate, phosphate, silicate and ammonium are per definition (dissolved) inorganic nutrients

- p7, l1-2: Pico III and Pico I do not have comparable cell sizes; one is about 1 micrometer and the one is about 2.9 micrometer in diameter, maybe you meant Pico II and III?

- p7, l4: was this conversion factor used for all organisms or just the Synechococcus? There are studies that clearly show that the carbon density changes with cell

volume with the density being lower at higher volumes (see Verity et al. 1992 L&O or Menden-Deuer and Lessard 2000 L&O) If the same conversion factor was used for all organisms, this would likely bias the results significantly

- p7, l11: why not use the term total prokaryotes, because this is what it actually is, and not the heterotrophic prokaryotes which clearly should not include Synechococcus or other photoautotrophic organisms (the 10% argument is not correct here in my opinion)

- p7, l15: final concentrations of what, molar? micromolar? micrograms per kg?

- p11, l19: the $R^2$ is 0.49 in the figure and the regression line doesn't seem like it would be 0.98 either

- p11, l22-25: this part is hard to read. Please rephrase.

- p12, l3: wasn't the start of the mesocosm day -5 or day 0 rather than day 13?

- p12, l5: the decline following what?

- p12, l20: there is no such thing as net abundance; you can have net rates but not net abundances (also check throughout the manuscript)

- p12/13: "This may have stimulated the gross growth in M3 as compared to M1 (day 19; Fig. 3b) for a longer period in the high $fCO_2$ mesocosms, this accompanied by higher losses at low $CO_2$ resulted in a positive correlation of net growth rates with $fCO_2$ (Fig. 3e, $R^2 = 0.71$) and almost 2-fold higher net abundances at day 21 (Fig. 3a) correlating with $fCO_2$ (Fig. 3h, $R^2 = 0.84$)." I have honestly no idea what this sentence means. It is unnecessarily long and confusing. Further, the 'net abundances', please see comment above, and maybe either $CO_2$ or $fCO_2$ treatments could be used consistently throughout the manuscript

- p13, l13-17: unnecessarily long and confusing sentence

- p14, l25: what are "$CO_2$ days"?

- p18, l17-19: How do the authors infer a bacterial production rate of about 0.6 dˆ-1 when grazing is about 0.3-0.5 dˆ-1? Is that due to a net positive growth in bacterial abundance? If so, it would be good to mention here. Otherwise the reader might assume steady state as I did here.

- p18, l27: "Also Pico II showed positively correlated net growth rates with CO2 enrichment, but somewhat later into phase I (days 12-17) due to reduced losses." Awkward start of the sentence, maybe: "Net growth rates of Pico II correlated positively with CO2 enrichment. . . . ."

- p20, l25-28: these are mainly results and then one other article is mentioned; but what does this now mean for your data? Do you think that TEP production was a factor regulating the abundances in your study? the actual discussion is missing

- p21, l9-11: This comes out of the blue. How did you examine this?

- p22, l2: ". . . has a very different physiology,. . ." different from what?

- p22, l17: DOC could have come also from sloppy feeding?

- p23, l17: do you mean remineralization of organic matter rather than nutrients?

- p23, l22: "multiple other factors"? Please name them here.

- Fig. 2: Instead of calling it the 'total prokaryotic phytoplankton', just call it what it is, the Synechococcus population

- Fig. 2: How is the p<0.1 indicated? Is it possibly also the category 'p>0.05'?

- I don't see any 'f' in this figure (and some of the following figures).

- What are the black dots here and in other plots (and I don't mean the single asterisks)?

- Fig. 2: panel b here and in following figures: I understand why the authors want to present the data together, but the plots are really obscured this way and it makes it hard for the reader to discern any data from them. I would suggest to split them into

two panels

- Fig. 2: Here and following figures, what do you mean by "otherwise no data is a zero"?

- Fig. 6: The legend says that linear regression statistics are provided in the plot, however, I couldn't see a p value.

- Fig. 10: the grazing rate and lysis rate are both loss rates; nevertheless, one of them is presented on a negative scale while the other is presented on a positive scale. I find it confusing.

- Suppl. Table S1: What are the units here?

- Suppl. Table S2: What are the units here?

- Fig. S1 and S2 legends: the upper layer is mentioned here but the measurements are from the total water mass, i.e. 0.3-17 m rather than 0.3-10 m

- Fig. S2: panel f is missing

- Fig. S3 and S4 please use the proper symbol for micromol and not umol

---

## Referee Comment (RC2) · Anonymous Referee #2 · 19 May 2016

The authors present mesocomos experiment in which they test the effect of ocean acidification on microbial community, by increasing CO2 levels. Since the ongoing climate change this research is of high importance and data presented in this msc are very valuable. The authors focused on the phytoplankton size fraction ËĆ20 $\mu$m, as well as to heterotrophic procaryotes and viruses. The experiment set u is well explained and the msc is in general well written. I do recommend the msc to be published after major revision Key points that I would recommend to be answered: 1. All dough, the target organisms are of key importance for ecosystem functioning, I do not agree that the results are relevant if not being in correlation to the whole phytoplankton community, at least presented as Chl a concentration. If those data exist I highly recommend including them in the msc. 2. In Material and Methods it is stated that the samples of surrounding water were taken, but there are not presented in results or at least discussed. It is essential to discuss those data. I would recommend including those data into the graphs reporting about the microbial community changes. Also, I am not sure if I have understood it correctly – but it seems that $CO_2$ was added to all mesocosms? In that way you do not have control and any change in microbial community could have been because of some other factor? What about the temperature? What is the usual phytoplankton development dynamics in the Baltic Sea? Also, I do not see any shifts in microbial community, but changes in specific group abundances during the experiment. 3. Since every experiment need to be repetitive and results comparable with other study site and experiment set up it would be necessary to know the community structure of microbial community. I am aware of the difficulty of taxonomical recognition of size fraction below $20\mu$m, but it is essential. Flow cytometry in an excellent tool, but in an experiment set up of this range I do not believe it is enough. I did like the way how the investigated groups were divided, but the next step in research should be the taxonomical identification. Not all organisms in the same size fraction have same physiological response to environment drivers. Also, the authors in the discussion, do not discuss their data, but cite different authors and their research. If you did not go to the species level – those data cannot be discussed. 4. The analyzed groups were good explained except the cyanobacteria. The authors distinguish Synechococcus, but no Prochlorococcus. Since the oligotrophy Prochlorococcus could develop in the environment and it can be separated by flow cytometry. Then stated that the prokaryotes include bacteria, archea and unicellular cyanobacteria (together marked as HP) – what cyanobacteria could it be? 5. The tittle also does not represent the results – there is not shift in microbial community presented. It is not easy to follow the results and discussion with given abbreviations (M1, M5...) for mesocosm experiments. It is not clear enough. Maybe a table would be a good way to explain the abbreviations. The discussion needs to be rewritten. The results would need to be discussed in more detail. The cited literature and results are maybe not the best choice for the results presented.

---

## Author Response (AR1)

Interactive comment (in bold, italic) on "Shifts in the microbial community in the Baltic Sea with increasing CO2" by K. J. Crawfurd et al.

*We thank the reviewers for taking the time to review this manuscript and for the pertinent and constructive comments they have raised. Wherever possible we have incorporated their suggestions and if not I hope that we have clearly explained our reasoning.*

**Anonymous Referee #1**
My main criticisms here are that when I look at the figures, to me it seems as if overall shifts in the microbial community structure (or more precisely changes in the abundances of selected plankton members) do actually change in all the mesocosms but that changes are more pronounced in the 'high CO2' treatments. This means that 1. the title is misleading, and 2. parts of the interpretation and discussion of the data are misleading. However, this fact is not discussed at all in this manuscript.

1. ***We understand Reviewer's point of view and changed the title into "Shifts in the size structure of the microbial community in the Baltic Sea with increasing $CO_2$".***
2. ***We clarify in the Discussion of the revised manuscript that the extent of temporal dynamics differed for the high $fCO_2$ mesocosms (and not the dynamics itself).***

3. Unfortunately, the authors omit any discussion of factors other than the two that they investigated. For example, what about the changing temperature during the experiments; it ranges from 8-15 deg C, but this isn't discussed anywhere.
*These variables affect the overall dynamics and not specifically the differences between the mescoosms (although we cannot exclude co-stressor effects by other variables on top of $CO_2$ enrichment for the negative impact particularly – e.g. Pico III and Nano I). We make a statement in the Discussion of revised manuscript.*

4. They also omit any discussion about any differences between the mesocosms and the surrounding waters and what the differences could mean (as far as I can see from the Suppl. figures, there are differences).
*We chose not to include them in the main figures for 2 reasons. Firstly they are not directly comparable, this location is subject to water movement and during the experiment distinctly different water masses with different physical and biological signatures moved into the surrounding water. Secondly, and due to the previous reasoning, including the phytoplankton abundances in the surrounding waters makes the figures more difficult to read. Occassionally the abundances are much greater than in the mesocosms and it is then harder to discern differences between the mesocosms (see Supplementary Table S1 and Fig S1). Overall, microbial temporal dynamics are largely comparable, with a few exceptions: i.e., phytoplankton Nano I and II show much higher abundances in the outside water whilst all the picoplankton abundances are lower in the surrounding waters. We will add a description on the outside water microbial dynamics in the Results section of the revised manuscript, and add discussion on what may have caused the differences.*

5. Further, I find it stunning that the total phytoplankton doesn't vary that much over the different mesocosm treatments (Fig. 1). The authors do not acknowledge or discuss this anywhere.
***Total phytoplankton is numerically dominated by Synechococcus, making up to 74% of total (as explained in section 3.1), and Synechococcus especially did not show large variations between the $fCO_2$ treatments (mesocosms).***

6. The authors mention that no nutrients were added to the mesocosms to "resemble the natural bottom-up environmental conditions." Although I can understand why the authors did not add nutrients, however, I doubt that this resembles the natural environmental conditions over a period of six weeks. By enclosing the water masses, the 'natural' nutrient supply, which is either horizontally or vertically, is cut off. But this discussion might also have to be carried out across the different companion manuscripts on these mesocosm experiments submitted to the special issue in BG. But no matter how this discussion turns out, the authors should discuss it in this manuscript. Maybe it is reason, for example, why the start and end abundances of the experiments are sometime quite similar while changes happened in between.
***Reviewer is correctly stating that we have not discussed lateral and vertical transport of nutrients, but the summer situation is largely driven by regenerative nutrient supply (Kuosa, 1991). The summer situation is one with vertical stratification and low nutrient concentrations resulting in small-sized phytoplankton dominance that are typically well-controlled by grazing and viral lysis (Kuosa, 1991 and demonstrated by our results). We will specify this more clearly in the Discussion of the revised manuscript. We also measured nutrients outside the mesocosms and found that nitrate, the limiting nutrient, was at similar concentrations inside and outside the mesocosms. Phosphate did increase outside the mesocosms but only after day 25. Silicate was higher and more variable outside the mesocosms.***

7. The discussion section could benefit from a bit more 'discussion' rather than just the listing of other articles. How do the results you present actually fit into the literature and what does it mean for your data when other studies have shown certain effects (also see comments below).
***We will rework the Discussion accordingly Reviewer's comment.***

8. Several sentences and paragraphs are quite lengthy and not easy to understand
(all the way to not understandable at all).
***We have carefully checked the manuscript and shortened / clarified where we thought necessary (as Reviewer is not specifically mentioning section), and we also ask another native English speaking colleague to read the manuscript for clarity.***

9. The introduction is lengthy and repetitive in some places and could be condensed.
***We do not agree with Reviewer that the Introduction is lengthy, but we realize it is repetitive at times. We removed redundancies while securing readability.***

10. Throughout the manuscript, the numbers of the mesocosms are used, e.g. M1 or M3. This is very confusing especially in the discussion. Could be exchanged for LOW1 or HIGH2 or something that designates a treatment to that number, especially since M1 and M5 seem to be replicates, as well as M6 and M7, and M3 and M8.

*The notation used is consistent across all manuscripts in this special issue and the mesocosms with their mean $fCO_2$ are presented in Fig. 1. However, we see Reviewer's point and will include a Table, as well as specify better at the start of the Results section as well as in the Discussion which mesocosms are LOW and HIGH.*

*Table 1. $fCO_2$ concentrations (µatm) as an average for the duration of the experiment following $CO_2$ addition and specification of this $CO_2$ level as low, medium or high. \*denotes mesocosms sampled for grazing and viral lysis assays*

| Mesocosm | M1* | M5 | M7 | M6 | M3* | M8 |
|---|---|---|---|---|---|---|
| $CO_2$ Level | LOW | LOW | INTERMEDIATE | INTERMEDIATE | HIGH | HIGH |
| Mean $fCO_2$ (µatm) days 1-43 | 365 | 368 | 497 | 821 | 1007 | 1231 |

11.  what are the 'failed' experiments, are they 'samples lost'? or outliers?

*Failed experiments include very low cell abundance samples, complicating proper analysis (and consequently results) of the diluted series, as well as results displaying a positive slope rather than a negative slope for apparent growth rates versus fraction natural water (thus  where the dilution does not result in a reduction in mortality). An explanation is now given in the text (M&M section 2.3).  We also make reference to paper by Kimmance & Brussaard 2010 describing such issues, as well as Stoecker et al. 2015 which suggest potential causes for positive regressions.*

12. - p2, l3: salinity in the Baltic Sea ranges from near-freshwater to near-full seawater, I wouldn't necessarily call it extremely low salinity implying a negative effect, especially since it varies a lot throughout the Baltic Sea.

*Reviewer is correct. However, during our study salinity was around 5.7 only. We have deleted this sentence and provided the information about the sampling location in the next sentence (stating also there the actual salinity).*

13. - p2, l6: "We examined the effects of ocean acidification in the microbial community during. . ." Do you mean on the community structure or on the carbon export or on primary production rates? Please specify in the abstract.

*We specified this in the Abstract of the revised manuscript, making clear we examined effects on microbial community structure.*

14. - p2, l25: the threats don't face the marine ecosystems but the marine ecosystems face the threats
*Thank you, we corrected this.*

15. - p3, l2-9: This paragraph doesn't fit here and disrupts the flow. I would place it to where you describe your experiments.
*We have amended the Introduction and moved this information with additional clarification to M&M section, and to paragraph concerning general effect of lower salinity on pH buffer capacity in Discussion.*

16. - p3, l10-12: reads awkwardly, split into two sentences
*We split into 2 sentences.*

17. - p3 l26- p4, l3: this is repetitive
*The Introduction is revised and we modified repetitive sections.*

18. - p4, l20: which key knowledge, it's not clear from this sentence
*Altered.*

19. - p4, l24: delete the 'top-down control'
*We rephrased this sentence.*

20. - p5 and following: the experimental set-up could be made much clearer, maybe use a sketch for this
*We follow the general overview paper in this same special issue and make reference to this paper (by Paul et al. 2015), describing the experimental set-up very well (also with figures). We rephrased the sentence making reference to Paul et al. paper in order to clarify this better.*

21. - p5, l23: nitrate, phosphate, silicate and ammonium are per definition (dissolved) inorganic nutrients
*Agreed and we altered accordingly.*

22. - p7, l1-2: Pico III and Pico I do not have comparable cell sizes; one is about 1 micrometer and the one is about 2.9 micrometer in diameter, maybe you meant Pico II and III?
*Yes, we thank Reviewer for noting this and have corrected accordingly.*

23. - p7, l4: was this conversion factor used for all organisms or just the Synechococcus? There are studies that clearly show that the carbon density changes with cell volume with the density being lower at higher volumes (see Verity et al. 1992 L&O or Menden-Deuer and Lessard 2000 L&O) If the same conversion factor was used for all organisms, this would likely bias the results significantly
*We have recalculated applying conversion factors of 237 fg C $\mu m^{-3}$ (Worden et al.2004) and 196.5 fg C $\mu m^{-3}$ for pico- and nano-sized phytoplankton (Garrison et al. 2000),*

*respectively according to Mojica et al. 2015 and will use the revised Figures in the manuscript. However the overall dynamics are the same.*

[Figure]

[Figure]

*Fig.8. POC calculated from mean cell abundances assuming cells to be spherical and applying conversion factors of 237 fg C $\mu m^{-3}$ (Worden et al.2004) and 196.5 fg C $\mu m^{-3}$ for pico- and nano-sized plankton (Garrison et al. 2000), respectively according to Mojica et al. (2015). Error bars show one standard deviation.   a) Temporal dynamics of Pico I and II b) Temporal dynamics of POC for all other eukaryotes ie. Pico III, Nano I and II.*

24. - p7, l11: why not use the term total prokaryotes, because this is what it actually is, and not the heterotrophic prokaryotes which clearly should not include Synechococcus or other photoautotrophic organisms (the 10% argument is not correct here in my opinion).
***Synechoccus makes up for around 10% of the total prokaryotes in our study, but we understand Reviewer's comment and changed 'heterotrophic prokaryotes' into 'prokaryotes' (and thus also HP into prokaryotes).***

25. - p7, l15: final concentrations of what, molar? micromolar? micrograms per kg?
***It is a final concentration of the commercial stock, which does not have a specified unit. This is the common way of expressing these final concentrations, but we moved 'commercial stock' directly following 'final concentration of' to improve understanding.***

26. - p11, l19: the R^2 is 0.49 in the figure and the regression line doesn't seem like it would be 0.98 either
***Actually it is 0.98 and the figure was incorrect; we appologize for the confusion and thank the Reviewer for noting. We have replaced it with the correct one.***

[Figure]

27. - p11, l22-25: this part is hard to read. Please rephrase.
***We rephrased and split into separate sentences.***

28. - p12, l3: wasn't the start of the mesocosm day -5 or day 0 rather than day 13?
***We altered the text to make it clearer that we referred to the bloom period.***

29. - p12, l5: the decline following what?
***Should be the 'following decline', which we now corrected.***

30. - p12, l20: there is no such thing as net abundance; you can have net rates but not net abundances (also check throughout the manuscript)
*Reviewer is correct and we deleted 'net'.*

31. - p12/13: "This may have stimulated the gross growth in M3 as compared to M1 (day 19; Fig. 3b) for a longer period in the high $fCO_2$ mesocosms, this accompanied by higher losses at low $CO_2$ resulted in a positive correlation of net growth rates with $fCO_2$ (Fig. 3e, $R^2 = 0.71$) and almost 2-fold higher net abundances at day 21 (Fig. 3a) correlating with $fCO_2$ (Fig. 3h, $R^2 = 0.84$)." I have honestly no idea what this sentence means. It is unnecessarily long and confusing. Further, the 'net abundances',please see comment above, and
*We reduced the length of the sentence by splitting it into two and corrected according to Reviewer's comment.*

maybe either $CO_2$ or $fCO_2$ treatments could be used consistently throughout the manuscript
*Noted and we have altered this to $fCO_2$ throughout (in agreement with the general overview paper by Paul et al. (2015).*

32. - p13, l13-17: unnecessarily long and confusing sentence
*We understand Reviewer's concern and have split the sentence into two sentences to improve readability.*

33. - p14, l25: what are "$CO_2$ days"?
*Sentence was indeed confusing and we have clarified it now.*

34. - p18, l17-19: How do the authors infer a bacterial production rate of about 0.6 $d^{-1}$ when grazing is about 0.3-0.5 $d^{-1}$? Is that due to a net positive growth in bacterial abundance? If so, it would be good to mention here. Otherwise the reader might assume steady state as I did here.
*Indeed this is due to a positive net growth rate, as stated in the preceding sentence. For clarity we changed the terminology to 'gross growth rates' now. Furthermore, we have moved the actual rate information to the Results to accommodate also Reviewer's comment to include less results and more discussion. We added more discussion on the estimated gross growth rate in comparison to bacterial production rates measured by others (Hornick et al., this issue).*

35. - p18, l27: "Also Pico II showed positively correlated net growth rates with $CO_2$ enrichment, but somewhat later into phase I (days 12-17) due to reduced losses." Awkward start of the sentence, maybe: "Net growth rates of Pico II correlated positively with $CO_2$ enrichment. . .. ."
*We thank Reviewer for the improvement and have corrected accordingly.*

36. - p20, l25-28: these are mainly results and then one other article is mentioned; but what does this now mean for your data? Do you think that TEP production was a factor regulating the abundances in your study? the actual discussion is missing

*We realize the sentence on TEP was a bit of a stand-alone and we chose to delete the sentence as it no longer fits the reworked discussion of these results.*

37.- p21, l9-11: This comes out of the blue. How did you examine this?
*This is examined using mytomycin C which induces prophage to go into the lytic state. It is explained in the M&M section 2.3. and we have also added this to the Results section in the revised manuscript.*

38. - p22, l2: ". . . has a very different physiology,. . ." different from what?
*We meant different from picoeukaryotes. We clarified this now in the text.*

39. - p22, l17: DOC could have come also from sloppy feeding?
*We added this option to the discussion on the topic.*

40. - p23, l17: do you mean remineralization of organic matter rather than nutrients?
*Yes thanks, we apologize for the mistake an altered the text accordingly.*

41. - p23, l22: "multiple other factors"? Please name them here.
*We now rephrased the sentence and specifically named SST and stratification.*

42. - Fig. 2: Instead of calling it the 'total prokaryotic phytoplankton', just call it what it is, the Synechococcus population
*We have altered the figure accordingly.*

43. - Fig. 2: How is the p<0.1 indicated? Is it possibly also the category 'p>0.05'?
- What are the black dots here and in other plots (and I don't mean the single asterisks)?
*The black dots are the p<0.1 indicators. We have clarified the legend accordingly.*

44. - I don't see any 'f' in this figure (and some of the following figures).
*True, this was our mistake as not all figures have failed experiments (f). We have deleted this from those figure legends.*

45.- Fig. 2: panel b here and in following figures: I understand why the authors want to present the data together, but the plots are really obscured this way and it makes it hard for the reader to discern any data from them. I would suggest to split them into two panels

**We have split panel b into two panels for clarity**

46.- Fig. 2: Here and following figures, what do you mean by "otherwise no data is a zero"?
*We referred to assays with true zero rates. We understand the misunderstanding and now indicate true zeros (thus not failed assays) with a "0" and made this clear in the figure legends.*

47. - Fig. 6: The legend says that linear regression statistics are provided in the plot, however, I couldn't see a p value.

*We apologize as this was a left-over from an earlier version. We show only the $r^2$, and made this clear in the legends of the revised manuscript.*

48.- Fig. 10: the grazing rate and lysis rate are both loss rates; nevertheless, one of them is presented on a negative scale while the other is presented on a positive scale. I find it confusing.
*We agree with Reviewer and have changed the figure now such that all loss rates are presented on a negative scale.*

[Figure]

49. - Suppl. Table S1: What are the units here?
*They are abundances per milliliter; we have edited this.*

50. - Suppl. Table S2: What are the units here?
*They are rates per day; we have edited this.*

51. - Fig. S1 and S2 legends: the upper layer is mentioned here but the measurements are from the total water mass, i.e. 0.3-17 m rather than 0.3-10 m
*We corrected the figure legend.*

52. - Fig. S2: panel f is missing
*We corrected the figure legend.*

53. - Fig. S3 and S4 please use the proper symbol for micromol and not umol
*We corrected accordingly.*

*Garrison, D. L., and others. 2000. Microbial food web structure in the Arabian Sea: A US JGOFS study. Deep-Sea Res. Part II 47: 1387-1422. doi:10.1016/S0967-0645(99)001 48-4*

*Hornick T., Bach, L. T., Crawfurd, K.J., Spilling, K., Achterberg E.P., Brussaard, C.P.D., Riebesell, U., Grossart, H-P. Effect of ocean acidification on bacterial dynamics during a low productive late summer situation in the Baltic Sea (in prep.) Biogeosciences, 2015.*

Kimmance, S. A. and Brussaard, C. P. D.: Estimation of viral-induced phytoplankton mortality using the, in Manual of Aquatic Viral Ecology, pp. 65–73., 2010.

Kuosa, H.: Picoplanktonic algae in the northern Baltic Sea: seasonal dynamics and flagellate grazing, Mar. Ecol. Prog. Ser., 73(2-3), 269–276, doi:10.3354/meps073269, 1991.

Mojica, K. D. A., van de Poll, W. H., Kehoe, M., Huisman, J., Timmermans, K. R., Buma, A. G. J., van der Woerd, H. J., Hahn-Woernle, L., Dijkstra, H. A. and Brussaard, C. P. D.: Phytoplankton community structure in relation to vertical stratification along a north-south gradient in the Northeast Atlantic Ocean, Limnol. Oceanogr., 60(5), 1498–1521, doi:10.1002/lno.10113, 2015.

Paul, A. J., Bach, L. T., Boxhammer, T., Czerny, J., Hellemann, D., Trense, Y., Nausch, M., Sswat, M., Riebesell, U., Road, M., Lismore, E. and Way, E.: Effect of elevated $CO_2$ on organic matter pools and fluxes in a summer , post spring-bloom Baltic Sea plankton community,Biogeosciences , 1–60, 2015.

Stoecker, D. K., Nejstgaard, J. C., Madhusoodhanan, R., Pohnert, G., Wolfram, S., Jakobsen, H. H., Šulčius, S. and Larsen, A. (2015), Underestimation of microzooplankton grazing in dilution experiments due to inhibition of phytoplankton growth. Limnol. Oceanogr., 60: 1426–1438. doi:10.1002/lno.10106

Worden, A. Z., J. K. Nolan, and B. Palenik. 2004. Assessing the dynamics and ecology of marine picophytoplankton: The importance of the eukaryotic component. Limnol.Oceanogr. 49: 168-179. doi:10.4319/lo.2004.49.1.0168

Biogeosciences Discuss., doi:10.5194/bg-2015-606-RC1, 2016
Interactive comment (in bold, italic) on "Shifts in the microbial community in the Baltic Sea with increasing $CO_2$" by K. J. Crawfurd et al.

**Also supplied as pdf with Figs embedded in the text (see supplement)**

***We thank the reviewers for taking the time to review this manuscript and for the pertinent and constructive comments they have raised. Wherever possible we have incorporated their suggestions and if not I hope that we have clearly explained our reasoning.***

**Anonymous Referee #2**

The authors present mesocomos experiment in which they test the effect of ocean acidification on microbial community, by increasing CO2 levels. Since the ongoing climate change this research is of high importance and data presented in this msc are very valuable. The authors focused on the phytoplankton size fraction ËC20´ µm, as well as to heterotrophic procaryotes and viruses. The experiment set u is well explained and the msc is in general well written. I do recommend the msc to be published after major revision Key points that I would recommend to be answered:

1. All though, the target organisms are of key importance for ecosystem functioning, I do not agree that the results are relevant if not being in correlation to the whole phytoplankton community, at least presented as Chl a concentration. If those data exist I highly recommend including them in the msc.

***We believe abundance and cell size of the different phytoplankton groups are of key importance as Chlorophyll a consisted mainly of algal groups smaller than 20 µm cell diameter (Paul et al. 2016). Already at the start of the experiment less than 5% was larger than 20 µm diameter. At day 5 even 70% was smaller than 2 µm. Therefore adding Chl a concentration will not add to the current study. We made this clear in the Discussion of the revised manuscript.***

2. In Material and Methods it is stated that the samples of surrounding water were taken, but there are not presented in results or at least discussed. It is essential to discuss those data. I would recommend including those data into the graphs reporting about the microbial community changes.

*We chose not to include them in the main figures for 2 reasons. Firstly they are not directly comparable, this location is subject to water movement and during the experiment distinctly different water masses with different physical and biological signatures moved into the surrounding water. Secondly, and due to the previous reasoning, including the phytoplankton abundances in the surrounding waters makes the figures more difficult to read. Occassionally the abundances are much greater than in the mesocosms and it is then harder to discern differences between the mesocosms (see Supplementary Table S1 and Fig S1). Overall, microbial temporal dynamics are largely comparable, with a few exceptions: i.e., phytoplankton Nano I and II show much higher abundances in the outside water whilst all the picoplankton abundances are lower in the surrounding waters. We will add a description on the outside water microbial dynamics in the Results section of the revised manuscript, and add discussion on what may have caused the differences.*

3. Also, I am not sure if I have understood it correctly – but it seems that CO2 was added to all mesocosms? In that way you do not have control and any change in microbial community could have been because of some other factor?

*We realize that the text was not clear and have improved this in the revised manuscript, i.e., all mesocosms were sparged with water so that a similar water treatment occurred, but no $CO_2$ was added to the mesocosms that served as present-day controls.*

4. What about the temperature?

*The temperature was similar for all mesocosms as well as the surrounding water and therefore can only potentially have influenced the dynamics of the microbial populations but not the extent of change between the different mesocosms. We present temperature now briefly at the start of the Results section (of the revised manuscript).*

5. What is the usual phytoplankton development dynamics in the Baltic Sea?

*We now briefly commented on this at the start of the Discussion.*

6. Also, I do not see any shifts in microbial community, but changes abundances during the experiment.

*We acknowledge that and clarify in the Discussion of the revised manuscript that the extent of temporal dynamics differed for the high pCO2 mesocosms (and not the dynamics*

*itself). Also we changed the title of the manuscript to "Shifts in the size structure of the microbial community in the Baltic Sea with increasing CO$_2$" to make this clear.*

7. Since every experiment need to be repetitive and results comparable with other study site and experiment set up it would be necessary to know the community structure of microbial community. I am aware of the difficulty of taxonomical recognition of size fraction below 20µm, but it is essential. Flow cytometry in an excellent tool, but in an experiment set up of this range I do not believe it is enough. I did like the way how the investigated groups were divided, but the next step in research should be the taxonomical identification.

*We agree that taxonomic identification would be a next step but would need flow cytometry sorting and genomics, which goes beyond the scope of the current study. Linking to taxonomic identifcation by phytoplankton pigment composition analysis is only partly possible as a few large cells may obscure the share of a certain group as compared to total Chl a. Paul et al. (2015) showed that the smaller fraction (<2 µm) was likely to be chlorophytes and prasinophtes. This is mentioned in the Discussion (section 4.1).*

8. Not all organisms in the same size fraction have same physiological response to environment drivers.

*We recognise this and discussed this in relation to the Pico I and Syn data. We will add a similar line of reasoning for potential differences within a group even as not all are necessarily the same species.*

9. Also, the authors in the discussion, do not discuss their data, but cite different authors and their research. If you did not go to the species level – those data cannot be discussed.

*We checked the Discussion and amended where needed.*

10. The analyzed groups were good explained except the cyanobacteria. The authors distinguish Synechococcus, but no Prochlorococcus. Since the oligotrophy Prochlorococcus could develop in the environment and it can be separated by flow cytometry. Then stated that the prokaryotes include bacteria, archea and unicellular cyanobacteria (together marked as HP) – what cyanobacteria could it be?

*Prochlorococcus can indeed be distinguished by flow cytometry, but was not present during this experiment. Therefore cyanobacteria and Synechococcus are used interchangeably during the manuscript. We will make a statement that Prochlorococcus was not observed (Results section).*

11. The tittle also does not represent the results – there is not shift in microbial community presented.

*We have altered the title to "Shifts in the size structure of the microbial community in the Baltic Sea with increasing $fCO_2$ " to more accurately reflect the results.*

12. It is not easy to follow the results and discussion with given abbreviations (M1, M5. . .) for mesocosm experiments. It is not clear enough. Maybe a table would be a good way to explain the abbreviations.

*The notation used is consistent across all manuscripts in this special issue and the mesocosms with their mean $fCO_2$ are presented in Fig. 1. However, we see Reviewer's point of view and will include a Table with the necessary information. See Attachment 1.*

*Table 1. $fCO_2$ concentrations (μatm) as an average for the duration of the experiment following $CO_2$ addition and specification of this $CO_2$ level as low, medium or high. \*denotes mesocosms sampled for grazing and viral lysis assays*

| Mesocosm | M1* | M5 | M7 | M6 | M3* | M8 |
|---|---|---|---|---|---|---|
| $CO_2$ Level | LOW | LOW | INTERMEDIATE | INTERMEDIATE | HIGH | HIGH |
| Mean $fCO_2$ (μatm) days 1-43 | 365 | 368 | 497 | 821 | 1007 | 1231 |

13. The discussion needs to be rewritten. The results would need to be discussed in more detail. The cited literature and results are maybe not the best choice for the results presented.

*It is not clear which examples the reviewer is referring to. However, we have tried to improve the discussion according to Reviewer's comments.*

*Paul, A. J., Bach, L. T., Boxhammer, T., Czerny, J., Hellemann, D., Trense, Y., Nausch, M., Sswat, M., Riebesell, U., Road, M., Lismore, E. and Way, E.: Effect of elevated $CO_2$ on organic matter pools and fluxes in a summer , post spring-bloom Baltic Sea plankton community,Biogeosciences , 1–60, 2015.*

**Changes made:**

1. All comments addressed in text-see track changed version below
2. Addition of Table 1.
3. Addition of Fig 10d
4. All Figs. altered as reviewer requested

**Shifts in the size structure of the microbial community in the Baltic Sea with increasing $f$CO$_2$**

K. J. Crawfurd[1], U. Riebesell[2] , C. P. D. Brussaard[1,3]

[1]{NIOZ Royal Netherlands Institute for Sea Research, Department of Marine Microbiology and

Biogeochemistry, and Utrecht University, P.O. Box 59, 1790 AB Den Burg, Texel, The Netherlands}

Department of Biological Oceanography, NIOZ – Royal Netherlands Institute for Sea Research, PO Box

59, 1790 AB Den Burg, Texel, The Netherlands}

[2]{GEOMAR Helmholtz Centre for Ocean Research Kiel, Biological Oceanography, Düsternbrooker

Weg 20, 24105, Kiel, Germany}

[3]{Aquatic Microbiology, Institute for Biodiversity and Ecosystem Dynamics, University of

Amsterdam, P.O. Box 94248, 1090 GE Amsterdam, The Netherlands}

Correspondence to: kate.crawfurd@nioz.nlgmail.com

**Abstract:**

Ocean acidification, due to dissolution of anthropogenically produced carbon dioxide is considered a major threat to marine ecosystems. The Baltic Sea, The Gulf of Finland, in the Baltic Sea haswith extremely low salinity and thus low pH buffering capacity, so is likely to experience stronger variation in pH than the open ocean with increasing atmospheric carbon dioxide. We examined the effects of ocean acidification on the microbial community structure in the Gulf of Finland, Baltic Sea, during the, low salinity (around 5.7), and inorganic nitrogen and phosophorus depleted, summer. u Using large volume *in situ* mesocosms to simulate present to future and far future scenarios,. W we observedsaw distinct trends with increasing $f$CO$_2$ in each of the 6 groups of phytoplankton with diameters below 20 µm that we enumerated by flow cytometry (<20 µm cell diameter). Of these groups 2two picoeukaryotic groups increased in abundance whilst the other groups, including prokaryotic *Synechococcus* spp., decreased with increasing $f$CO$_2$. Gross growth rates increased with increasing $f$CO$_2$ in the dominant picoeukaryote group sufficient to double their abundances whilst reduced grazing losses allowed the other picoeukaryotes to flourish at higher $f$CO$_2$. Significant increases in lysis rates were seen at higher $f$CO$_2$ in these two picoeukaryote groups. Converting abundances to particulate organic carbon we saw a large shift in the partitioning of carbon between the size fractions which lasted throughout the experiment. The heterotrophic prokaryotes largely followed the algal biomass with responses to increasing $f$CO$_2$ reflecting the altered phytoplankton community dynamics. Similarly, higher viral abundances at higher $f$CO$_2$ seemed related to increased prokaryote biomass. Viral lysis and grazing were equally both important in controlling prokaryotic abundances. Overall our results point to a shift towards a more regenerative system with potentially increased productivity but reduced carbon export.

**1 Introduction**

Comment [k1]: Extensively altered so no track changes

Ocean acidification (OA) caused by anthropogenic carbon dioxide (CO$_2$) release and its subsequent dissolution in the oceans is considered one of the great threats that facing marine ecosystems face (Turley and Boot, 2010). Direct and indirect effects are predicted to have a large impact on marine these ecosystems (IPCC, 2007). Phytoplankton production has been found susceptible to OA, depending on the phytoplankton community composition (eg. Hein and Sand-Jensen, 1997; Tortell et al., 2002; Leonardos and Geider, 2005; Engel et al., 2007; Feng et al., 2009); )(Hein and Sand-Jensen, 1997)(Leonardos and Geider, 2005;. Calcification of coccolithophores, which influence sedimentation via calcium carbonate ballasting, is generally reduced (Meyer and Riebesell, 2015).

Diatoms, important for organic matter burial, have been found to benefit in some cases (Feng et al., 2009) but not in others (Tortell et al., 2002). Certain cyanobacteria, including diazatrophs, have been seen to benefit from elevated $CO_2$ concentrations (Qiu and Gao, 2002; Barcelos e Ramos et al., 2007; Hutchins, 2007). Direct $CO_2$ effects are also reported  for small-sized photoautotrophic eukaryotes (Engel et al., 2007; Meakin and Wyman, 2011; Brussaard et al., 2013).

Marine phytoplankton are responsible for approximately half of global primary production (Field et al., 1998), whereby shelf sea communities contribute 15-30% of this (Kulinski and Pempkowiak, 2011). Whilst environmental factors, such as temperature, light, nutrients and $CO_2$ concentration, regulate gross primary production bottom-up,  loss- factors (i.e., grazing, viral lysis and sedimentation) determine the fate of the carbon fixed by phytoplankton. Ingested carbon transfers to higher trophic levels, sinking of phytoplankton and faeces may lead to carbon storage in sediments, and viral lysis is a major driver of carbon release to dissolved and detrital organic matter (DOM; Wilhelm and Suttle, 1999; Brussaard et al., 2005; Lønborg et al., 2013). Through viral lysis the cell content of the host is released into the surrounding water and utilized by heterotrophic bacteria, thereby stimulating the microbial loop (Brussaard et al., 2008; Sheik et al., 2014). Bacteria may also be affected either directly by OA, or indirectly via changes in the quality or quantity of DOM (Weinbauer et al., 2011). Viral lysis has been found to be at least as important a loss factor as microzooplankton grazing- for natural bacterio- and phytoplankton (Weinbauer, 2004; Baudoux et al., 2006; Evans and Brussaard, 2012; Mojica et al., 2016).

The effect of ocean acidification on the relative share of these key loss processes is, however, still understudied for most ecosystems, particularly for brackish coastal systems. Low salinity affects the pH buffering capacity due to low total alkalinity and is as such of interest for OA studies. Here we report on the temporal dynamics of microbes (phytoplankton,  prokaryotes and viruses) under the influence of enhanced $CO_2$ concentrations and in relation to viral lysis and grazing control. Using large mesocosms at *in situ* light and temperature, the Baltic Sea pelagic ecosystem was exposed to a range of increasing *f*CO$_2$ concentrations from ambient to future and far- future concentrations. This study was performed during summer in the Gulf of Finland near

Tvärminne, with salinity around 5.7 and low dissolved inorganic nitrogen and phosphorus concentrations. During the 43 day long experiment  the smallest picoeukaryotic phytoplankton especially showed distinct responses to the treatment conditions.

**2 Materials and Methods**

**2.1 Study site and experimental set-up**

The study was conducted in the Tvärminne Storfjärden (59° 51.5' N, 23° 15.5' E) between 14 June and 7 August, 2012. Nine mesocosms each enclosing ~ 55 m$^3$ of water with a depth of 17 m were moored in a square arrangement within the archipelago. For details on the experimental set-up, carbonate chemistry dynamics and nutrient concentrations throughout the experiment we refer to the general overview paper  by Paul et al. (2015, this issue).  After deployment the mesocosms were kept open for 5 days with 3 mm mesh screening over the top and bottom openings before being closed at the bottom and pulled above the sea surface at the top.

Photosynthetically active radiation (PAR) transparent plastic hoods (open on the side) prevented rain and bird droppings from entering the mesocosms. Six mesocosms were sampled for the current study, unfortunately three were lost due to leakage.  Initial fugacity of CO$_2$ (*f*CO$_2$) was 240 µatm.

The mean *f*CO$_2$ during the experiment, i.e. days 1-43, for the individual mesocosms was as follows: M1,

365 µatm; M3, 1007 µatm; M5, 368 µatm; M6, 821 µatm; M7, 497 µatm; M8, 1231 µatm .

Throughout this study we refer to *f*CO$_2$ which takes into account the non-ideal behavior of CO$_2$ gas and is the standard measurement required  for gas exchange calculations (Pfeil et al., 2013).

For $f\text{CO}_2$ manipulations, natural seawater was saturated with $\text{CO}_2$ and then injected evenly throughout the whole depth of the mesocosms in four steps between days 0 to 3 until target $f\text{CO}_2$

was reached. On day 15 a further $f\text{CO}_2$ addition was made to the top 7 m of mesocosms 3, 6, and 8

to replace $\text{CO}_2$ lost due to outgassing.  The remaining mesocosms received similar treatment without

$\text{CO}_2$. Initial nutrient concentrations, i.e.  nitrate, phosphate, silicate and ammonium, were 0.05 (µmol L$^{-1}$), 0.15 (µmol L$^{-1}$), 6.2 (µmol L$^{-1}$) and 0.2 (µmol L$^{-1}$), respectively, and stayed low  for the duration of the experiment (Paul et al., 2015, this issue).  Salinity was around 5.7, temperature was initially ≈8°C and rose to ≈15°C on day 15 before falling to ≈8°C again.

Collective sampling was performed daily in the morning, using an integrated water sampler, from the top (0-10 m) and from the whole water column (0-17 m) of all mesocosms and the surrounding water.  Subsamples were obtained for enumeration of phytoplankton,  prokaryotes and viruses. Samples for viral lysis and grazing were taken from 5 m depth using a gentle vacuum-driven pump system. Samples were protected against daylight and warming by thick black plastic bags containing wet ice. In the laboratory the samples were processed at *in situ* temperature and dimmed light. As viral lysis and grazing rates were determined from samples taken from 5 m depth, samples for microbial abundances reported were taken from the top 10 m integrated samples. For abundances from 0-17 m and the surrounding water see Supplementary data (Table S1 and  Fig.S1).

The experiment has been divided into 4 phases based on major physical and biological changes occurring (Paul et al., 2015). Phase 0 before $\text{CO}_2$ addition (days -5 to 0), phase I (days 1-16), phase II

(days 17-22) and phase III (days 23-43). Throughout this study the data are presented using 3 colors (blue, grey and red), representing low (mesocosms M1 and M5) intermediate (M6 and M7) and high (M3 and M8) $f\text{CO}_2$ (Table 1) .

**2.2 Microbial abundances**

Microbes were enumerated using a Becton Dickinson FACSCalibur flow cytometer (FCM) equipped with a 488 nm argon laser. The photoautotrophic cells (<20 μm) were counted directly fresh and were discriminated by their autofluorescent pigments (Marie et al., 1999). The samples were held on wet ice in the dark until counting. Based on their chlorophyll red autofluorescence and the presence of phycoerythrin orange autofluorescence in combination with side scatter signal, the phytoplankton community could be divided into 6 clusters. Phytoplankton cell size of the different phytoplankton clusters was determined by gentle filtration through 25 mm diameter polycarbonate filters (Whatman) with a range of pore sizes (12, 10, 8, 5, 3, 2, 1, and 0.8 μm) according to Veldhuis and

Kraay (2004). Average cell sizes of the different phytoplankton groups were 1, 1, 3, 2.9, 5.2, and 8.8

μm diameter for the prokaryotic cyanobacteria *Synechococcus* spp. (SYN), picoeukaryotic phytoplankton I, II and III (Pico I-III), and nanoeukaryotic phytoplankton I, and II (Nano I, II), respectively. Pico III was discriminated from Pico I (comparable average cell size) by  higher orange autofluorescence. Cyanobacterial species *Prochlorococcus* spp. were not observed during this experiment. Assuming the cells to be spherical and  applying conversion factors of 237 fg C μm$^{-3}$ (Worden et al., 2004) and 196.5 fg C μm$^{-3}$ (Garrison et al., 2000)

for pico- and nano-sized plankton, respectively, cellular carbon was calculated based on the average cell diameters. Net growth and loss rates of phytoplankton and heterotrophic prokaryotes were derived from exponential regression analysis of the cell abundances.

[revised manuscript text omitted]

and day 24 (phase II; Fig. 1a). At the end of phase I the high $f\mathrm{CO_2}$ mesocosms displayed higher phytoplankton abundance than the present day (low) $f\mathrm{CO_2}$, whereas the opposite was found for days

17-22. These trends were largely due to the prokaryotic cyanobacteria *Synechococcus* spp., making up on average 74% of total abundance.  In contrast, the total eukaryotic phytoplankton showed a strong positive effect of $f\mathrm{CO_2}$ (Fig. 1b), due to the response of Pico I and II. For all phytoplankton groups, except *Synechococcus* and Pico III, we found that the algal abundances in the surrounding water (Table S1) largely comparable to the temporal dynamics in the mesocosms, with only occasionally higher abundances for the nanoeukaryotic phytoplankton groups and lower abundances for Pico I and II (Table S1, Fig. S1). The surrounding waters were more similar to the low $f\mathrm{CO_2}$ than the high $f\mathrm{CO_2}$ mesocosms, demonstrating that the differences between the low and high $f\mathrm{CO_2}$

mesocosms are the effect of the elevated $f\mathrm{CO_2}$. Phytoplankton, prokaryotes and viral abundances in the 0-17m samples were generally lower but showed similar dynamics (Figs. S1 and S2).

**3.1.1 *Synechococcus***

*Synechococcus* (SYN) showed an initial peak in abundance on day 4 (Fig. 2a), then abundances declined, most so for the low $fCO_2$ mesocosms from days 4-7. The net growth rate was  strongly negatively correlated with $fCO_2$ ($R^2$=0.98, Fig. 2d). The loss measurements (only grazing, no viral lysis detected) confirmed that the total loss rate for the low $fCO_2$ mesocosm M1 was significantly higher than for the high $fCO_2$ mesocosm M3 on day 10 (0.56 vs 0.27 d$^{-1}$), whilst the gross growth rate did not differ significantly (Fig. 2b). Cell abundances increased again from day 12.In the low $fCO_2$ mesocosms this continued until the bloom at day 24, whilst the high $CO_2$ mesocosms peaked at day 15 and then dropped again before increasing from days 19-24. Despite the deviation in temporal dynamics between the treatments SYN abundance peaked at day 24 in all mesocosms with around 4.5 x 10$^5$ cells ml$^{-1}$ (Fig 2a) and was negatively correlated with $fCO_2$ ($R^2$=0.77). total net production during this bloom was greater in the low $fCO_2$ mesocosms than in the high ones as initial abundances were lower  (day 13) and peak abundances higher (day 24; Fig. 2a). This could be explained by a higher total loss rate for M3 than M1 on day 17 (0.33 vs 0.17). The followingfollowing~~ (days 24-28) seemed largely due to reduced gross growth rates (Fig. 2b). Thereafter the trend was not so clear until the end of the experiment.

**3.1.2 Picoeukaryotes I**

Pico I was numerically the second most dominant group of phytoplankton, 26% of total phytoplankton abundances on average in the high $fCO_2$ mesocosms and 21% in the low $CO_2$ mesocosms. This amounts to 15% of total POC at high $fCO_2$, 10% at low $fCO_2$ (mean of total POC). The initial increase (peak in abundance at day 5, Fig. 3a) of these small-sized (mean cell diameter ≈1 µm, comparable to SYN) phytoplankton already showed a slight positive trend and strong correlation with $fCO_2$ for the net growth rate (Fig. 3d, $R^2$=0.95) and abundance (Fig. 3g, $R^2$=0.8). The higher loss rates (days 5 to 9; Fig. 3e) resulted in a decrease in abundance, which was stronger for the low $f$CO$_2$ mesocosms (as illustrated by M1) due to the significantly higher gross growth rates for the high $f$CO$_2$ mesocosm (represented by M3; Fig. 3b). The positive correlation of

Pico I peak abundance with $f$CO$_2$ on day 13 (Fig. 3g, R$^2$=0.94) was lost upon another decline in abundance. Significantly higher losses at high $f$CO$_2$, a combination of grazing and lysis, resulted in a more dramatic crash at high $f$CO$_2$ and abundances becoming similar again around day 17 (Fig. 3a).

Viral lysis was a significant loss factor compared to grazing, i.e. overall on average 45% and 70% of total losses in M1 and M3, respectively (Table S2). An extra addition of CO$_2$ was given to M3, M6 and

M8 because their $f$CO$_2$  had approached that of the remaining mesocosms. This may have stimulated the gross growth in M3  for a longer period in the high $f$CO$_2$ mesocosms as compared to M1 (day 19; Fig. 3b).Combined with higher losses at low $f$CO$_2$   positive correlation of net growth rates with $f$CO$_2$ was seen (Fig. 3e, R$^2$=0.71),  and almost 2-fold higher  abundances at high $f$CO$_2$ on day 21 (Fig.

3ai, R$^2$=0.84). Pico I was thus greatly stimulated by increased $f$CO$_2$, from day 3 throughout the experiment. Standing stock of Pico I remained higher at high $f$CO$_2$ for the further duration of the experiment (Fig. 3a), with gross growth matched by total losses (Fig.3b).

Surprisingly the higher abundances did not stimulate higher losses during this period, grazing rates were very low in both M1 and M3, and viral lysis was totally responsible for losses on day 31 in both mesocosms (Table S2).

**3.1.3 Picoeukaryotes II**

A group of larger picoeukaryotes, Pico II (mean diameter of 3 μm) bloomed exactly during the period

Pico I was low in standing stock (days 13-21, Fig. 4a) and the peak abundance (day 17) correlated positively with $f$CO$_2$ (Fig. 4d). Relatively high total losses of (0.46 and 0.58 d$^{-1}$

in the low and high $f$CO$_2$ mesocosms, respectively (average days 6-13) accompanied the high gross growth rates (0.69 and 0.72 d$^{-1}$) for the same period (Fig. 4b). These indicate high turnover and explain the slow rate of increase in cell abundance until day 13 (Fig. 4a). During the bloom period of Pico II, losses were smaller than the gross growth rate, more so it seems for M3 than M1 (Fig. 4b).

Resultant net growth rates correlated with $f$CO$_2$ (Fig. 4d e, $R^2$=0.82) with peak abundances 1.4 fold higher at high $f$CO$_2$ (Fig. 4a ). Higher losses at high $f$CO$_2$ then contributed to the faster decline in abundances at high $f$CO$_2$. Phase III was a period of low turnover for Pico II with low gross growth and loss rates resulting in quite stable cell abundances, still higher at high $f$CO$_2$, until day 29 after which they declined in all mesocosms (Fig. 4a).

**3.1.4 Picoeukaryotes III**

Another group with around 2.9 μm cell diameter could be discriminated from Pico II by its higher orange autofluorescence mainly, and as such may represent small-sized cryptophytes. This is just at the lower size range of small cryptophyte (Klaveness, 1989).  This group (Pico III) had its highest abundances during phases II and III (days 17-43, Fig. 5a), with a distinct negative correlation to $f$CO$_2$

(Fig. 5d 5e, $R^2$=0.91). Already directly upon the first $f$CO$_2$ addition (days 0-4) the abundances declined for the high $f$CO$_2$ mesocosms (Fig. 5a) with net growth rates negatively correlated to $f$CO$_2$ (Fig. 5d e,

$R^2$=0.94). Gross growth rates were indeed significantly higher for M1 than M3 at days 1, 4 , and 10

(Fig. 5b). Abundances of the Pico III group in the ambient Fjord surrounding water followed the low

$f$CO$_2$ mesocosms perfectly during this first period, indicating that the crash in the high $f$CO$_2$

mesocosms was indeed a direct (negative) effect of $f$CO$_2$ (Table S1). A similar response of Pico III

abundance halting in the high $f$CO$_2$ mesocosms and strongly increasing in the low $f$CO$_2$ mesocosms occurred directly after the additional $f$CO$_2$ purge (day 15). Losses were largely due to microzooplankton grazing. Unfortunately about half of the loss assays in the second half of the experiment failed (for unknown reasons), yet the successful assays suggest that losses were minor (Fig. 5b). There may also be larger cryptophytes present in the community, not counted by the flow cytometer because our data show Pico III most dominant in phase III whilst the specific pigment data shows a decline from phases 0 to -III.

**3.1.5 Nanoeukaryotes I**

The nanoeukaryotes group Nano I consisted of cells with a mean diameter of 5.2 μm and were found with maximum abundances of 5.5 x$10^2$ ml$^{-1}$ (Fig. 6a). After an initial peak at day 6, the lower $f$CO$_2$

mesocosms showed the highest numbers at day 17 (Fig. 6a). This seems initiated by 2.3-fold higher total loss rates for M3 than M1 on days 6 and 10 (Fig. 6b) in combination with 2-fold lower gross growth rates on day 10 (Fig. 6b). Ultimately, this lead to net growth rates correlating negatively with $f$CO$_2$ for days 10-12 (Fig. 6d, R$^2$=0.83). Viral lysis occurred  predominantly in the high $f$CO$_2$ mesocosm throughout the experiment with rates ranging from 0.13 to 0.7 day$^{-1}$

(making up 16 to 98 % of total losses; Table S2). A group of viruses which had a flow cytometric signal typical for viruses infecting nanoeukaryotes (V4) were identified but no obvious correlation was found with any of the phytoplankton groups. Lower total loss rates at days 13 and 17 in both mesocosms allowed a small increase in abundance, peaking on day 17 and negatively correlated to

$f$CO$_2$ (Fig. 6e, R$^2$=0.67).

**3.1.6 Nanoeukaryotes II**

The temporal dynamics of Nano II were rather erratic (Fig. 7a). Nano II were the largest in size and may have been made up by different phytoplankton species, however due to their low numbers we were unable to discriminate separate groups. The peak in abundance at day 16 showed a negative correlation to $f$CO$_2$ (Fig. 7e, R$^2$=0.61), and was the result of an overall reduced net growth rate with

$f$CO$_2$ (Fig. 7d, R$^2$=0.56). The subsequent decline seems the result of reduced gross growth rate (to even zero) and increased loss rate (day 20; Fig. 7b).

**3.1.7 Algal POC**

The calculated mean algal POC shows that $f$CO$_2$ had a clear positive effect on the biomass of Pico I

and II (Fig.  8a; p<0.0001). The effect became noticeable only a few days into the experiment and the mean Pico I and II POC concentration in the high $f$CO$_2$ mesocosms stayed high

 for the entire duration of the experiment. At the same time the remaining algal groups showed reduced POC at enhanced $f$CO$_2$ (the sum of Pico III, and Nano I and II and *Synechococcus*

spp.; Fig. 8b, p<0.01). Particularly Pico III showed a nearly instant and markedly negative response to increased $f$CO$_2$ concentration (Fig. S3a). This was a lasting effect as the strongest difference was found in the second half of the experiment. For Nano I and II the higher algal POC concentrations became only apparent from the end of phase I and during phase II

(days 14-20; Fig. S3b).

Due to its small cell size, the numerically dominant SYN accounted on average for 40% of total POC.

Due to the exclusion of 3 mesocosms (see Material and Methods), the number of $f$CO$_2$ treatments is reduced to 6, which limits the statistical power of the results. Still, our data show that the responses of the different phytoplankton groups to ocean acidification were evident and consistent.

**3.2 Prokaryote population dynamics**

The prokaryotic temporal dynamics in the mesocosms  resembled that in the outside waters (Fig. S2). In general prokaryote abundance in the mesocosms followed the total algal biomass, with an initial increase during the first days following the closure of the mesocosms (Fig. 9a). The increase was mainly due to the HDNA-prokaryotes (Fig. 9b). The total prokaryote abundance increased initially at a net growth rate of 0.19 d$^{-1}$, and more specifically at 0.22 and 0.14 d$^{-1}$ for the high and low

DNA prokaryotes respectively; (Fig. 9b and c). There was no significant difference in prokaryote abundance between the treatments at the first peak (day 4). However, grazing was significantly lower (0.3 d$^{-1}$) in high (M3) than in low (M1; 0.5 d$^{-1}$) CO$_2$ treatments, on both days 0 and 4, and at the same time viral lysis was slightly (3% higher at high CO$_2$ higher in the high (M1) as compared to the low $f$CO$_2$ mesocosm (M3) (Figs. 10a 10b and bc). The decline in prokaryote abundances from days 5 to 9 seemed due to declining phytoplankton biomass (Fig. 1a) and increasing viral lysis rates (12-16 % d$^{-1}$ representing 39% of total losses in M1 and 37% in M3 on day 11, Fig. 10b10c). The viral production assays did not show evidence of lysogeny. Viral lysis assays showed no evidence of lysogeny for the prokaryotic community during the experiment (all phases).

From days 10-15 prokaryote dynamics became clearly affected by $f$CO$_2$ with significantly higher abundances and net growth rates at higher $f$CO$_2$ (Fig. 9a). although we cannot exclude an indirect response due to altered algal dynamics in response to higher $f$CO$_2$. Both the HDNA and the LDNA-prokaryotes (peak abundance on day 13, Fig. 9b and c) showed significant correlation with $f$CO$_2$ (R$^2$= 0.92 and 0.79, respectively, total prokaryote R$^2$= 0.88, Fig. 10c10d). In the higher $f$CO$_2$ mesocosms the decline in prokaryote abundance following the peak at day 13 was largely the result of decreasing HDNA-prokaryote numbers (Fig. 9b). Grazing was indeed significantly higher in the high $f$CO$_2$ mesocosm M3 but the data for viral lysis were inconclusive due to a failed assay (for technical reasons) for M1 at day 14 (Fig. 10a 10b and bc). The significantly (p<0.01) higher viral abundances, particularly due to the V3 group, with highest green fluorescence, for the high $f$CO$_2$ mesocosms around that time (Figs. 11a and b) around that time do seem to indicate that viral lysis in the high $f$CO$_2$ mesocosms was higher.

During phase II prokaryote abundances increased steadily until day 24 (for both HDNA and LDNA), corresponding to increased algal biomass (Fig. 110e) and low grazing rates (0.1-0.2 d$^{-1}$; Fig. 10a10b).

Although the overall higher prokaryote standing stock in the low $f$CO$_2$ mesocosms was due to enhanced growth around day 16 (Fig. 9a), the net growth rates were comparable after day 17.

Moreover, the higher abundances were only found for the HDNA-prokaryotes (Fig. 9b and c). Viral lysis rates were higher for the low $f$CO$_2$ mesocosms (Fig. 10c). The higher prokaryote abundances in the low $f$CO$_2$ mesocosms appear thus  due to the lower grazing prior to the increase, i.e. at the end of phase I (day 14).

prokaryote abundance ultimately declined again during days 28-35, but less in M1  than in the other mesocosms (Fig. 9a). We unfortunately have no data of the prokaryote loss rates after day

25, however viral abundances increased at a steady rate of 2.2x10$^6$ d$^{-1}$ (to a maximum of 0.9x10$^8$ ml$^{-1}$

by day 39; Fig. 11a), implying that viral lysis was at least partly responsible for the decline in prokaryote abundance. There was no significant difference in viral abundances between the treatments during this period.

**4 Discussion**

At the start of the experiment the trophic conditions were typical for the Baltic Sea in summer, with depleted nutrient conditions, particularly nitrate (Paul et al., 2015), and a vertically stratified water column following the, diatom-dominated, spring bloom (Kuosa, 1991). The summer phytoplankton community was dominated by pico- and nano-sized phytoplankton, and  these phytoplankton groups are of key importance during the experiment. Already at the start of the experiment

 than 5% of the phytoplankton community was  smaller than 20 µm cell diameter

, and by day 5, 70% was smaller than 2 µm (Paul et al., 2015). The picoeukaryotic photoautotrophs Pico I and II showed a very strong fertilization effect with enhanced $f\mathrm{CO_2}$, directly following the initial $\mathrm{CO_2}$ additions until the end of the experiment.  At the same time, the rest of the phytoplankton (Pico III, Nano I and II, and the prokaryote *Synechococcus* spp.) showed reduced abundances at higher $f\mathrm{CO_2}$. These shifts in the size structure of the community  could be explained by examining the gross growth rates in combination with the losses of the individual groups.

Overall, microbial temporal dynamics in the  mesocosms were largely comparable to the surrounding water, with a few exceptions: i.e., phytoplankton Nano I and II

occasionally showed much higher abundances  whilst all the picoplankton abundances were lower in the surrounding waters. Higher abundances of nano-sized phytoplankton in the surrounding water  were likely due to upwelling of cold, $\mathrm{CO_2}$-rich deep water to the surface, bringing in inorganic nutrients,  particularly silicate (Paul et al., 2015). Average temperatures in all the mesocosms and surrounding waters were similar, with the upwelling reducing the temperature  from around 15 to 8°C during phase II.

 Along with reduced PAR (Paul et al., 2015) this generally reduced gross growth of the different phytoplankton groups

 however no synergistic effects with $f\mathrm{CO_2}$ could be ascertained.

The microbial population dynamics in the surrounding water more closely resembled those in the ambient $f\mathrm{CO_2}$

mesocosms, and more importantly the differences were in contrast to the shifts in phytoplankton group dynamics in response to $CO_2$ enrichment. This implies that enhanced $fCO_2$ was indeed responsible for the changes seen.

**4.1 Phase 0 (days -5 to 0), before $CO_2$ addition**

In most experimental work nutrients have been added to stimulate phytoplankton growth, therefore little data exists for oligotrophic phytoplankton communities (Brussaard et al., 2013) with smaller sized algae typically dominating as they are better competitors for the growth-limiting nutrients (Raven, 1998; Veldhuis et al., 2005). Phase 0 shows the natural state of the ecosystem at the start of the experiment, with indeed a summer community dominated by picophytoplankton. Consistency in phytoplankton abundances across the mesocosms confirmed good replication and baseline data prior to $CO_2$ manipulation. The flow cytometric phytoplankton community was dominated by cyanobacteria *Synechococcus* spp. (SYN) and the smallest picoeukaryotes (Pico I; both around 1 μm).

(Brussaard et al., 2013),  in which smaller sized algae, which are better competitors for nutrients tend to dominate (Raven, 1998; Veldhuis et al., 2005). From the start of the experiment the flow cytometric phytoplankton community (<20 μm cell diameter) was dominated by *Synechococcus* spp.

(SYN) and the smallest picoeukaryotes, (Pico I; both around 1 μm).  Picoeukaryotes are found in high numbers at this site throughout the year and *Synechococcus* only in summer when the temperatures are higher (Kuosa, 1991). Microscopic identification of picoeukaryotes is extremely difficult and no species have been described for the region (Kuosa, 1991), however, pigment analyses suggest that

Pico I and II are likely to be prasinophytes or other chlorophytes  (Paul et al., 2015). Ideally, performing molecular analyses on the specific algal groups sorted by flow cytometry  aids to identify group composition at the species level. Biomass of *Synechococcus* and Pico I increased steadily upon closure of the mesocosms due to high gross growth rates whilst the other groups dropped slightly in abundance. Our grazing rates of *Synechococcus* compare well to the average reported estimate of microzooplankton grazing on cyanobacteria in July in this region of 0.3 d$^{-1}$ (range 0.18-0.53 d$^{-1}$, Kuosa, 1991). The net growth rates of the total prokaryotic community The net growth rates of the pprokaryotes (0.19 d$^{-1}$) also increased, at rates of 0.22 d$^{-1}$ and 0.14 d$^{-1}$ for the high and low DNA (HDNAprokaryote and LDNAprokaryote), respectivelywere also comparable, similar to rates reported for this region (Kuosa, 1991). Because the losses (strongly dominated by grazing) Grazing rates were betweenaround 0.3-0.5 d$^{-1}$, thewith viral lysis rates of <2% d$^{-1}$), indicating that bacterialtheir gross growthproduction rates must have been around 0.5-0.67 d$^{-1}$.

**4.2 Phase I (days 1-16)**

According to Paul and coauthors (2015) this phasePhase I was characterised by high productivity and high organic matter turnover. Indeed we saw all phytoplankton groups bloom and we measured relatively high losses by grazing and viral lysis for all groups during phase I, responsible for the referred high turnover of organic matter. Certainly The prokaryotes responded positively to the increased algal productivity and viral lysis. More specifically, during phase I Pico I benefitted directly and most from enhanced $f$CO$_2$ as demonstrated by their significantly ($p<0.05$) higher gross growth rates. Net growth rates of Ppico II correlated positively with CO$_2$ enrichment, but somewhat later into phase I (days 12-17) due to reduced losses.

The stimulation of Pico I by elevated $f$CO$_2$ may be due to a stronger reliance on diffusive CO$_2$ entry compared to larger cells. Model simulations reveal that whilst near-cell CO$_2$/pH conditions are close to those of the bulk water for cells <5 μm in diameter, they diverge as cell diameters increase (Wolf-Gladrow and Riebesell, 1997; Flynn et al., 2012). This is due to the size-dependent thickness of the diffusive boundary layer, which determines the diffusional transport across the boundary layer and to the cell surface (Wolf-Gladrow and Riebesell, 1997; Flynn et al., 2012). It is suggested that larger cells may be more able to cope with $f$CO$_2$ variability as their carbon acquisition is more geared towards dealing with low CO$_2$ concentrations in their diffusive boundary, e.g. by means of active carbon acquisition and bicarbonate utilization (Wolf-Gladrow and Riebesell, 1997; Flynn et al., 2012).

However, as the Baltic Sea experiences particularly large seasonal fluctuations in pH and $f$CO$_2$

(Jansson et al., 2013) due to the low buffering capacity of the waters,  phytoplankton here may be expected to have a high degree of physiological plasticity. Previous mesocosm studies have reported enhanced abundances of the picoeukaryotic photoautotroph prasinophyte Micromonas pusilla at higher $f$CO$_2$ (Engel et al., 2007; Meakin and Wyman, 2011; Maat et al., 2014). Another summer mesocosm study in the Arctic revealed that even smaller picoeukaryotes, similar to Pico I in our study, showed a positive response to enhanced $f$CO$_2$ (Brussaard et al., 2013). Furthermore, Schaum et al. (2012) found that 16 ecotypes of Ostreococus tauri (also another prasinophyte similar in size to

Pico I) increased in growth rate by 1.4-1.7 fold at 1,000 µatm pCO$_2$ compared to 400 µatm pCO$_2$.  All ecotypes increased their photosynthetic rates and those with most plasticity, most able to vary their photosynthetic rate in response to changes in $f$CO$_2$, were most likely to increase in frequency in the community. It is likely that the picoeukaryotes in our study, which show stimulation by $f$CO$_2$ are adapted to a highly variable carbonate system regime and are able to increase their photosynthetic rate when additional CO$_2$ is available. This ability could allow them to outcompete other phytoplankton (e.g., nanoeukaryotes during phase I) in an environment where nutrients are scarce.

In this experiment nanoeukaryotes may have been outcompeted during phase I.

A net loss of 60% of the mean standing stock of Pico I at low, and 42% at high $f$CO$_2$ after day 5 was likely due to grazing, on average 0.26 d$^{-1}$, and lysis 0.18 d$^{-1}$ (lysis in M3 only). In general, grazing was a substantial loss factor for all phytoplankton groups during this period and additionally Pico I and II,

Nano I and II experienced noteworthy viral mediated mortality. The high grazing rates coincided with high abundances of the ciliate Myrionecta rubra at the start of the experiment (Lischka et al. 2015).

After day 10 abundances of most of the phytoplankton groups increased,  corresponding with a

Pico II population dynamics were, despite high gross growth rates, controlled by grazing at the start of the experiment, and only after a reduction in losses during phase II (more so for the high CO$_2$ mesocosms) could a bloom develop. For Nano I and Nano II the gross growth rates seemed to increase at higher $f$CO$_2$, but at the same time the losses also increased. However, differences in growth and loss rates were not statistically significant and thus it stays difficult to underpin why these phytoplankton groups peaked to higher abundances at lower $f$CO$_2$ in phase I. Potentially released competition for nutrients towards the end of phase I (the numerically dominant Pico I and SYN had declined in abundance by then) aided the increase of the nanoeukaryotes.

In general, grazing was a substantial loss factor for all phytoplankton groups during this period and additionally Pico I and II, Nano I and II experienced noteworthy viral mediated mortality. The high grazing rates coincided with high abundances of the ciliate *Myrionecta rubra* at the start of the experiment (Lischka et al., 2015).  After day 10 *M. rubra* abundances declined and correspondingly, abundances of most of the phytoplankton groups increased (Lischka et al., 2015). Occasionally grazing rates between the high $f$CO$_2$ (M3) and present-day low $f$CO$_2$ (M1) mesocosms differed significantly although no general trend could be observed. Very few studies have examined the effects of OA on microzooplankton grazing of phytoplankton (Suffrian et al., 2008; Rose et al., 2009; Brussaard et al., 2013). In neither of 2 mesocosm experiments did Suffrian et al. (2008) nor Brussaard (2013) see significant effects on grazing rates.  However, in an on-board continuous culture experiment Rose et al. (2009) found that at elevated CO$_2$ concentrations higher prey abundances led to higher grazing rates. Similarly, Pico III in the current study during phase I, was strongly negatively affected by CO$_2$ and showed congruently lower grazing rates at higher $f$CO$_2$. Nonetheless, this did not seem to hold for the high abundance groups SYN and Pico I, nor for Pico II with comparable abundances to Pico III. Alternatively the significantly reduced gross growth rates at high $f\text{CO}_2$ are the more likely cause for the clear differences in population dynamics between high and low $f\text{CO}_2$ treatments.

In contrast, higher gross growth rates alongside a predominance of viral lysis at high $f\text{CO}_2$ was seen in both Pico II and Nano I during phase I.  Metabolically active cells were reported to be infected at higher rates and phytoplankton growing at higher growth rates produced more viral progeny, which could explain this observation (Bratbak et al., 1998; Weinbauer, 2004; Maat et al., 2014). Direct effects of higher $f\text{CO}_2$ on viruses themselves are not expected as marine virus isolates were found to be quite stable (both particle and infectivity) over the range of pH obtained in the present study (Danovaro et al., 2011; Mojica and Brussaard, 2014).  Besides lytic infection, there is the potential for a lysogenic viral life cycle, during which viral DNA is integrated in the host as a prophage (Weinbauer, 2004). We found, however, no evidence that the share of lysogeny compared to the lytic cycle was affected. In fact, the percentage lysogeny was found insignificant during the entire campaign. Mean viral abundances were higher under $\text{CO}_2$ enrichment towards the end of phase I, which is expected to be in response to increased phytoplankton and prokaryote biomass.

.

During Phase I the high turnover of phytoplankton biomass led to increasing growth of heterotrophic prokaryotes (Hornick et al., 2016).  The enhanced net abundances (this study) were heavily grazed and additionally viral lysis became increasingly important (to 60% of total losses at the end of phase I).  Bermúdez et al. (2016) reported highest biomass of protozoans around t15. This was predominantly the heterotrophic choanoflagellate *Calliacantha natans* (Hornick et al., 2016). *Calliacantha natans* feeds selectively only on particles <1 µm in diameter (Marchant and Scott, 1993) and so could graze on heterotrophic bacteria. During the second half of phase I significantly more prokaryotes were recorded in the high $f\text{CO}_2$ mesocosms, which was likely due to increased availability of dissolved organic carbon at high $f$CO$_2$  higher rates of viral lysis of Pico II and Nano I initially (day 6) and Pico I and Nano II consecutively (day 10).

Assuming a cellular carbon conversion for phytoplankton cells of 237 fg C $\mu$m$^{-3}$ (Worden et al., 2004) and 196.5 fg C $\mu$m$^{-3}$ (Garrison et al., 2000) for pico- and nano-sized plankton, respectively, we calculated that viral lysis of phytoplankton  between days 9 and 13 resulted in the release of 1.1 and 12.4 ng C ml$^{-1}$ for M1 and M3, respectively. Similarly, assuming a bacterial growth efficiency of 30% and cellular carbon conversion of 7 fgC cell$^{-1}$ (Hornick et al., 2016), we estimated that the amount of organic carbon required to support bacterial growth during this period (taking into account the loss of bacterial carbon due to grazing and viral lysis) was 0.7 ngC ml$^{-1}$ in M1 and 11.0 ngC ml$^{-1}$ in M3. Viral lysis of phytoplankton was thus an important source of organic carbon for the bacterial community and may have led to the observed differences between treatments.

 total prokaryote  The significantly more abundant prokaryotes in the high $f$CO$_2$  during the second half of phase I  This  to increased availability of  organic carbon at high $f$CO$_2$  by higher rates of viral lysis of phytoplankton.  Pico II and Nano I initially (day 6) but consecutively also Pico I and Nano II (day 10) Pico I and II, Nano I and II at higher $f$CO$_2$ ~~The increased prokaryote standing stock did not sustain and was mainly grazed down to abundances comparable to the present-day $f$CO$_2$ mesocosms at day 16 Stimulation of prokaryote abundances was seen in a previous mesocosm campaign apparently due to higher availability of TEP and aminopeptidase activity (Endres et al., 2014). Increased TEP production is often associated with low nutrient, high $f$CO$_2$conditions (Weinbauer et al., 2011).~~

were higher under at higher $fCO_2$ levels during phase I, which is expected to be in  a response to increased

bacteriophages

lysogenic viral life cycle, during which viral DNA is integrated in found, however, no evidence that the share of

examined whether increased $fCO_2$ concentrations affect

**4.3 Phase II (days 17-30)**

Phase II displayed a second peak in total phytoplankton abundances related to increased picophytoplankton but reduced nanophytoplankton. Reduced microzooplankton grazing pressure on the picoeukaryotes and *Synechococcus* after day 17 allowed them to increase in  abundance during Phase II. Microzooplankton abundances were reduced as compared to the start of the experiment (approximately an order of magnitude lower) and mesozooplankton increased (Lischka et al., 2015). Thus increased grazing of mesozooplankton on microzooplankton may have resulted in reduced grazing of, and proliferation of, picophytoplankton.

*Synechococcus* bloomed during phase II,  with significantly  lower abundances at higher $f\text{CO}_2$. So although  Pico I benefitted from $\text{CO}_2$ enrichment, the similar sized *Synechococcus* did not. *Synechococcus* has shown diverse, strain-specific responses to $\text{CO}_2$ enrichment (Fu et al., 2007; Lu et al., 2006; Traving et al., 2014). As a prokaryote, *Synechococcus* has very different physiology from picoeukaryotes , needing extremely efficient CCMs due to the inefficiency of its Rubisco. Able to concentrate $\text{CO}_2$ to up to 1000-fold higher than the external medium (Badger and Andrews, 1982), they may attain maximal growth rates .  at the present-day  $\text{CO}_2$ concentration (Low-Décarie et al., 2014).

The prokaryote abundance increased steadily during Phase II, again matching total phytoplankton dynamics. Following the initially higher prokaryote abundances at higher $f\text{CO}_2$ in Phase I, we found during phase II  decreased abundances of HDNA-prokaryotes at high $f\text{CO}_2$.  This fits with the reported reduced bacterial production (Hornick et al., 2015) and respiration measurements (Spilling et al., 2015) in these mesocosms during this time. - The differences were due to an indirect effect on the prokaryotes of reduced phytoplankton growth by SYN, Pico III and Nano I leading to lower POC concentrations at higher $f\text{CO}_2$. This was caused by reduced temperature and PAR (Paul et al., 2015). Indeed we saw only low grazing rates for this period and no significant differences in loss by either grazing or lysis, or in DOC (Paul et al., 2015). The steady increase in viral abundances from day 22 onwards indicates that viral lysis of the prokaryotes was substantial, which is confirmed by the halting of prokaryote growth, reduced bacterial production (Hornick et al., 2016) and ultimate decline in prokaryote abundance (this study). The estimated average viral burst size during phase III, obtained from the increase in total viral abundance and concomitant decline in bacterial abundances, was about 30 which is comparable to published values (Parada et al, 2006; Wommack and Colwell, 2000). Viral lysis rates of prokaryotes were measured until day 25 and indicated that on average 10-15% of the total population lysed per day (day 18-25). The final prokaryote abundance at the end of the experiment was in line with a continued lysis in that order of magnitude (corrected for reduced bacterial production; Hornick et al.,

2016).  Overall, the increased prokaryote activity during the first half of phase II, the relatively low phytoplankton activity during this phase and the (virally induced) mortality of the prokaryote community during the second half of phase II promotes the mineralization and increase in concentration of phosphate (particularly in the low $f$CO$_2$ mesocosms; Paul et al, 2015).  To what extent elevated CO$_2$ concentration affects the reduction in P-release from biomass (Nausch et al., 2016), reduced respiration and bacterial production rates as seen in this study (Hornick et al., 2016; Spilling et al., 2016) needs to be explored still.

**4.4 Phase III (days 31-43)**

The positive growth response of the picoeukaryotes to earlier CO$_2$- enrichment was still clearly reflected in the Chlorophyll a concentration, particulate organic carbon and phosphorus, but and also in the dissolved organic carbon (DOC) pools in Phase III (Paul et al., 2015). This increase in DOC at high $f$CO$_2$ (Paul et al., 2015), may originate from viral lysis of prokaryotes and phytoplankton (Suttle 2005, LønborgLonborg et al., 2013).  We measured indeed higher viral lysis rates for SYN, Pico II and Nano I, and similar lysis rates but- higher standing stock of Pico I at high $f$CO$_2$ on day 31. Additionally after day 22 total viral abundances increased steadily until the end of the experiment. Alternatively, increased $f$CO$_2$ coupled with low nutrient availabilitys may also havecan stimulated photosynthetic release of DOC and subsequently transparent exopolymer particles (TEP) formation (Engel, 2002; Borchard and Engel, 2012).  TEP formation also results from sloppy feeding ( Hasegawa et al., 2001; Møller, 2007) and viral lysis, , mayand -is thought to promote aggregation and sinking of particulate organic matter (Brussaard et al., 2008; Lønborg et al., 2013). and Under the current conditions this would offset the reduced sedimentation associated with smaller cells (Sommer et al., 2002). However, no difference in sedimentation rates was reported -between $f$CO$_2$ treatments for the current study indicating that the change in phytoplankton community composition did not result in altered transport of POC (live or dead) (Paul et al., 2015).  Still, Tthis may have been (partly)

obscured by the negative correlation of diatoms, reported to have relatively higher sedimentation rates (Riebesell, 1989; Waite et al., 1997), with $f$CO$_2$ during phase III (Paul et al., 2015). At this stage it is hard to draw a final conclusion because at the same time there was a positive correlation with $f$CO$_2$ for larger-sized diatoms (>20 μm) (Paul et al., 2015). Because of the general urgency to know more about carbon sequestration, we recommend future studies on OA to focus not only on potential shifts in sedimentation due to changes in phytoplankton community composition, but also as a result of changes in phytoplankton size class in combination with the relative share of grazing and viral lysis (Brussaard et al., 2008).

Additionally, a significant negative correlation was reported between diatom abundance and $f$CO$_2$ during phase III (Paul et al., 2015), which is similar to a previous ocean acidification mesocosm experiment (Brussaard et al., 2013). Brussaard et al. (2013) suggested that diatom growth was reduced due to increased uptake of the growth limiting nutrients by the picoeukaryotes at high $f$CO$_2$ and may result in reduced sedimentation.

**5 Conclusions**

Firstly, our data explain the majority of the phytoplankton dynamics in this mesocosm experiment as more than 90% of the Chl $a$ was found in the <20 μm size fraction (Paul et al., 2015). Indeed these data allow us to examine the more detailed changes in community dynamics which are not obvious in the bulk measurements. Distinct shifts between more abundant pico-sized (0.2-3 μm) and nano-sized (3-20 μm) photoautotrophs were seen during the experiment which were also reflected in size-fractionated Chl a concentrations (Paul et al., 2015). Whilst other evident shifts in abundance and net growth rates between different picoeukaryote groups could only be revealed with the current approach of using flow cytometry. Moreover, the complementary grazing and lysis loss rates (along with the gross growth rates) allowed for a more notable explanation of changes in the phytoplankton and prokaryote community.

Secondly, oOur study shows that $CO_2$ enrichment favours the net growth of the very small-sized (1 µm) picoeukaryotic phytoplankton. This positive response with $fCO_2$ is very specific, as neither *Synechococcus* spp., Pico III, nor nor of the nanoeukaryotic phytoplankton groups displayed enhanced growth. Increasing atmosperic $fCO_2$ leads to a number of further global changes, e.g. increasing sea surface temperatures (SST) which in turn strengthens verticalincrease stratification and , shaooalllowsing the mixed layerreducing nutrient supply toin surface waters depth (Sarmiento et al., 1998; Toggweiler and Russell, 2008). } Such changes in physicochemical conditions have been reported to favouur small cells, largely because of reduced nutrient supply to the surface waters (Cermeño et al., 2008; Riebesell et al., 2009; Li et al., 2009; Craig et al., 2013; Li et al., 2009; Mojica et al., 20152016 a , b; Riebesell et al., 2009). The study by Mojica et al. (2016) shows that under such conditions the share of viral lysis vs grazing for a variety of phytopelankton groups increases, thereby promoting a more regenerative system. The additional increase in abundance of specifically the small picoeukaryotes in direct response to increased levels of CO₂ has been reported a few times earlier () and as such seems a general feature.

TAlso the overall activity of prokaryotes is expected to be affected not only by viral lysis of phytoplankton and prokaryotes themselves, butbut also by higher SST. This due to resultings in increased enzyme activities, bacterial production butand also, respiration rates, polysaccharide release and TEP formation ( Piontek et al., 2009; Wohlers et al., 2009; Borchard et al., 2011; Engel et al., 2011; Wohlers-Zöllner et al., 2011; Piontek et al., 2009; Wohlers et al., 2009; Wohlers-Zöllner et al., 2011). Enhanced bacterial re-mineralization of organic matter could further increase the autotrophic production by the small-sized phytoplankton ( Riebesell et al., 2009; Riebesell and Tortell, 2011; Engel et al., 2013); Riebesell and Tortell, 2011; Riebesell et al., 2009). At the same time,  viral lysis and  microbial respiration  the biological pump is negatively affected by the production of atmospheric $CO_2$ (del Giorgio and Duarte, 2002). the evidence presented in the current study  that  $CO_2$ enrichment  small-sized picoeukaryotic phytoplankton, which is further strengthened by increased SST and  vertical stratification. By and large these  will tend to reduce carbon sequestration .

**Author Contribution**

Design and overall coordination of research by CB. Organization and performance of analyses in the field by KC. Data analysis by KC and CB. Design and Coordination of the overall KOSMOS mesocosm project by UR. All authors contributed to the writing of the paper.

**Acknowledgements**

This project was funded through grants to C.B. by the Darwin project, the Netherlands Institute for Sea Research (NIOZ), and the EU project MESOAQUA (grant agreement number 228224). We thank the KOSMOS project organisers and team, in particular Andrea Ludwig, the staff of the Tvärminne Zoological Station and the diving team. We give special thanks to Anna Noordeloos, Kirsten Kooiman and Richard Doggen for their technical assistance during this campaign. We also gratefully acknowledge the captain and crew of R/V ALKOR for their work transporting, deploying and recovering the mesocosms. The collaborative mesocoms campaign was funded by BMBF projects BIOACID II (FKZ 03F06550) and SOPRAN Phase II (FKZ 03F0611).

Table 1. $f$CO$_2$ concentrations (µatm) as an average for the duration of the experiment following CO$_2$ addition and specification of this CO$_2$ level as low, medium or high. *denotes mesocosms sampled for grazing and viral lysis assays

| Mesocosm | M1* | M5 | M7 | M6 | M3* | M8 |
|---|---|---|---|---|---|---|
| CO$_2$ Level | LOW | LOW | INTERMEDIATE | INTERMEDIATE | HIGH | HIGH |
| Mean $f$CO$_2$ (µatm) days 1-43 | 365 | 368 | 497 | 821 | 1007 | 1231 |
| Symbol | | | | | | |

**Figure captions**

**Fig. 1. a)** Temporal dynamics of depth-integrated upper layer (0.3–10 m) total phytoplankton and **b)**

total eukaryotic phytoplankton, ie. all except the prokaryotic photoautotroph *Synechococcus* spp..

Lines indicate the start and end of phase II. The colours and symbols used in the legend are consistent throughout subsequent figures and, in parenthesis, is shown the mean $f$CO$_2$ across the duration of the experiment ie. days 1-43 .

**Fig. 2. a)** Temporal dynamics of depth-integrated upper layer (0.3–10 m) total prokaryotic phytoplankton, *Synechococcus* spp., whereby the lines indicate the different phases (I-III). **b)** Gross growth rates and total loss rates in mesocosms M1 and M3. Gross growth displayed as bars above the X- axis and total losses as bars below the X-axis. Significant differences between mesocosms are marked: p≤0.001\*\*\*, p≤0.01\*\*, p ≤0.05\*,p ≤0.1 · **c)** Abundances for mesocosm M1 (low $f$CO$_2$, blue line) and mesocosm M3 (high CO$_2$, red line). **d)** Specific growth rates derived from exponential regression of the net SYN abundances, versus average $f$CO$_2$ for days 4-7.

**Fig. 3. a)** Temporal dynamics of depth-integrated upper layer (0.3–10 m) picophytoplankton I (Pico I).

**b)** Gross growth rates and total loss rates in mesocosms M1 and M3. Gross growth displayed as bars above the X- axis and total losses as bars below the X-axis. Significant differences between mesocosms are marked: p≤0.001\*\*\*, p≤0.01\*\*, p ≤0.05\*,p ≤0.1 · **c)** Abundances for mesocosm M1

(low $f$CO$_2$, blue line) and mesocosm M3 (high CO$_2$, red line). **d)** Specific growth rates derived from exponential regression of the net Pico I abundances, versus average $f$CO$_2$ for days 1-5; **e)** days 5-

9; **f)** days 18–t21, a negative growth rate indicates cell loss. **g)** Phytoplankton cell abundance versus actual $f$CO$_2$ for Pico I on days 5; **h)** 13 **i)** 21.

**Fig. 4. a)** Temporal dynamics of depth-integrated upper layer (0.3–10 m) picoeukaryotic phytoplankton II (Pico II). **b)** Gross growth rates and total loss rates in mesocosms M1 and M3. Gross growth displayed as bars above the X- axis and total losses as bars below the X-axis. A rate of zero is displayed as a 0 in the colour of the mesocosm it relates to. Significant differences between mesocosms are marked: p≤0.001\*\*\*, p≤0.01\*\*, p ≤0.05\*,p ≤0.1 · **c)** Abundances for mesocosm M1

(control, blue line) and mesocosm M3 (high $CO_2$, red line). **d)** Specific growth rate determined from the net Pico II abundances, versus average $f$$CO_2$ for days 12-17. **e)** Phytoplankton cell abundance versus actual $f$$CO_2$ for Pico I on day 17.

**Fig. 5. a)** Temporal dynamics of depth-integrated upper layer (0.3–10 m) picoeukaryotic phytoplankton III (Pico III). **b)** Gross growth rates and total loss rates in mesocosms M1 and M3.

Gross growth displayed as bars above the X- axis and total losses as bars below the X-axis. No data indicates a failed experiment and a rate of zero as a 0 in the colour of the mesocosm it relates to.

Significant differences between mesocosms are marked: p≤0.001\*\*\*, p≤0.01\*\*, p ≤0.05\*,p ≤0.1 · **c)**

Abundances for mesocosm M1 (low $f$$CO_2$, blue line) and mesocosm M3 (high $CO_2$, red line). **d)**

Specific growth rate determined from the net Pico III abundances, versus average $f$$CO_2$ for days 1-2.

**e)** Phytoplankton cell abundance versus actual $f$$CO_2$ for Pico I on day 24.

**Fig. 6. a)** Temporal dynamics of depth-integrated upper layer (0.3–10 m) nanoeukaryotic phytoplankton I (Nano I). **b)** Gross growth rates and total loss rates in mesocosms M1 and M3. Gross growth displayed as bars above the X- axis and total losses as bars below the X-axis. No data indicates a failed experiment and a rate of zero as a 0 in the colour of the mesocosm it relates to. Significant differences between mesocosms are marked: p≤0.001***, p≤0.01**, p ≤0.05*,p ≤0.1 · **c)** Abundances for mesocosm M1 (low $f$CO$_2$, blue line) and mesocosm M3 (high CO$_2$, red line). **d)** Specific growth rate determined from the net Nano I abundances, versus average $f$CO$_2$ for days 10-12, a negative growth rate indicates cell loss  **e)** Phytoplankton cell abundance versus actual $f$CO$_2$ for Nano

I on day 17.

**Fig.7. a)** Temporal dynamics of depth-integrated upper layer (0.3–10 m) nanoeukaryotic phytoplankton II (Nano II).  **b)** Gross growth rates and total loss rates in mesocosms M1 and M3.

Gross growth displayed as bars above the X- axis and total losses as bars below the X-axis. No data indicates a failed experiment and a rate of zero as a 0 in the colour of the mesocosm it relates to.

Significant differences between mesocosms are marked: p≤0.001***, p≤0.01**, p ≤0.05*,p ≤0.1 · **c)**

Abundances for mesocosm M1 (low $f$CO$_2$, blue line) and mesocosm M3 (high CO$_2$, red line). **d)**

Specific growth rate determined from the net Nano II abundances, versus average $f$CO$_2$ for days 6-17

(M1, days 6-16) **e)** Phytoplankton cell abundance versus actual $f$CO$_2$ for Nano II on day 17 (M1, day

16).

**Fig.8.** POC calculated from mean cell abundances applying conversion factors of 237 fg C μm$^{-3}$

(Worden et al.2004) and 196.5 fg C μm$^{-3}$ (Garrison et al. 2000) for pico- and nano-sized plankton ,respectively , cellular carbon was calculated based on the average cell diameters. **a)** Temporal dynamics of Pico I and II **b)** Temporal dynamics of POC for all other  eukaryotic groups ie. Pico III, Nano I and II.

**Fig.9. a)** Temporal dynamics of depth-integrated upper layer (0.3–10 m) total heterotrophic prokaryotes (HP)  **b)** High DNA fluorescence heterotrophic prokaryotes (HDNA-HP)  **c)** Low DNA

fluorescence heterotrophic prokaryotes (LDNA-HP) .

**Fig.10. a)** M1 (low $f$CO$_2$) and M3 (high CO$_2$) temporal dynamics of total heterotrophic prokaryotes (HP) abundances   **b)** grazing rates (d$^{-1}$) (bars below the X-axis). Significant differences between mesocosms are marked: p≤0.001\*\*\*, p≤0.01\*\*, p ≤0.05\*,p ≤0.1· **c)** Viral lysis as percentage  of  HP

standing stock in mescocosm M1 (low $f$CO$_2$, blue) and M3 (high $f$CO$_2$, red) **d)** Total HP cell abundance versus actual $f$CO$_2$ on day 13. **e)** Mean prokaryote abundances in high (3,6,8) and low CO$_2$

mesocosms (1,5,7) vs total particulate organic carbon (POC) calculated from total cell abundances, ie.

all groups measured by flow cytometry, for both series R$^2$=0.7.

**Fig.11. a)** Temporal dynamics of depth-integrated upper layer (0.3–10 m) total virus abundances,  **b)**

Virus group V3, discriminated by its higher green nucleic acid-specific fluorescence.

[Figure]

[Figure]

[Figure]

[Figure]

[Figure]

**d**

$y = 6E\text{-}05x + 0.08$
$R^2 = 0.95$

**g**

$y = 13.829x + 90926$
$R^2 = 0.80$

**e**

$y = 7E\text{-}05x - 0.21$
$R^2 = 0.89$

**h**

$y = 35.963x + 31665$
$R^2 = 0.94$

**f**

$y = 0.0002x - 0.03$
$R^2 = 0.71$

**i**

$y = 35.7x + 43253$
$R^2 = 0.84$

net growth rate (d$^{-1}$)

Abundance (x10$^5$ ml$^{-1}$)

fCO$_2$ ($\mu$atm)

fCO$_2$ ($\mu$atm)

---

## Author Response (AR2)

We thank the reviewers for taking the time to review this manuscript and we have carefully considered and addressed their comments (in blue italic).

**Reviewer 1:**

As I also reviewed the original submission, I am omitting a summary of the manuscript here. Although the authors changed the title of their manuscript, the interpretation of the presented data hasn't changed much (or at least it doesn't come across to the reader). At the moment it is relatively hard to judge the presented manuscript/data as there are still many cases where the statements/conclusions/claims by the authors are not supported by the results. Unfortunately, the authors do not present any further analyses or statistics to back up their statements (I have given a few examples below; however, this list is not comprehensive!).

We have added new statistical analyses including a non-metric multidimensional scaling (NMDS) plot of autotrophic eukaryote and prokaryote abundances which shows the development of the community in time. This shows that there is a "divergence" between the community development of the low and high fCO2 mesocosms. An analysis of dissimilarities (ANOSIM) shows that this is significant (p=0.01)(Lines 307-308 of revised manuscript and Fig.2). The NMDS also allowed us to identify the periods during which divergence occurred. Linear regressions of net growth rates against fCO2 (averaged over the period) for the individual groups allowed us to further examine the differences (Lines 308-312; Table 2, Figs. S2 and S3). Mesocosms M1 and M3 are clearly in two separate clusters. (Lines 314-315; Fig. 2). We have also carefully examined all claims made and tried to ensure that all are adequately statistically tested and that these test results are presented in the text.

An overall suggestion to the authors would be to look for someone who would be able and willing to aid with the manuscript re-structuring and re-writing, but who is himself or herself not involved in these experiments (someone in their departments maybe? I think it would help to get an outside-the-box opinion here for the interpretation of the data). At the current state, there seem to be too many misinterpretations in the manuscript in order to recommend publication (even with some revisions). I have made some more specific comments below.

We asked two colleagues to contribute (statistical analysis and interpretation and writing); both are now included as co-authors.

In general, the authors try to make (big) claims about the effect of CO2 on the microbial community; however, basically provide neither analyses/statistics that support these claims nor is it visible in the figures.

We have added further statistical analyses as described above. We have also taken care not to overstate the differences we see.

Further, the authors have not given any type of error on their measurements making it hard to see actual differences.

A series of mesocosms with incremental steps of  $fCO_2$  addition was chosen as the experimental design which minimises the risk of failure due to losing mesocosms. This was favoured over a replicated approach (Lines 129-133). Statistical analyses are therefore regressions and do not have errors attached. Unfortunately we were unable to perform grazing and lysis experiments on replicate mesocosms due to the logistics of sample preparation and analysis on site. We have added these considerations into the text (Lines 221-222; 253-256 of revised manuscript).

Overall, in many places it seems that the authors see what they want to see (i.e. an effect of CO2 concentrations on the microbial community).

We have added further statistical analyses to support all claims made on  $fCO_2$ -induced differences.

Again, I would recommend to add an overview of the experimental setup to the methods section. I understand that the experimental setup is described in a different manuscript for the special issue. However, as a reader of this manuscript, I actually do not want to have to read another manuscript first in order to find out what the authors did here; at least a summary would be needed here. *We added a section describing the experimental set-up (Lines 110-125 of revised manuscript)*.

Many places in the manuscript are still not written to the extent that they can be easily followed, for example: "The higher loss rates (days 5 to 9; Fig. 3e) resulted in a decrease in abundance, which was stronger for the low fCO2 mesocosms (as illustrated by M1) due to the significantly higher gross growth rates for the high fCO2 mesocosm (represented by M3; Fig. 3b). The positive correlation of Pico I peak abundance with fCO2 on day 13 (Fig. 3h, R2=0.94) was lost upon another decline in abundance. Significantly higher losses at high fCO2, a combination of grazing and lysis, resulted in a more dramatic crash at high fCO2 and abundances becoming similar again around day 17 (Fig. 3a)." This paragraph is simply not understandable for someone not working on these experiments without having to read the sentences multiple times (which disrupts the flow reading dramatically).

We have improved the readability of the entire manuscript and it has been reviewed and edited by two additional co-authors.

The figures were not labeled at the end of the manuscript (no matter whether this is because of forgetting to label them or whether the submission software omitted to do so); so, for the purpose of the review I assumed that the figures were 1-11 in the way they came out of the printer and I am referring to them in that way.

We apologise for the inconvenience this may have caused and made certain that all are clearly labelled in the revised manuscript.

In general, I think that the authors should try to improve the readability of their manuscript by increasing the quality of their figures, e.g., not having figures that span nearly two full pages (see Figure 3) or omitting repetitive data.

We have completely re-organised the figures to improve their appearance and readability. We have taken note of the reviewer's comments and tried to ensure that they are well presented and have omitted repetitive data.

Showing the ambient data together with the mesocosm is essential to judge whether changes in the mesocosms were, for example, induced by enclosement or were naturally happening outside as well. This does not mean that, in case changes were induced by enclosement, that differences between mesocosms are not useful; they are valuable data. Not presenting the ambient microbial community leaves the reader with guessing what happened and potentially suspecting that the data was omitted because they would alter the conclusions of the paper. If the authors would like to increase the understandability of their manuscript, I highly recommend adding the ambient data to the current main manuscript figures. For example, Suppl. Figure 1 shows that changes in the Synechococcus population are pretty much the result of 'bottle effects' (I agree that a mesocosm is a large bottle, but it is an enclosement). Is this possibly related to cleaning procedures or stirring of the mesocosms as outlined in Paul et al.? While other groups are coherent with the ambient water.

We acknowledge the reviewer's concerns and have added these data into the main figures and Results sections generally (Lines 316-323) and of each group (Figs. 1,3,6 and Figs. S1, 6,7). They are shown as black lines in accordance with the other manuscripts in the special issue. We originally excluded microbial data from the outside water because of the dynamic water movement (including periods of upwelling) in the surrounding waters of this region, which complicates comparisons between them. We now also discuss microbial dynamics in relation to outside water (Lines 490-520 of revised manuscript).

In response to the example given by the reviewer: the mesocosms were kept open for 5 days (t-10 to t-5) for rinsing and free exchange of <3mm plankton. The bags were then closed and bubbled for 3.5 min with compressed air (t-5) to ensure a homogenous water column. Five more days were then allowed before  $CO_2$  manipulation (t-5-t0). This time line suggests that the bubbling could have stimulated Synechococcus growth. Alternatively the surrounding waters, which are very dynamic, may have altered resulting in lower abundances outside the mesocosms. We have acknowledged this in the Results: "Phytoplankton abundances in the surrounding water started to differ from the mesocosms during Phase 0 (on average 44 % lower) which was primarily due to lower abundances of SYN. This effect was seen from day -1, prior to  $CO_2$  addition but following bubbling with compressed air (day -5)." (Lines 316-318) and Discussion: "During Phase 0, the microbial assemblage showed good replicability between all mesocosms, however they had already began to deviate from the community in the surrounding waters. This was most likely a consequence of water movement altering the physical conditions and biological composition of the surrounding water body. The dynamic nature of water movement in this region has been shown to alter the entire phytoplankton community several times over within a few months, due to fluctuations in nutrient supply, advection, replacement/mixing of water masses and water temperature (Lips and Lips, 2010). Alternatively, effects of enclosure and the techniques (bubbling) used to ensure a homogenous water column may have stimulated SYN within the mesocosms." (Lines 491-499).

After looking into Paul et al. for the experimental setup, it seemed that the CO2 concentrations were actually measured during the mesocosms, why aren't the actual measured concentrations used for any of the plots instead of the targeted concentrations (which were only achieved in the first few days but weren't maintained)? This seems such an obvious thing to do and a fact that is completely ignored by the authors.

Actually, we did work with the measured values. Table 1 lists the average  $fCO_2$  for each mesocosm over the duration of the experiment. Linear regressions of abundance or growth rate are plotted against actual  $fCO_2$  for the day or period analysed in the specific mesocosms. We have better clarified this in the text and Figure captions.

Abstract: "Of these groups 2 picoeukaryotic groups increased in abundance whilst the other groups, including prokaryotic Synechococcus spp., decreased with increasing fCO2." Looking at both Figure 2a and Suppl. Figure 1a, I seriously do not see where this statement comes from! There is a short period of time (~ 7-16 days) where Syn. are lower in the low fCO2 than in the higher ones but for most of the experiment it is the other way around. Unfortunately, the authors also do not supply any other analysis or statistics that could possibly back up their claim. Unfortunately, these kind of statements and interpretation is occurring throughout the manuscript, which leads me to the suggestion that the current manuscript is far from being acceptable for publication at this time.

We believe that the reviewer has misread the sentence. We were indeed stating, as does the reviewer, that Synechococcus abundance generally decreased at higher  $fCO_2$ . Peak abundance regression against  $fCO_2$  was provided to support this. We now also provide significant linear regression of net growth rates against  $fCO_2$  (Table 2 Fig. S3a) to support this and have added further analyses in the Results section "After day 16, SYN abundances increased in all mesocosms and during this period (days 16-24) net growth rates had a significant negative correlation to  $fCO_2$  (p=0.05,  $R^2 = 0.63$ ; Figs. 3a, Table 2 and Fig. S3a). Consequently, the net increase in SYN abundances during this period was on average 20 % higher at low compared to high  $fCO_2$ ."(Lines 333-337 of revised manuscript). (We have added similar data to the results of each of the phytoplankton groups to support our claims).

Figure 1: The authors claim that the decline in total phytoplankton around the end of phase 2 and throughout phase 3 is due to a decrease in Synechococcus. This is supported by Figure 2a. However, Suppl Figure 1a shows Synechococcus populations staying up rather than declining to concentrations in Figure 1. After doing some research through this manuscript, I found that the difference between the figures is the depth reference here, which is 0.3-10 m for Figure 1 and 0-17 m for Suppl. Figure 1. This difference actually suggests that either the distribution of Synechococcus is not equal throughout the water column or that there is a sinking out of Synechococcus? I don't know why this is, maybe the authors should think about this deviation which is currently not acknowledged anywhere.

Indeed in the top 10m samples Synechococcus accounts for >90% of the loss in total phytoplankton abundance for days 24-28. In the 0-17 m samples they account for 60%. The distribution of Synechococcus is not equal throughout the water column; as a result of vertical stratification phytoplankton abundances were higher in the surface waters. There is no noticeable sinking of Synechococcus, otherwise they would be higher in the 0-17 m samples.

Figure 1: The authors say that the total phytoplankton abundance at the end of phase 1 is significantly higher in the high fCO2 treatment than the low fCO2 treatments. Again, I am really sorry, but I cannot

see this in the plots and the authors again don't provide any analysis or statistics for this. (see p 10 lines 18-20).

By this we meant the second half of Phase I. We have re-written this section to clarify and now show additional statistical analyses to support our claims (Lines 308-311 of revised manuscript).

Panel d: What are the errors on these measurements?

They are the specific growth rates for individual mesocosms for a specific time period so there is no error.

I wonder whether these differences are actual changes or just variation ?

During re-writing we decided to omit this panel. All the linear regressions of net growth against  $fCO_2$ , as averaged for the specific time period for each mesocosm, now have p-values to test for significance.

Further, this is comparing growth rates from time point 10 days to growth and loss rates of time points 4-7. In my opinion, this is totally misleading as day 10 is obviously different than 4-7 as can be seen in panel a. (see p 11, 110-13).

This section has been re-written such that we state more clearly that we are examining differences between the mesocosms from days 3-13 "Abundances of SYN showed distinct variability between the different  $CO_2$  treatments, starting on day 7, with the low  $CO_2$  mesocosms exhibiting nearly 20 % lower abundances between days 11-15 as compared to high f $CO_2$  mesocosms (Fig. 3a). SYN net growth rates during days 3-13 (NMDS-based period 1) were positively correlated with  $CO_2$  (p=0.10,  $R^2$ =0.53; Table 2, Fig. S2a). One explanation for higher net growth rates at elevated  $CO_2$  could be the significantly (p<0.05) higher grazing rate in the low f $CO_2$  mesocosm M1 (0.56 d-1) compared to the high f $CO_2$  M3 (0.27 d-1) as measured on day 10 (Fig. 4a). "(Lines 327-333 of revised manuscript). The significantly higher loss rates on day 10 may therefore serve to explain the lower abundances at lower f $CO_2$  during that time.

Figure 2: Panel c is absolutely unnecessary here (and in following figures); it is the same data as in panel a and its addition here suggests that it is indeed different data and is making the figure unnecessarily large.

We have completely reorganised the figures to improve readability and presentation, this panel has been removed.

Figure 2a legend: It is not necessary to present 4 (!) different significance levels here. For most claims in the manuscript no analyses or statistics are provided while here it is over the top. Either s.th. is significantly different or not.

*Agreed, we now only present <0.05 significance.*

p 10, 1 21 and following: If CO2 has a strong positive effect on Pico 1 and 2, how can they then be comparable in abundances to the surrounding water? Maybe I don't understand this sentence.

*Pico I and II are clearly different in abundance from the surrounding waters, we have removed this during re-writing and apologise if this was not clear.*

Suppl. Figure 1: This is (probably) the most informative figure of the entire manuscript. *We have added the surrounding water abundances to the main figures to make them more informative.*

p 11, 13-5: This claim is not supported by any analysis. Any time series statistics, PCA, similarity analysis, community composition comparisons ?

We have removed this particular sentence, have added new statistical analyses and carefully checked that all claims are fully backed up.

p 17, 1 17-18: this is an odd comparison in size fractions and the results are not exclusive of each other.

We removed this sentence as the statement is also made in the sentence before.

p 18, 1 20-22: There is consistency among the mesocosms maybe, but there is already a deviation to the surrounding water indicating an effect of enclosement, at least on some groups.

We have added comparisons between mesocosms and surrounding waters to the Results and Discussion sections (Lines 316-321 and 490-520, respectively).

Figure S3 is missing! –a) pico iii poc b)Nano I and II POC c) total POC *We apologise for the inconvenience of this omission; it has been rectified.*

**Reviewer 3**

This study investigates the effect of different fCO2 levels on different microbial groups as determined by flow cytometry in mesococm experiments in the Gulf of Finland. Clear and differential effects of fCO2 levels were found for the various microbial groups. In two mesocosms (low and high fCO2), viral infection and grazing were assessed, parameters that have not been very often measured in ocean acidification experiments.

The experimental approach and the methods are appropriate and appropriately described. Useful information on the potential effect of ocean acidification on different microbial groups was obtained.

The differences between M1 and M3 (growth rates, viral infection and grazing) are a bit overemphasized. Differences between two mesocosms can certainly be calculated and this might be ok in some experimental approaches. But in this case samples have to be taken repeatedly within one mesocosm to estimate the entire variability of the approach. This was certainly not done for the viral infection and grazing measurements. Also, the authors use this analysis to make a comparison between two treatments, but this can also be done when the experimental treatments are replicated. I appreciate the effort that the authors made, but they put to much weight on this analysis. They are right not to mention the effect in the abstract and should reduce the reference to these findings in the discussion. It is ok to discuss that briefly as potential effect, however, in the absence of statistical prove, this part short be less prominent.

We appreciate reviewer's appreciation of the research. We have further reduced emphasis on the loss assay results. It is indeed unfortunate that we were not able to replicate the experiments on more mesocosms but it was logistically / practically not feasible. The experiments were carried out in triplicate within mesocosms and the results are tested statistically but we understand the reviewers concerns that there may be differences between two mesocosms rather than treatments. The new non-metric multidimensional scaling (NMDS) plot does show that M1 and M3 do diverge from each other and cluster well with the other low and high fCO2 mesocosms respectively.

Despite the comments of one reviewer, the presentation -especially in the discussion- remains poor. Many sentences are awkward or hard to understand.

We have rewritten and improved readability of the manuscript and it has been read and edited by two additional co-authors.

Also, the arguments are sometimes presented in a sloppy way. Although I think I understand what the authors want to say and have no serious objection, the argumentation has to be improved and presented with more rigor.

We have added and improved our statistical analyses which has allowed us to be more specific and has strengthened our argumentation. We have added new statistical analyses including a non-metric multidimensional scaling (NMDS) plot of autotrophic eukaryote and prokaryote abundances which shows the development of the community in time. This shows that there is a "divergence" between the community development of the low and high  $fCO_2$  mesocosms. An analysis of dissimilarities (ANOSIM) shows that this is significant (p=0.01)(Lines 307-308 of revised manuscript and Fig.2). The NMDS also allowed us to identify the periods during which divergence occurred. Linear regressions of net growth rates against  $fCO_2$  (averaged over the period) for the individual groups allowed us to further examine the differences (Lines 308-312; Table 2,Figs. S2 and 3). Mesocosms *M1* and *M3* are clearly in two separate clusters. (Lines 314-315; Fig. 2). We have also carefully examined all claims made and tried to ensure that all are adequately statistically tested and that these test results are presented in the text.

**Minor comments:**

Page 2, Line 5/6: Does this mean that TOTAL algal biomass is related to prokaryotic biomass, despote of differential effects of fCO2 on algal groups? Please specify.

We have added two new Figures, one showing total algal biomass which closely resembles the Figure showing total prokaryote abundances (Fig. S7c) and another which shows a positive correlation between total algal biomass and prokaryote abundance (Fig. 8). We have also altered this sentence to: " Dynamics of the prokaryote community closely followed trends in total algal biomass despite differential effects of  $fCO_2$  on algal groups." (Lines 39-40 of revised manuscript).

Page 3, Lines 21-23: specify the direction of the responses.

Sentence changed to: "Our data show, that over the 43 day long experiment, enhanced  $CO_2$  concentrations elicited distinct shifts in the microbial community, most notably an increase in the net growth of small picoeukaryotic phytoplankton." (Lines 104-106 of revised manuscript).

Page 10, Results: I think there should be a short part on phase 0. We have added this at the beginning of the Results section (Lines 300-306 of revised manuscript).

Page 11, Line 10: 'most so'. generally? *This section has been rewritten more specifically (Lines 327-330).*

**Line 19: Are low and high fCO2 mesocosms nowhere defined.**

Initially as high, intermediate and low in Table 1. and in the Materials & Methods section. We now also added an extra comment at the end of the statistics section where we refer to only high and low mesocosms (Lines 294-296).

Line 22: What does 'not so clear' mean? Please use a more rigorous (scientific language). This comment holds also for many other occasions; not to all will be referred to. *We have extensively rewritten the manuscript and added statistical analyses to overcome this issue.*

Page 12, Line 2-4: I.e., 26% of total phytoplankton abundances were found on average in the high high fCO2 mescosms?

Sentence rewritten: "Pico-I was the numerically dominant group of eukaryotic phytoplankton, accounting for an average 21-26 % of total phytoplankton abundances." (Lines 348-349)

Please use more verbs in your sentences, this would increase the readability of the text. This holds for the entire text.

The manuscript has been rewritten and edited by two additional co-authors.

Line 22: What exactly do you mean by 'matched'. *We have rewritten this as 'comparable to total loss rates' (Line 371-372 of revised manuscript).*

Page 13, Lines 6-7: These? Please explain better what you mean. *We have removed the sentence*

Line 8: explain better why the word seems is appropriate here. We have rewritten this section to make it more rigorous (Lines 364-379).

Line 17 remove just Line 22: remove indeed Line 24: remove perfectly Line 25: remove indeed We have made the changes accordingly.

Page 14 Line 2: largely - be more precise *This sentence has been removed*

Line 3 after 'reasons', add: see materials and methods *This has been added (Lines 399-400 of revised manuscript)..*

Lines 4-7: Awkward sentence Sentence removed during rewriting

Line 23: Dynamics is I think a word that requires the singular. *Corrected.*

Line 23-25: No idea what the authors mean.

Sentence rewritten as: "The temporal dynamics of Nano-II, the least abundant phytoplankton group analysed in our study, displayed the largest variability (Fig. 3f), perhaps due to the spread of this cluster in flow cytographs (which may indicate that this group represents several different phytoplankton species)." (Lines 422-424).

Line 7: change into to to

Sentence has been altered to "The mean combined biomass of Pico-I and Pico-II showed a strong positive correlation with  $fCO_2$  throughout the experiment (p<0.05,  $R^2=0.95$ ; Fig. 5a), an effect already noticeable by day 2." (Lines 432-433)

**Line 2-3: A correlation could be calculated**

A generalized linear model was used to test the relationship between prokaryote abundance and carbon biomass with an ARMA correlation structure of order 3 to account for temporal autocorrelation. This has been included as Fig. 8 and in the Results section "Prokaryote abundance in the mesocosms was positively related to total algal biomass independent of treatment (p<0.05,  $R^2$ =0.33; Fig. 8) and generally followed total algal biomass (Fig. S7c)" (Lines 445-446).

Page 16, Lines 2-3: give test and significance level *This sentence was removed however all results are now supported by significance levels.*

Page 16/17; There was only one mesocosm for low fCO2 level, right? *Yes, we have now clarified this (Lines 459 of revised manuscript).*

Discussion

1st paragraph: There is no real explanation given. Only net growth and loss rates are observed. What has been actually 'examined'?

The Discussion has been completely re-written to improve readability.

2nd paragraph: Why would only nano-sized and not other phytoplankton profit from the nutrient upwelling? Also, the argumentation is not very clear.

We have added the following text: "A relaxation from nutrient limitation in vertically stratified waters disproportionately favours larger-sized phytoplankton, due to their higher nutrient requirements and lower capacity to compete at low concentrations dictated by their lower surface to volume ratio (Raven, 1998; Veldhuis et al., 2005)" (Lines 505-508)

1st paragraph: 'difficult' is sufficient. In addition, the authors could just refer to the pigment analysis. The molecular analyses were not done, so there is no need to refer to them.

We have re-written this section and refer only to the pigment analyses (Lines 572-573 of revised manuscript).

Again, no explanation is given, just rates are listed. The reader has to try to figure out what the authors have calculated. This has to be explained better.

We have added explanations and tried to improve the clarity and rigor of the arguments throughout the Discussion.

2nd paragraph: what are 'relatively' high loss? How can, based on net changes of abundances and some loss rates estimates, turnover of organic matter be calculateds. *This has been removed during re-writing*.

Following pages

Same problems as with other parts of the discussion. Please spend more time on a better and especially more rigorous explanation and argumentation! It is necessary that one can follow your arguments in the text!

Again, we have improved and thoroughly re-written the Discussion.

Figures: The legend starts should start with a caption that specifies the intention of the graph. *We have improved the Figure legends*

| 1  | Shifts in the size structure of the Alterations in microbial community in the Baltic Sea-composition                                                  |                        | Style Definition: Normal: Font color:                                                                        |
|----|-------------------------------------------------------------------------------------------------------------------------------------------------------|------------------------|--------------------------------------------------------------------------------------------------------------|
|    |                                                                                                                                                       | $\mathbf{X}$           | Custom Color(RGB(0,0,10))                                                                                    |
| 2  | with increasing fCO 2 : a mesocosm study in the eastern Baltic Sea                                                                         | $\left  \right\rangle$ | Style Definition:
apple-converted-space                                                                   |
| 3  |                                                                                                                                                       |                        | Style Definition: Comment Reference                                                                          |
| 4  | K.                                                                                                                                                    |                        | Style Definition: pb_toc_link                                                                                |
|    |                                                                                                                                                       |                        | Style Definition: Line Number                                                                                |
| 5  | Katharine J. Crawfurd 1 , U. Riebesell 2 , C.Santiago Alvarez-Fernandez 2 , Kristina D. A. Mojica 3 , Ulf |                        | Style Definition: author                                                                                     |
|    |                                                                                                                                                       |                        | Style Definition: pubyear                                                                                    |
| 6  | Riebesell 4 , Corina P. D. Brussaard 1,45                                                                                       |                        | Style Definition: articletitle                                                                               |
| -  |                                                                                                                                                       |                        | Style Definition: journaltitle                                                                               |
| /  |                                                                                                                                                       |                        | Style Definition: vol                                                                                        |
| 8  | [1]{NIO7 Royal Netherlands Institute for Sea Research, Department of Marine Microbiology and                                                          |                        | Style Definition: slug-pub-date                                                                              |
| 0  | [1][NO2 Noyal Nethenands institute for sea Research, Department of Marine Microbiology and                                                            |                        | Style Definition: slug-vol                                                                                   |
| 9  | Biogeochemistry, and Utrecht University, P.O. Box 59, 1790 AB Den Burg, Texel, The Netherlands}                                                       |                        | Style Definition: slug-issue                                                                                 |
|    |                                                                                                                                                       |                        | Style Definition: slug-pages                                                                                 |
| 10 | •                                                                                                                                                     |                        | Style Definition: Balloon Text: Font color: Custom Color(RGB(0,0,10))                                        |
| 11 | {2[2]{Alfred-Wegener-Institut Helmholtz-Zentrum für Polar- und Meeresforschung, Biologische                                                           |                        | Style Definition: List Paragraph: Font color: Custom Color(RGB(0,0,10))                               |
| 12 | Anstalt Helgoland, 27498, Helgoland, Germany}                                                                                                         |                        | Style Definition: Comment Text:
Font color: Custom Color(RGB(0,0,10))                                     |
| 13 |                                                                                                                                                       |                        | Style Definition: Comment Subject:
Font color: Custom Color(RGB(0,0,10))                                  |
| 14 | [3] {Department of Botany and Plant Pathology, Cordley Hall 2082, Oregon State University, Corvallis,                                                 |                        | Style Definition: Normal (Web): Font color: Custom Color(RGB(0,0,10)), Space Before: Auto Affer: Auto |
| 15 | Oregon 97331-29052, USA}                                                                                                                       |                        | Style Definition: Header: Font color:                                                                        |
| 16 |                                                                                                                                                       |                        | line numbers                                                                                                 |
| 17 | [4]{GEOMAR Helmholtz Centre for Ocean Research Kiel, Biological Oceanography, Düsternbrooker                                                          |                        | Custom Color(RGB(0,0,10)), Suppress
line numbers                                                          |
| 18 | Weg 20, 24105, Kiel, Germany}                                                                                                                         |                        | Style Definition: Hyperlink: Font color: Blue                                                                |
| 10 |                                                                                                                                                       |                        | Formatted: Justified                                                                                         |
| 19 | <del>[3</del>                                                                                                                                         |                        | Formatted: Header distance from edge: 0", Numbering: Continuous                                       |
| 20 | [5] {Aquatic Microbiology, Institute for Biodiversity and Ecosystem Dynamics, University of                                                           |                        | Formatted: Justified                                                                                         |
| 21 | Amsterdam, P.O. Box 94248, 1090 GE Amsterdam, The Netherlands}                                                                                        |                        | Formatted: Font: Not Bold, Not
Superscript/ Subscript                                                     |
|    |                                                                                                                                                       | $\langle   \rangle$    | Formatted: English (U.K.)                                                                                    |
| 22 | ــــــــــــــــــــــــــــــــــــ                                                                                                           |                        | Formatted: Justified                                                                                         |
| 22 | Correspondence to: K Crowfurd (kate crowfurd@gmail.com) and C D D D                                                                                   |                        | Formatted: Justified                                                                                         |
| 23 | correspondence to. K. Crawiuru ( kate.crawiuru@gmaii.com ) and C. P. D. Brussaaru                                                       |                        | Formatted: English (U.S.)                                                                                    |
| 24 | (corina.brussaard@nioz.nl)                                                                                                                            |                        | Formatted: Font: Italic                                                                                      |
| 25 | •                                                                                                                                                     | _                      | Formatted: Font: Bold                                                                                        |
|    |                                                                                                                                                       |                        | Formatted: Justified                                                                                         |
| 26 | 1 1                                                                                                                                            |                        |                                                                                                              |

| 27 |                                                                                                                                |                           |
|----|--------------------------------------------------------------------------------------------------------------------------------|---------------------------|
| 28 | Abstract-                                                                                                                      | Formatted: Justified      |
| 29 | Ocean acidification, due to resulting from the dissolution of anthropogenically                                                | Formatted: Font: Not Bold |
| 30 | producedanthropogenic carbon dioxide in the atmosphere, is considered a major threat to marine                                 |                           |
| 31 | ecosystems. WeHere we examined the effects of ocean acidification on the microbial community                                   |                           |
| 32 | structuredynamics in the Gulf of Finland, eastern Baltic Sea, during the, summer of 2012 when                                  |                           |
| 33 | inorganic nitrogen and phosphorus were highly depleted, summer. Using large. Large volume in situ                              | Formatted: Font: Not Ital |
| 34 | mesocosms were employed to simulatemimic present to future and far future CO2_scenarios, we                                    |                           |
| 35 | observed distinct trends with increasing $fCO_2$ in each of the 6. All six groups of phytoplankton                             |                           |
| 36 | enumerated by flow cytometry (<20 $\mu$ m cell diameter). Of these groups 2) showed distinct trends in                         |                           |
| 37 | net growth and abundance with CO 2 enrichment. The picoeukaryotic groups increased in abundance                     |                           |
| 38 | phytoplankton groups Pico-I and II displayed enhanced abundances, whilst the other groups,                                     |                           |
| 39 | including prokaryoticPico-III, Synechococcus spp., decreased with increasingand the nanoeukaryotic                             |                           |
| 40 | phytoplankton groups were negatively affected by elevated fCO 2 . Gross growth rates                                |                           |
| 41 | increasedSpecifically, the numerically dominant eukaryote, Pico-I, demonstrated increases in gross                             |                           |
| 42 | growth rate with increasing $fCO_2$ in the dominant picoeukaryote group sufficient to double their                             |                           |
| 43 | abundances whilst reduced losses allowed the other picoeukaryotes to flourish at higher fCO2.                                  |                           |
| 44 | Converting abundances to particulate organic carbon we saw a large shift in the partitioning of                                |                           |
| 45 | carbon between the size fractions which lasted throughout the experiment. The prokaryotes largely.                             |                           |
| 46 | Dynamics of the prokaryote community closely followed thetrends in total algal biomass with                                    |                           |
| 47 | responses to increasing fCO 2 reflecting the altered phytoplankton community dynamics.despite                       |                           |
| 48 | differential effects of fCO 2 on algal groups. Similarly, higher-viral abundances at higher fCO 2 seemed |                           |

[revised manuscript text omitted]
 Gulf                                                                                                                                                                                                                                                                                                                                                                                                                                                                                                                                                                                                                                                                                                                                |                                                               |
| 139                                           | of FinlandBaltic Sea near Tvärminne, with salinity around 5.7 and low dissolved inorganic nitrogen                                                                                                                                                                                                                                                                                                                                                                                                                                                                                                                                                                                                                                                                                                                                 |                                                               |
| 140                                           | and phosphorus concentrations. During 
[revised manuscript text omitted]
 <math>fCO_2</math> was reached. Initial fugacity of <math>CO_2</math> (<math>fCO_2</math>) was 240 µatm. For <math>fCO_2</math> |                           |
| 189 | manipulations, 50 $\mu$ m filtered natural seawater was saturated with CO 2 and then injected evenly                                           |                           |
| 190 | throughout the whole depth of the mesocosms as described by Riebesell et al. (2013). Two                                                                  |                           |
| 191 | mesocosms functioned as controls and were treated in four steps between days 0 to 3 until target                                                          |                           |
| 192 | fCO2_was reacheda similar manner using only filtered seawater. On day 15, a furthersupplementary                                                          |                           |
| 193 | $fCO_2$ addition was made to the top 7 m of mesocosms numbered 3, 6, and 8 to replace CO 2 lost due                                     |                           |
| 194 | to outgassing. The remaining mesocosms received similar treatment without CO 2 . (Paul et al., 2015;                                           |                           |
| 195 | Spilling et al., 2016). Throughout this study we refer to fCO 2 which accounts for the non-ideal                                               |                           |
| 196 | behavior of CO 2 gas and is considered the standard measurement required for gas exchange (Pfeil et                                            |                           |
| 197 | al., 2012).                                                                                                                                        |                           |
| 198 | Initial nutrient concentrations <del>, i.e. nitrate, phosphate, silicate and ammonium, _</del> were 0.05 µmol L -1 ,◄                          | Formatted: Justified      |
| 199 | 0.15 $\mu$ mol L -1 , 6.2 $\mu$ mol L -1 and 0.2 $\mu$ mol L -1 for nitrate, phosphate, silicate and ammonium,           |                           |
| 200 | respectively, and stayed. Nutrient concentrations remained low for the duration of the experiment                                                         |                           |
| 201 | (Paul et al., 2015, this issue)) and no nutrients were added. Salinity was relatively constant around                                                     | Formatted: English (U.K.) |
| 202 | 5.7 <del>,. Temperature was more variable; on average</del> temperature <del>was initially ≈8°C and rose to</del>                                         |                           |
| 203 | <del>≈15°C</del> within the mesocosms (0-17 m) increased from ~8 °C to a maximum on day 15 before falling to                                              |                           |
| 204 | $\approx$ 8°C of ~15 °C and then decreased again. to ~8 °C by day 30. For further details of the experimental                                             |                           |
|     |                                                                                                                                                           |                           |

<mark>1</mark>8

ĺ

| 205 | set-up, carbonate chemistry dynamics and nutrient concentrations throughout the experiment we                     |                             |
|-----|-------------------------------------------------------------------------------------------------------------------|-----------------------------|
| 206 | refer to the general overview paper by Paul et al. (2015).                                                        |                             |
| 207 |                                                                                                                   |                             |
|     |                                                                                                                   |                             |
| 208 | Collective sampling was performed daily in the every morning, using and epth integrated water-                    | Formatted: Justified        |
| 209 | sampler, from the top (0 10 m) and from the whole water column (0 17 m) of samplers (IWS, HYDRO-                  |                             |
| 210 | BIOS, Kiel). These sampling devices were gently lowered through the water column collecting ~5 L of               |                             |
| 211 | water gradually between 0-10 m (top) or 0-17 m (whole water column). Water was collected from all                 |                             |
| 212 | mesocosms and the surrounding water.— Subsamples were obtained for enumeration of                                 |                             |
| 213 | phytoplankton, prokaryotes and viruses. Samples for viral lysis and grazing experiments were taken                |                             |
| 214 | from 5 m depth using a gentle vacuum-driven pump system. Samples were protected against                           |                             |
| 215 | daylightsunlight and warming by thick black plastic bags containing wet ice. In the laboratory the                |                             |
| 216 | samplesSamples were processed at in situ temperature and dimmed(representative of 5 m depth)                      | Formatted: Font: Not Italic |
| 217 | under dim light and handled using nitrile gloves. As viral lysis and grazing rates were determined                |                             |
| 218 | from samples taken from 5 m depth, samples for microbial abundances reported here were taken                      |                             |
| 219 | from the top 10 m integrated samples. For abundances from 0-17 m and the surrounding water see                    |                             |
| 220 | Supplementary data (Table S1 and Fig.S1).                                                                         |                             |
| 221 |                                                                                                                   |                             |
| 222 | The experimentexperimental period has been divided into 4 four phases based on major physical and                 | Formatted: Justified        |
|     | The experiment experimental period has been arriaded into T rear phases based on major physical and |                             |
| 223 | biological changes occurring (Paul et al., 2015). Phase 0 before $CO_2$ addition (days -5 to 0),                  |                             |
| 224 | phasePhase I (days 1-16), phasePhase II (days 17-2230) and phasePhase III (days 2331-43).                         |                             |
| 225 | Throughout this studymanuscript the data are presented using 3three colors (blue, grey and red),                  |                             |
| 226 | representing low (mesocosms M1 and M5) intermediate (M6 and M7) and high (M3 and M8) $fCO_2$                      |                             |
| 227 | concentrations (Table 1).                                                                                  |                             |
| 228 |                                                                                                                   |                             |
| 229 | 2.2 Microbial abundances                                                                                          |                             |
|     |                                                                                                                   |                             |

<mark>1</mark>9

l

[revised manuscript text omitted]

| 333 | and tracked over time (24 h). Wilhelm et al. (2002). Here free viruses are removed from a sample of                    | Formatted: English (U.K.)                                               |
|-----|------------------------------------------------------------------------------------------------------------------------|-------------------------------------------------------------------------|
| 334 | prokaryotes, samples are then taken every 3 hours for 24 hours for virus enumeration. Any viruses in                   |                                                                         |
| 335 | the samples must come from lysing bacteria and thus the rate of bacterial lysis can be estimated                       |                                                                         |
| 336 | using an appropriate burst size. Briefly, free viruses were removedBriefly, free viruses were reduced                  |                                                                         |
| 337 | from a 300 ml sample of whole water by re-circulation over a 0.2 $\mu m$ pore size polyether sulfone                   |                                                                         |
| 338 | membrane (PES) tangential flow filter (Vivaflow 50, Vivascience) at a filtrate expulsion rate of 40 ml                 |                                                                         |
| 339 | min -1 . A total of 900 ml of The concentrated sample was then reconstituted to the original volume         |                                                                         |
| 340 | using virus-free seawater, freshly produced by 30 kDa ultrafiltration using a PES membrane (Vivaflow                   |                                                                         |
| 341 | 200, Vivascience) was added in . This process was repeated a total of three stepstimes to gradually                    |                                                                         |
| 342 | wash away free viruses. Finally the sample was diluted back to the original 300 ml volume with virus-                  |                                                                         |
| 343 | free seawater. The samples After the final reconstitution, 50 ml aliquots were aliquoted distributed                   |                                                                         |
| 344 | into six <del>50 ml</del> -polycarbonate tubes. MytomycinMitomycin C (Sigma-Aldrich) (final concentration, 1 μg |                                                                         |
| 345 | ml -1 , -maintained at 4_°C), which induces lysogenic bacteria (Weinbauer and Suttle, 1996) was added       | Formatted: Dutch (Netherlands)                                          |
| 346 | to three a second series of the six tubes triplicate samples for each mesocosm-studied. A third series                 |                                                                         |
| 347 | of incubations with 0.2 $\mu m$ filtered samples was used as a control for viral loss (e.g. viruses adhering           |                                                                         |
| 348 | to the tube walls) and showed no significant loss of free viruses during the incubations. At the start of              |                                                                         |
| 349 | the experiment, 1 ml subsamples were immediately removed from each tube and fixed as previously                        |                                                                         |
| 350 | described for viral and bacterial abundance. The samples were dark incubated at in situ temperature             | Formatted: Font: Not Italic                                             |
| 351 | in the dark-and 1 ml subsamples were then-taken after 3h, 6h, 9h, 12hat 3 h, 6 h, 9 h, 12 h and 24h.                   |                                                                         |
| 352 | Viruses were later enumerated by the method of Brussaard (2004) to determine their rate of                             |                                                                         |
| 353 | production over time. 24 h. Virus production was determined from linear regression of viral                            |                                                                         |
| 354 | abundance over time (time period used for regression analysis may vary. Viral production due to                        |                                                                         |
| 355 | induction of lysogeny was calculated as the difference between sampling days, depending on the                         | Formatted: Font color: Black, English (U.K.), Pattern: Clear (White)    |
| 356 | temporal virus abundance dynamics).production in the unamended samples and production of                               |                                                                         |
| 357 | samples to which mitomycin C was added. Although mortality experiments were performed with                             | Formatted: Font color: Black, English
(U.K.), Pattern: Clear (White) |
| 358 | initially planned to be employed for mesocosms 1, 2, and 3 as representing low, mid and high $fCO_2$                   |                                                                         |
|     | <del>1</del> 14                                                                                                        |                                                                         |

| 359 | conditions, mesocosm 2 was lostcompromised due to leakageDueAdditionally, due to logistical                                                                    |                 |                                    |
|-----|----------------------------------------------------------------------------------------------------------------------------------------------------------------|-----------------|------------------------------------|
| 360 | reasons weassays were only able to perform these assaysperformed until day 21.                                                                                 |                 |                                    |
| 261 |                                                                                                                                                                |                 |                                    |
| 301 |                                                                                                                                                                |                 |                                    |
| 362 | To determine grazing rates on prokaryotes, fluorescently labeled bacteria (FLBFLBs) were prepared                                                              |                 | Formatted: Font: 11 pt             |
|     |                                                                                                                                                                |                 |                                    |
| 363 | from <del>cultured Halomonas haloduransenriched natural bacterial assemblages (originating from the</del>                                               |                 |                                    |
| 364 | North Sea), labeled with 5945-([4,6-Dichlorotriazinyl AminofluoresceinDichlorotriazin-2-yl]amino)                                                              |                 | Formatted: Font: 11 pt, Not Italic |
| 265 | fluorensis (DTAE, 2000) General Advict 40 up m 1-1 according to Charrond Charret et (1992). Frequen                                                 | $\overline{\ }$ | Formatted: Font: 11 pt             |
| 365 | riuorescein (DTAF, 36565 Sigma-Aldrich 40 μg mi ) according to Sherr <del>and Sherret al. (1993). Frozen</del>                            |                 | Formatted: Font: 11 pt             |
| 366 | ampoules containing preyof FLB (1-5 % of total bacteriabacterial abundance) were added to triplicate                                                           | $\backslash$    | Formatted: Font: 11 pt             |
| 367 | 1 Linculation bottles containing whole water gently passed through 200 um mech. Twenty                                                                         |                 | Formatted: Font: 11 pt             |
| 307 | I E incubation bottles containing whole water gently passed through 200 µm mesh. Twenty                                                                        |                 |                                    |
| 368 | milliliterml samples were taken immediately after addition (0 h) and the headspace was removed by                                                              |                 |                                    |
| 369 | gently squeezing air from the bottle <del> so that no air bubble remained. The. The 1 L bottles were</del>                                       |                 |                                    |
| 370 | incubated on a slow turning wheel (1 rpm) at in situ light and temperature conditions (representative                                                          |                 |                                    |
| 371 | of 5 m depth) for 24 h. Sampling was repeated after 24 h. All samples were fixed <del>with</del> to a 1 % final                                                |                 |                                    |
|     |                                                                                                                                                                |                 |                                    |
| 372 | concentration $0.2 \ \mu m$ filtered of gluteraldehyde (0.2 \ \mu m filtered; 25 % EM-grade, 25%) and), stained                                                |                 |                                    |
| 373 | with-(in the dark for 30 min at 4 °C) with 4',6-Diamidino-2-Phenylindole, Dihydrochloride (DAPI)                                                               |                 |                                    |
| 374 | solution ( 0.2 μm filtered- <del>(;</del> Acrodisc ® <del>25mm</del> 25 mm Syringe filters, PALL Life Sciences <del>) DAPI at a</del> ; 2 μg |                 |                                    |
| 375 | ml -1 , final concentration- <del>of 2 μg ml-1; (</del> Sherr et al., 1993) Samples were incubated for 30 min at 4°C                     |                 | Formatted: Superscript             |
|     |                                                                                                                                                                |                 | Formatted: Dutch (Netherlands)     |
| 376 | and stored in the dark. The 1 L bottles were incubated on a slow turning wheel (1 rpm) at in situ light                                                 |                 |                                    |
| 377 | and temperature conditions for 24 h. 24 h samples were then taken in the same manner as for 0 h.                                                               |                 |                                    |
| 378 | Samples were and filtered onto 25 mm, 0.2 $\mu$ m black polycarbonate filters (GE Healthcare life                                                              |                 |                                    |
| 379 | sciences <del>), ). Filters were then mounted on microscopic slides and stored at -20_°C until analysis. FLBs</del>                                            |                 | Formatted: Font: 11 pt             |
| 380 | present on a $\approx 0.75$ mm 2 area were counted using a Zeiss Axioplan 2 microscopeGrazing (µ                                                    |                 | Formatted: Font: 11 pt             |
|     |                                                                                                                                                                | $\langle $      | Formatted: Font: 11 pt             |
| 381 | $d^{-1}$ ) was measured according to                                                                                                                           |                 | Formatted: Font: 11 pt             |
| 202 | NI NI X Ut                                                                                                                                                     |                 | Formatted: Font: 11 pt             |
| 382 | $N_{T24} = N_{T0} + e^{-t}$                                                                                                                                    |                 | Formatted: Font: 11 pt             |
|     |                                                                                                                                                                |                 | Formatted: Font: 11 pt             |
| 383 | where $N_{T24}$ and $N_{T0}$ are the number of FLBs present at 24 h and 0 h, respectively.                                                                     | $\nvdash$       | Formatted: Justified               |
|     | ± 15                                                                                                                                                    |                 |                             |

| 385 | 2.4 Statistics                                                                                               |
|-----|--------------------------------------------------------------------------------------------------------------|
| 386 | Non-metric multidimensional scaling (NMDS) was used to follow microbial community development                |
| 387 | in each mesocosm over the experimental period. NMDS is an ordination technique which represents              |
| 388 | the dissimilarities obtained from an abundance data matrix in a 2-dimensional space (Legendre and            |
| 389 | Legendre, 1998). In this case, the data matrix was comprised of abundance data for each                      |
| 390 | phytoplankton group in each mesocosm for every day of sampling. The treatment effect was                     |
| 391 | assessed by analysis of similarity (ANOSIM; Clarke, 1993) and inspection of the NMDS biplot. ANOSIM          |
| 392 | compares the mean of ranked dissimilarities of mesocosms between fCO 2 treatments (low: 1, 5, 7;  |
| 393 | high: 6, 3, 8) to the mean of ranked dissimilarities within treatments per phase. The NMDS plots             |
| 394 | allowed divergence periods in the development and community composition between treatments to                |
| 395 | be visually assessed (period 1 from day 3-13 and period 2 from days 16-24). Net growth rates of each         |
| 396 | of the different microbial groups were calculated for these identified divergence periods.                   |
| 397 | Relationships between net growth rates and peak cell abundances with fCO 2 were evaluated by      |
| 398 | linear regression against the average fCO 2 per mesocosm during each period or peak day. A        |
| 399 | generalized linear model was used to test the relationship between prokaryote abundance and                  |
| 400 | carbon biomass with an ARMA correlation structure of order 3 to account for temporal                         |
| 401 | autocorrelation. The model fulfilled all assumptions such as homoscedasticity and avoiding                   |
| 402 | autocorrelation of the residuals (Zuur et al., 2007). A significance threshold of $p \leq 0.05$ was used and |
| 403 | significance is denoted by an asterisk (*). All analyses were performed using the statistical software       |
| 404 | program R, using packages nlme (Pinheiro et al., 2017) and vegan (Oksanen et al., 2017) (R core              |
| 405 | Team, 2017). Where average of low and high mesocosm abundance data are reported, values                      |
| 406 | represent the average of mesocosms 1, 5, 7 (mean fCO 2 365-497 µatm) and 6, 3, 8 (821-1231 µatm), |
| 407 | respectively.                                                                                                |
| 408 |                                                                                                              |

409 3 Results

I

384

| 410 | 3.1 Total phytoplankton dynamics in response to CO 2 enrichment                                                                     |  |  |
|-----|------------------------------------------------------------------------------------------------------------------------------------------------|--|--|
| 411 | During Phase 0, low variability in phytoplankton abundances of the different mesocosms (1.5 $\pm$ 0.05 x                                       |  |  |
| 412 | 10 5 ml -1 ) indicated good replicability of initial conditions prior to CO 2 manipulation (Fig. 1). This was |  |  |
| 413 | further supported by the high similarity between microbial communities of the different mesocosms                                              |  |  |
| 414 | as indicated by the tight clustering of points in the NMDS plot during this period (Fig. 2). During                                            |  |  |
| 415 | Phase 0, the phytoplankton community (<20 $\mu$ m) was dominated by pico-sized autotrophs, with the                                            |  |  |
| 416 | prokaryotic cyanobacteria Synechococcus (SYN) and Pico-I accounting for 69 % and 27 % of total                                                 |  |  |
| 417 | abundance, respectively. After $CO_2$ addition, there were two primary peaks in phytoplankton, which                                           |  |  |
| 418 | occurred on day 4 in Phase I and day 24 in Phase II (Fig. 1a). Microzooplankton grazing rates were                                      |  |  |
| 419 | estimated from the regression coefficient of the apparent                                                                                      |  |  |
| 420 | growth rate versus fraction of natural seawater for the 0.45 $\mu$ m series, with the combined rate of                                         |  |  |
| 421 | viral induced lysis and microzooplankton grazing being estimated from a similar regression for the                                             |  |  |
| 422 | 30 kDa series (Baudoux et al., 2006; Kimmance and Brussaard, 2010). The phytoplankton community                                                |  |  |
| 423 | became significantly different over time in the different treatments (ANOSIM, p=0.01, Fig. 2). Two                                             |  |  |
| 424 | periods were identified based on their divergence (Fig.2), the first (NMDS-based period 1) followed                                            |  |  |
| 425 | the initial peak in abundance (days 3-13) with highest abundances occurring in the elevated $CO_2$                                             |  |  |
| 426 | mesocosms (Fig. 1a). During the second period (NMDS-based period 2, days 16-24), abundances                                                    |  |  |
| 427 | were higher in the low fCO 2 mesocosms (Fig. 1a). In general the NMDS plot shows that throughout                                    |  |  |
| 428 | the experiment, mesocosm M1 followed the same basic trajectory as mesocosms M5 and M7, whilst                                                  |  |  |
| 429 | mesocosm M3 followed M6 and M8 (Fig. 2). Thus, the two mesocosms (representing high and low                                                    |  |  |
| 430 | fCO2treatments) deviated from each other during Phase I and were clearly separated during Phases II                          |  |  |
| 431 | and III (Fig. 2).                                                                                                                              |  |  |
| 432 | Phytoplankton abundances in the surrounding water started to differ from the mesocosms during                                                  |  |  |
| 433 | Phase 0 (on average 44 % lower) which was primarily due to lower abundances of SYN. This effect                                                |  |  |
| 434 | was seen from day -1, prior to CO 2 addition but following bubbling with compressed air (day -5). On                                |  |  |
| 435 | day 15, a deep mixing event occurred as a result of storm conditions (with consequent alterations in                                           |  |  |
|     |                                                                                                                                                |  |  |

**410 3.1 Total phytoplankton dynamics in response to CO2 enrichment**

| 436 | temperature and salinity) and as a result phytoplankton abundances in the surrounding open water                           |  |  |
|-----|----------------------------------------------------------------------------------------------------------------------------|--|--|
| 437 | diverged more strongly from the mesocosms but remained similar in their dynamics (Fig. 3).                                 |  |  |
| 438 | Microbial abundances in the 0-17 m samples were slightly lower but showed very similar dynamics to                         |  |  |
| 439 | those in the 0-10 m samples (Fig. S1).                                                                                     |  |  |
| 440 | A significant difference between the two regression coefficients (as tested by analysis of covariance)                     |  |  |
| 441 | indicated a significant viral lysis rate. Phytoplankton gross growth rate, in the absence of grazing and                   |  |  |
| 442 | viral lysis, was derived from the y intercept of the 30-kDa series regression. Similarly significant                       |  |  |
| 443 | differences between mesocosms M1 and M3 were determined by analysis of covariance of                                       |  |  |
| 444 | regression lines of the dilution series for the two mesocosms. Students T-tests were used to                               |  |  |
| 445 | determine significant differences between mesocosms for other parameters.                                                  |  |  |
| 446 |                                                                                                                            |  |  |
|     |                                                                                                                            |  |  |
| 447 |                                                                                                                            |  |  |
| 448 | <del>3 Results</del>                                                                                                       |  |  |
| 449 | 3.1 Phytoplankton population dynamics                                                                                      |  |  |
| 450 | Phytoplankton (total) showed two main peaks in abundance day 4, (phase I) and day 24 (phase II; Fig.                       |  |  |
| 451 | 1a)-Generally abundances were similar in all mesocosms except during the second half of phase I                            |  |  |
| 452 | (days 11–15) when they were greater at higher CO 2 and following this (days 17–22)greater at lower              |  |  |
| 453 | CO 2 concentrations. These trends were largely due to the prokaryotic cyanobacteria Synechococcus               |  |  |
| 454 | spp., making up on average 74% of total abundance. In contrast, the total eukaryotic phytoplankton                         |  |  |
| 455 | showed a strong positive effect of f CO 2 (Fig. 1b), due to the response of Pico I and II. Abundances in |  |  |
| 456 | the surrounding waters were more similar to the low fCO 2 than the high fCO 2 mesocosms,             |  |  |
| 457 | demonstrating that the differences between the low and high fCO 2 mesocosms are the effect of the               |  |  |
| 458 | elevated fCO 2 . Phytoplankton, prokaryotes and viral abundances in the 0-17m samples were                      |  |  |
| 459 | generally lower but showed similar dynamics (Figs. S1 and <mark>S2).</mark>                                                |  |  |
|     | 118                                                                                                                        |  |  |

**Comment [k2]:** Not sure that our plots allow us to really say this. Needs some backing up. See Figs of total abundances with Baltic

Comment [k3]: Probably but check

[revised manuscript text omitted]

displayed the largest variability (Fig. 3f), perhaps due to the spread of this cluster in flow cytographs
(which may indicate that this group represents several different phytoplankton species). No
significant relationship was found between net growth rate and fCO2 for this group for the two
NMDS-based periods (Table 2, Figs S2f and S3f) nor with the peak in abundances on day 17 (p=0.13,

| 563 | R2=0.46; Fig. S4h). Moreover, no consistent trend was detected in mortality rates (Fig. 4f). Similar to                 |                                                                                     |
|-----|-------------------------------------------------------------------------------------------------------------------------------------------|-------------------------------------------------------------------------------------|
| 564 | Nano-I, abundances in the surrounding water were often higher than in the mesocosms (max 3.5 x                                            |                                                                                     |
| 565 | $10^{2}$ ml -1 vs 1.1 x $10^{4}$ ml -1 , respectively; Figs. 3f and S6b).                                           |                                                                                     |
| 566 |                                                                                                                                           |                                                                                     |
| 567 | 3.1.4 Algal carbon biomass                                                                                                                |                                                                                     |
| 568 | The mean combined biomass of Pico-I and Pico-II showed a strong positive correlation with $fCO_2$ +-                                      | Formatted: Justified                                                                |
| 569 | throughout the experiment (p<0.05, R 2 =0.95; Fig. 5a), an effect already noticeable by day 2. Their                           |                                                                                     |
| 570 | biomass in the high $fCO_2$ mesocosms was, on average 11 % higher than in the low $fCO_2$ mesocosms                                       |                                                                                     |
| 571 | between days 10-20 and 20 % higher between days 20-39. Conversely, the remaining algal groups                                             |                                                                                     |
| 572 | showed an average 10 % reduction in carbon biomass at enhanced fCO 2 (days 3-39, the sum of SYN,                               |                                                                                     |
| 573 | Pico-III, Nano-I and II ; p<0.01; Fig. 5b). The most notable response was found for the biomass of                                        |                                                                                     |
| 574 | Pico-III, which showed an immediate negative response to CO 2 addition (Fig. S7a) and remained, on                             |                                                                                     |
| 575 | average, 29 % lower throughout the study period (days 2-39). For Nano-I and II the lower carbon                                           |                                                                                     |
| 576 | biomass only became apparent during the end of Phase I and beginning of Phase II (days 14-20; Fig.                                        |                                                                                     |
| 577 | S7b). Due to its small cell size, the numerically dominant SYN accounted for an average of 40 % of                                        |                                                                                     |
| 578 | total carbon biomass.                                                                                                                     | Formatted: Font: Bold                                                               |
| 579 |                                                                                                                                           |                                                                                     |
| 580 | 3.2 Prokaryote Synechococcus (SYN) showed an initial peak in abundance on day 4 (Fig. 2a), then                                           |                                                                                     |
| 581 | abundances declined, in all mesocosms until day 7. Between day 7 and 16 high CO 2 mesocosm                                     |                                                                                     |
| 582 | abundances stabilized but in the lower CO 2 mesocosms continued to drop until t12 before increasing                            |                                                                                     |
| 583 | again. This difference may be explained by higher grazing rates (no viral lysis detected), at lower CO 2                       | Comment [k4]: Although there is a strong correlation here there isn't really |
| 584 | as measured inM1 compared to M3 on day 10 (0.56 vs 0.27 d -1 )(Fig. 2b). Despite deviations in                                 | much difference in the actual abundances
for t4-7.                               |
| 585 | temporal dynamics between the treatments, SYN abundance peaked at day 24 in all mesocosms with                                            | Cut this statement and Fig 20                                                       |
| 586 | around 4.5 x 10 5 cells ml -1 (Fig 2a) and was negatively correlated with $fCO_2$ (R 2 =0.77). Total net |                                                                                     |
| 587 | production during this bloom was greater in the low f CO 2 mesocosms than in the high ones as initial                   |                                                                                     |
| 588 | abundances were lower (day 13) and peak abundances higher (day 24; Fig. 2a). The higher losses at                                         |                                                                                     |
|     | 173                                                                                                                                       |                                                                                     |
|     | 125                                                                                                                                       |                                                                                     |

| $\left  \right $ |
|------------------|
| Ć                |
|                  |
|                  |
|                  |
|                  |
|                  |
|                  |
|                  |
|                  |
|                  |
|                  |
|                  |
|                  |
|                  |
|                  |

Comment [k6]: Carry on working on this here Comment [k7]: Rewrite this

| 615 | may have stimulated the gross growth in M3 for a longer period in the high fCO 2 mesocosms as                                       |
|-----|------------------------------------------------------------------------------------------------------------------------------------------------|
| 616 | compared to M1 (day 19; Fig. 3b). Combined with higher losses at low fCO 2 a positive correlation of                                |
| 617 | net growth rates with fCO 2 was seen (Fig. 3f, R 2 =0.71), and almost 2-fold higher abundances at high                   |
| 618 | fCO 2 on day 21 (Fig. 3a, i, R 2 =0.84). Pico I was thus greatly stimulated by increased fCO 2 , from day 3   |
| 619 | throughout the experiment. Standing stock of Pico I remained higher at high fCO 2 for the further                                   |
| 620 | duration of the experiment (Fig. 3a), with gross growth matched by total losses (Fig.3b). Surprisingly                                         |
| 621 | the higher abundances did not stimulate higher losses during this period, grazing rates were very low                                          |
| 622 | in both M1 and M3, and viral lysis was totally responsible for losses on day 31 in both mesocosms                                              |
| 623 | <del>(Table S2).</del>                                                                                                                         |
| 624 |                                                                                                                                                |
|     |                                                                                                                                                |
| 625 | 3.1.3 Picoeukaryotes II                                                                                                                        |
| 626 | A group of larger picoeukaryotes, Pico II (mean diameter of 3 $\mu$ m) bloomed exactly during the period                                       |
| 627 | Pico I was low in standing stock (days 13-21, Fig. 4a) and the peak abundance (day 17) correlated                                              |
| 628 | positively with fCO 2 (Fig. 4d). Relatively high total losses of 0.46 and 0.58 d -1 in the low and high fCO 2 |
| 629 | mesocosms, respectively (average days 6-13) accompanied the high gross growth rates (0.69 and                                                  |
| 630 | 0.72 d -1 ) for the same period (Fig. 4b)These indicate high turnover and explain the slow rate of                                  |
| 631 | increase in cell abundance until day 13 (Fig. 4a)During the bloom period of Pico II, losses were                                               |
| 632 | smaller than the gross growth rate, more so it seems for M3 than M1 (Fig4b)Resultant net growth                                                |
| 633 | rates correlated with fCO 2 (Fig. 4d, R 2 =0.82) with peak abundances 1.4 fold higher at high fCO 2 (Fig.     |
| 634 | 4a ). Higher losses then contributed to the faster decline in abundances at high fCO 2 . Phase III was a                            |
| 635 | period of low turnover for Pico II with low gross growth and loss rates resulting in quite stable cell                                         |
| 636 | abundances, still higher at high $fCO_2$ , until day 29 after which they declined in all mesocosms (Fig.                                       |
| 637 | 4 <del>a).</del>                                                                                                                               |
| 638 |                                                                                                                                                |

I

**639 3.1.4 Picoeukaryotes III**

I

| 640 | Another group with around 2.9 $\mu$ m cell diameter could be discriminated from Pico II by its higher                                              |
|-----|----------------------------------------------------------------------------------------------------------------------------------------------------|
| 641 | orange autofluorescence, and as such may represent small sized cryptophytes. This is just at the                                                   |
| 642 | lower size range of small cryptophyte (Klaveness, 1989). This group (Pico III) had its highest                                                     |
| 643 | abundances during phases II and III (days 17-43, Fig. 5a), with a distinct negative correlation to fCO 2                                |
| 644 | (Fig. 5e, R 2 =0.91). Already directly upon the first CO 2 addition (days 0-4) the abundances declined for                   |
| 645 | the high $fCO_2$ mesocosms (Fig. 5a) with net growth rates negatively correlated to $fCO_2$ (Fig. 5d,                                              |
| 646 | R 2 =0.94). Gross growth rates were indeed significantly higher for M1 than M3 at days 1, 4 and 10                                      |
| 647 | (Fig5b)-Abundances of the Pico III group in the surrounding water followed the low fCO2                                                            |
| 648 | mesocosms perfectly during this first period, indicating that the crash in the high fCO 2 -mesocosms                                    |
| 649 | was indeed a direct (negative) effect of fCO 2 (Table S1). A similar response of Pico III abundance                                     |
| 650 | halting in the high fCO 2 mesocosms and strongly increasing in the low fCO 2 mesocosms occurred                              |
| 651 | directly after the additional fCO 2 -purge (day 15). Losses were largely due to microzooplankton                                        |
| 652 | grazing. Unfortunately about half of the loss assays in the second half of the experiment failed (for                                              |
| 653 | unknown reasons), yet the successful assays suggest that losses were minor (Fig. 5b). There may also                                               |
| 654 | be larger cryptophytes present in the community, not counted by the flow cytometer because our                                                     |
| 655 | data show Pico III most dominant in phase III whilst the specific pigment data shows a decline from                                                |
| 656 | phases 0 to III.                                                                                                                                   |
| 657 |                                                                                                                                                    |
| 658 | 3.1.5 Nanoeukaryotes I                                                                                                                             |
| 659 | The nanoeukaryotes group Nano I consisted of cells with a mean diameter of 5.2 $\mu m$ and were found                                              |
| 660 | with maximum abundances of 5.5 x10 2 ml -1 (Fig. 6a). 6a). After an initial peak at day 6, the lower fCO 2 |
| 661 | mesocosms showed the highest numbers at day 17 (Fig. <del>6a).</del> This seems initiated by 2.3 fold higher                                       |
| 662 | total loss rates for M3 than M1 on days 6 and 10 (Fig. 6b) in combination with 2 fold lower gross                                                  |
| 663 | growth rates on day 10 (Fig. 6b)Ultimately, this led to net growth rates correlating negatively with                                               |

| 664 | fCO 2 for days 10 12 (Fig. 6d, R 2 =0.83). Viral lysis occurred predominantly in the high fCO 2 mesocosm |                      |
|-----|-------------------------------------------------------------------------------------------------------------------------------------------|----------------------|
| 665 | throughout the experiment with rates ranging from 0.13 to 0.7 day -1 (making up 16 to 98% of total                             |                      |
| 666 | losses; Table S2). A group of viruses which had a flow cytometric signal typical for viruses infecting                                    |                      |
| 667 | nanoeukaryotes (V4) were identified but no obvious correlation This was found with any of the                                             |                      |
| 668 | phytoplankton groups. Lower total loss rates at days 13 and 17 in both mesocosms allowed a small                                          |                      |
| 669 | increase in abundance, peaking on day 17 and negatively correlated to $fCO_2$ (Fig. 6e, $R^2$ =0.67).                                     |                      |
| 670 |                                                                                                                                           |                      |
| 671 | 3.1.6 Nanoeukaryotes II                                                                                                                   |                      |
| 672 | The temporal dynamics of Nano II were rather erratic (Fig. primarily 7a)Nano II were the largest in                                       |                      |
| 673 | size and may have been made up by different phytoplankton species, however due to their low                                               |                      |
| 674 | numbers we were unable to discriminate separate groups. The peak in abundance at day 16 showed                                            |                      |
| 675 | a negative correlation to $fCO_2$ (Fig. 7e, R 2 =0.61), and was the result of an overall reduced net growth                    |                      |
| 676 | rate with $fCO_2$ (Fig. 7d, R 2 =0.56). The subsequent decline seems the result of reduced gross growth                        |                      |
| 677 | rate (to even zero) and increased loss rate (day 20; Fig. increases in the HNA7b).                                                        |                      |
| 678 |                                                                                                                                           |                      |
| 679 | 3.1.7 Algal POC                                                                                                                           |                      |
| 680 | The calculated mean algal POC shows that fCO 2 had a clear positive effect on the biomass of Pico I+                           | Formatted: Justified |
| 681 | and II (Fig. 8a; p<0.0001). The effect became noticeable only a few days into the experiment and the                                      |                      |
| 682 | mean Pico I and II POC concentrations in the high fCO2 mesocosms stayed high for the entire                                               |                      |
| 683 | duration of the experiment. At the same time the remaining algal groups showed reduced POC at                                             |                      |
| 684 | enhanced fCO 2 (the sum of Pico III, and Nano I and II and Synechococcus spp.; Fig. 8b, p<0.01).                               |                      |
| 685 | Particularly Pico III showed a nearly instant and markedly negative response to increased fCO2                                            |                      |
| 686 | concentration (Fig. 53a)This was a lasting effect as the strongest difference was found in the second                                     |                      |
| 687 | half of the experiment. For Nano I and II the higher algal POC concentrations became only apparent                                        |                      |
|     | 427                                                                                                                                       |                      |

| 688 | from the end of phase I and during phase II (days 14 20; Fig. S3b). Due to its small cell size, the                                            |                        |
|-----|------------------------------------------------------------------------------------------------------------------------------------------------|------------------------|
| 689 | numerically dominant SYN accounted on average for 40% of total POC. Due to the exclusion of 3                                                  |                        |
| 690 | mesocosms (see Material and Methods), the number of fCO 2 treatments is reduced to 6, which limits                                  |                        |
| 691 | the statistical power of the results. Still, our data show that the responses of the different                                                 |                        |
| 692 | phytoplankton groups to ocean acidification were evident and consistent-                                                                       | Formatted: Font: Bold  |
| 693 |                                                                                                                                                |                        |
| 694 | 3.2 Prokaryote-population dynamics                                                                                                             |                        |
| 695 | The prokaryotic temporal dynamics in the mesocosms resembled that in the outside waters (Fig. S2).                                             |                        |
| 696 | In general prokaryote abundance in the mesocosms followed the total algal biomass, with an initial                                             |                        |
| 697 | increase during the first days following the closure of the mesocosms (Fig. 9a). The increase was                                              |                        |
| 698 | mainly due to the HDNAgroup (Fig. 6b) which displayed higher net growth rates (0.22 d -1 ) compared                                 |                        |
| 699 | to the LNA-prokaryotes (Fig. 9b). The total prokaryote abundance increased initially at a net growth                                           |                        |
| 700 | rate of 0.19 d -1 , and more specifically at 0.22 and 0.14 0.14 d -1 for days -3 to 3; Fig. 6c). A similar,       | Formatted: Superscript |
| 701 | albeit somewhat lower, increase was also recorded in the surrounding waters (Fig. 6a). the high and                                     |                        |
| 702 | low DNA prokaryotes respectively (Fig. 9b and c). There was no significant difference in prokaryote                                            |                        |
| 703 | abundance between the treatments at the first peak (day 4). However, grazing was significantly                                                 |                        |
| 704 | lower (0.3 d -1 ) in high (M3) than in low (M1; 0.5 d -1 ) CO 2 treatments, on both days 0 and 4, and viral   |                        |
| 705 | lysis 3% higher at high CO 2 (Figs. 10b and c). The decline in prokaryote The decline of the first peak in                          |                        |
| 706 | prokaryote abundances from days 5 to 9 seemed due to decliningcoincided with the decay in                                                      |                        |
| 707 | phytoplankton abundance/ biomass ( <del>FigFigs</del> . 1a <del>)</del> and increasing S7c). Concurrently the share of viral     |                        |
| 708 | lysis <del>rates (12-16 % d-1increased,</del> representing 37- 39_% of total <del>losses in M1 and 37% in M3</del> mortality |                        |
| 709 | on day 11 (Fig. 7b) Fig. 10c). Viral lysis assays showed no evidence No measurable rates of lysogeny                                           |                        |
| 710 | were found for the prokaryotic community during the experimentexperimental period (all phases).                                                |                        |
| 711 |                                                                                                                                                |                        |

| 710 | From doug 10, to 15 prolyments dynamics (total, UNA, and UNA) became already metiosphyle offected by                                          |
|-----|-----------------------------------------------------------------------------------------------------------------------------------------------|
| /12 | _From days 10- to_ 15 prokaryote dynamics (total, HNA and LNA) became <del>cicanyhoticeably affected by•</del>           |
| 713 | fCO 2 -CO 2 concentration with significantly higher abundances and net growth rates at higher fCO 2 (Fig.    |
| 714 | 9a). Both the HDNA and the LDNA-prokaryotes (peak abundance on day 13, Fig. 9b and c) showeda                                                 |
| 715 | significant positive correlation with fCO 2 (R 2 = 0.92 and 0.79, respectively, total prokaryote R 2 = 0.88, |
| 716 | Fig. 10d). between net growth and fCO 2 during Phase I (days 3-13 NMDS-based period 1; Table 2, Fig.                               |
| 717 | S2 g and h). In the higher $fCO_2$ mesocosms, the decline in prokaryote abundance following the peak                                   |
| 718 | at dayoccurring between days 13 and 16 (Fig. 6a) was largely the result of (70 %) due to decreasing                                           |
| 719 | HDNA HNA -prokaryote numbers (Fig. 6b). 9b). GrazingThe grazing was indeed significantly1.6-fold                                |
| 720 | higher in the high fCO 2 mesocosm M3 but the data for viral lysis were inconclusive due compared to a                       |
| 721 | failed assay (for technical reasons) for M1 at(0.36 $\pm$ 0.13 and 0.14 $\pm$ 0.08 d -1 on day 14 (; Fig. 7a). 10b                 |
| 722 | and c). The significantly higher viral abundances, particularly due to At the V3 group with highest                                    |
| 723 | green fluorescence, for the high fCO 2 mesocosms around thatsame time (Figs. 11a and b) seem to                                    |
| 724 | indicate that viral lysis, virus abundance increased in the high fCO 2 mesocosms was higher. (Fig. 6d).                            |
| 725 |                                                                                                                                               |

726 During phasePhase II, prokaryote abundances increased steadily until day 24 (for both HDNAHNA+ 727 and LDNALNA), corresponding to increased algal biomass (Fig. 10e)Figs. 6 and lowS7c) and lowered 728 grazing rates (0.1-0.2 d+; Fig. 10b). Although the overall higher prokaryote standing stock in the low 729 fCO2 mesocosms was due to enhanced growth around day 16 (Fig. 9a), the net growth rates were 730 comparable after day 17. Moreover, the higher abundances were only found for the HDNA-731 prokaryotes (Fig. 9b and c). Viral lysis rates were higher for the low fCO2 mesocosms (Fig. 10c). The 732 higher prokaryote abundances in the low fCO2 mesocosms appear thus due to the lower grazing prior 733 to the increase, i.e. at the end of phase I (day 14). Fig. 7a). Specifically, during days 16-24 (NMDS-734 based period 2), the HNA-prokaryotes showed an average 10 % higher abundances in the low, as 735 compared to the high fCO2 mesocosms (Fig. 6b). However, a significant negative correlation of net 736 growth rates and fCO2 was only found for LNA (Table 2, Fig S3g and h). No significant differences in 737 loss rates between M1 and M3 were found during Phase II (p=0.22, 0.46 days 18 and 21 respectively;

| 738 | Fig. 7). Halfway through Phase II (day 24), the prokaryote abundance in the surrounding water                                                 |
|-----|-----------------------------------------------------------------------------------------------------------------------------------------------|
| 739 | leveled off (Fig. 6a). Prokaryote abundance ultimately declined again-during days 28-35, but less in                                          |
| 740 | M1 than in the other mesocosms (Fig. 9a). We unfortunately have no data of the prokaryote loss                                                |
| 741 | rates (Fig. 6a), whereby the net growth of LNA was again negatively correlated with enhanced CO 2                                  |
| 742 | (p=0.02, R 2 =0.76; Table 2, Fig S3g). Unfortunately, no experimental data on grazing and lysis of                                 |
| 743 | prokaryotes is present after day 25, however. However, viral abundances increased steadily at a                                               |
| 744 | steady rate of $2.2 \times 10^6 \text{ d}^{-1}$ (to a maximum of $0.9 \times 10^8$ ml -1 by day 39; Fig. 11a), implying that viral |
| 745 | lysis was at least partly responsible for the, concomitant with a decline in prokaryote abundance.                                            |
| 746 | (Fig. 6a and d). There was no significant difference in correlation between viral abundances between                                          |
| 747 | the treatmentsand fCO 2 during this period.Phases II and III (p=0.36, R 2 =0.21).                                       |
| 748 |                                                                                                                                               |
|     |                                                                                                                                               |

**750 4 Discussion**

749

751 In most experimental mesocosm studies, nutrients have been added to stimulate phytoplankton 752 growth (Schulz et al., 2017) therefore little data exists for oligotrophic phytoplankton communities. 753 In this study, we describe the impact of increased fCO2 on the brackish Baltic Sea microbial 754 community during summer (nutrient depleted; Paul et al., 2015). Small-sized phytoplankton 755 numerically dominated the autotrophic community, in particular SYN and Pico-I (both about 1 µm 756 cell diameter). Our results demonstrate variable effects of fCO2 manipulation on temporal 757 phytoplankton dynamics, dependent on phytoplankton group. In particular, Pico-I and Pico-II showed 758 significant positive responses, whilst the abundances of Pico-III, SYN and Nano

---

## Author Response (AR4)

*Dear Prof. Achterberg,*

*Thank-you for reviewing this manuscript, the previous reviews indeed helped us to strengthen the manuscript and we are very pleased that it is now acceptable for publication. We have made the minor amendments suggested, as can be seen in the marked up version attached, and addressed the one comment (in blue italic).*

Associate Editor Decision: Publish subject to minor revisions (Editor review) (21 Jun 2017) by E.P. Achterberg

Comments to the Author:

The resubmitted manuscript has substantially improved following two rounds of reviews. In particular the substantial and thorough amendments to the manuscript following the latest reviews have greatly improved the manuscript. The text flows well, and the scientific reasoning is well laid out and expressed.

The use of statistical techniques to show the different behaviours with time of the various phytoplankton communities that were subjected to different CO2 treatments has strongly improved the manuscript.

The authors have followed the recommendations by the reviewers and thereby greatly strengthened the manuscript.

I suggest a number of minor amendments to the manuscript before it can be published.

Line 29: replace dissolution with uptake *amended*

Line 29: after carbon dioxide write (CO2) *amended*

Line 29: replace in with by *amended*

Line 30: replace atmosphere with ocean *amended*

Line 32: replace highly with strongly *amended*

Line 37: replace CO2 with fugacity of CO2 (fCO2) *amended*

Line 39: their abundances with its abundance *amended*

Line 429: between "shift towards" place with increasing fCO2 *amended*

Line 50: replace activities with emissions *amended*

Line 50: write: primarily caused by the burning…… *amended*

Line 56: replace productivity with production *amended*

Line 81: replace nutrients with nutrient *amended*

Line 81: replace concentration with concentration *amended*

Line 84: write: carbon sequestration in deep waters and sediments *amended*

Line 101: replace under with with *amended*

Line 101: write: temperature conditions *amended*

Line 117: replace hung with reached *amended*

Line 127: replace fCO2 with fugacity of CO2 (fCO2) *amended*

Line 134: write: threshold fCO2 level, if present…… *amended*

Line 136: write: Initial fCO2 was 240…… *amended*

Line 166: replace concentrations with levels *amended*

Line 300 and 302: replace of with in *amended*

Line 306: write: phytoplankton abundance *amended*

Line 337: write: low fCO2 compared to high fCO2 *amended*

Line 448: replace on with for *amended*

Line 499: is there evidence from other mesocosm experiments of stimulation of SYN due to bubbling or enclosing? *Yes, this has been seen in a few other mesocosm experiments, we have added this into the text: "Alternatively, effects of enclosure and the techniques (bubbling) used to ensure a homogenous water column may have stimulated SYN within the mesocosms, which has been found to occur in several mesocosm experiments (Paulino et al., 2008; Gazeau et al., 2017)."(Line 498-501).*

Line 504: replace occur with upwell *amended*

Line 660: replace predicted with projected *amended*

[revised manuscript text omitted]